# Massively parallel characterization of transcriptional regulatory elements

Vikram Agarwal[1,2,19]✉, Fumitaka Inoue[3,4,5,19], Max Schubach[6], Dmitry Penzar[7,8,9], Beth K. Martin[1], Pyaree Mohan Dash[6], Pia Keukeleire[10], Zicong Zhang[5], Ajuni Sohota[3,4], Jingjing Zhao[3,4], Ilias Georgakopoulos-Soares[11], William S. Noble[1,12], Galip Gürkan Yardımcı[1,13,14], Ivan V. Kulakovskiy[7,8,15], Martin Kircher[6,10], Jay Shendure[1,16,17,18]✉ & Nadav Ahituv[3,4]✉

The human genome contains millions of candidate *cis*-regulatory elements (cCREs) with cell-type-specific activities that shape both health and many disease states[1]. However, we lack a functional understanding of the sequence features that control the activity and cell-type-specific features of these cCREs. Here we used lentivirus-based massively parallel reporter assays (lentiMPRAs) to test the regulatory activity of more than 680,000 sequences, representing an extensive set of annotated cCREs among three cell types (HepG2, K562 and WTC11), and found that 41.7% of these sequences were active. By testing sequences in both orientations, we find promoters to have strand-orientation biases and their 200-nucleotide cores to function as non-cell-type-specific 'on switches' that provide similar expression levels to their associated gene. By contrast, enhancers have weaker orientation biases, but increased tissue-specific characteristics. Utilizing our lentiMPRA data, we develop sequence-based models to predict cCRE function and variant effects with high accuracy, delineate regulatory motifs and model their combinatorial effects. Testing a lentiMPRA library encompassing 60,000 cCREs in all three cell types further identified factors that determine cell-type specificity. Collectively, our work provides an extensive catalogue of functional CREs in three widely used cell lines and showcases how large-scale functional measurements can be used to dissect regulatory grammar.

Sequence variation in *cis*-regulatory elements (CREs) is a major cause of human disease[2]. For example, the majority of genome-wide association studies (GWAS) implicate noncoding haplotypes bearing distal CREs, such as enhancers, in common diseases[2,3]. However, predicting the functional effects of nucleotide variation in CREs remains challenging. One of the major limitations is the lack of a comprehensive functional delineation of the probably millions of CREs in the human genome, many of which have tissue- or cell-type-specific activity. This impediment also limits the ability to develop machine learning tools that can predict tissue-specific CRE activity with high precision.

The emergence of genome-scale biochemical technologies to globally catalogue regions of open chromatin, transcription factor binding, histone modifications and mRNA expression levels has provided a framework to investigate gene regulatory and transcriptional landscapes in hundreds of cell types[4]. These efforts have led to the discovery of millions of cCREs in the human genome. However, these approaches are overwhelmingly descriptive and cannot confirm that any given cCRE is functional.

Massively parallel reporter assays (MPRAs) overcome these limitations by testing thousands of sequences or variants for regulatory activity in a multiplex fashion[5]. Previous work has utilized MPRAs or a derivative assay, the self-transcribing active regulatory region sequencing (STARR-seq), to test large numbers of cCREs for regulatory activity in human cells[6–9]. However, these assays rely on transient transfection, providing an episomal ('out of genome') readout, and are mostly limited to established cell types that can be robustly transfected and grown in large quantities. To address this, we developed a lentiMPRA[10], which enables reproducibility and multiplexability, extends to cell lines that are difficult to transfect such as neurons or organoids[11,12], and provides an 'in genome' readout. As lentiviral integrations are random, lentiMPRA measures the functional effect of cCREs averaged across different genomic locations. lentiMPRA is more strongly correlated with ENCODE annotations and sequence-based models[10] and provides higher cell-type

[1]Department of Genome Sciences, University of Washington, Seattle, WA, USA. [2]mRNA Center of Excellence, Sanofi, Waltham, MA, USA. [3]Department of Bioengineering and Therapeutic Sciences, University of California, San Francisco, San Francisco, CA, USA. [4]Institute for Human Genetics, University of California, San Francisco, San Francisco, CA, USA. [5]Institute for the Advanced Study of Human Biology (WPI-ASHBi), Kyoto University, Kyoto, Japan. [6]Berlin Institute of Health at Charité-Universitätsmedizin Berlin, Berlin, Germany. [7]Vavilov Institute of General Genetics, Russian Academy of Sciences, Moscow, Russia. [8]Institute of Protein Research, Russian Academy of Sciences, Pushchino, Russia. [9]Institute of Translational Medicine, Pirogov Russian National Research Medical University, Moscow, Russia. [10]Institute of Human Genetics, University Medical Center Schleswig-Holstein, University of Lübeck, Lübeck, Germany. [11]Institute for Personalized Medicine, Department of Biochemistry and Molecular Biology, The Pennsylvania State University College of Medicine, Hershey, PA, USA. [12]Paul G. Allen School of Computer Science and Engineering, University of Washington, Seattle, WA, USA. [13]Knight Cancer Institute, Oregon Health and Science University, Portland, OR, USA. [14]Cancer Early Detection Advanced Research Center, Oregon Health and Science University, Portland, OR, USA. [15]Life Improvement by Future Technologies (LIFT) Center, Moscow, Russia. [16]Howard Hughes Medical Institute, Seattle, WA, USA. [17]Brotman Baty Institute for Precision Medicine, University of Washington, Seattle, WA, USA. [18]Seattle Hub for Synthetic Biology, Seattle, Washington, USA. [19]These authors contributed equally: Vikram Agarwal, Fumitaka Inoue. ✉e-mail: vikram.agarwal@sanofi.com; shendure@uw.edu; nadav.ahituv@ucsf.edu

specificity predictions than episomal MPRA[13]. Furthermore, systematic comparison of lentiMPRA to eight other MPRA designs found strong correlations with episomal MPRA and STARR-seq, but also differences[14]. However, a limitation of lentiMPRA has been the number of sequences or variants that could be tested in a single experiment[10].

Here we applied an optimized lentiMPRA method and confirmed the reproducibility and reliability of this technology to test more than 200,000 sequences in a single experiment, which covers a major fraction of cCREs for any given human cell type[15]. We applied this method to substantially expand MPRA data for three ENCODE cell types, human hepatocytes (HepG2), lymphoblasts (K562) and induced pluripotent stem cells (iPS cells; WTC11), to examine the relative orientation dependence of promoters and enhancers. In addition, we tested 60,000 sequences in all three cell lines. With these data, we characterize the activity effect of a core promoter region and train models that can predict regulatory and nucleotide variant activity. We identify both biochemical and sequence-based features that are associated with cell-type-specific activity and provide a catalogue of thousands of functional cCREs that advances our understanding of genotype-to-phenotype associations in gene regulatory sequences.

## Optimization of lentiMPRA

To scale up lentiMPRA[10,14], we revised our established protocol to add random barcodes to the assayed sequences during the library amplification step along with the minimal promoter[16] (Extended Data Fig. 1a). Subsequently, element–barcode associations were reconstructed through sequencing (Extended Data Fig. 1b) and analysed with MPRAflow[16]. To evaluate the robustness of this revised lentiMPRA approach, we designed two pilot libraries (Supplementary Fig. 1a). As DNase accessibility, centred on the midpoint of a peak, has been shown to be a good predictor of regulatory elements[17] and MPRA activity[13,18], we used it as our main selection criteria for cCREs. The first pilot library encompassed 9,372 elements in HepG2 cells and consisted of: (1) 9,172 cCREs, centred at DNase hypersensitivity peaks that did not overlap promoters; (2) 50 positive and 50 negative controls of synthetically engineered sequences (that is, engineered to have multiple binding sites for known transcription factors or no known binding sites, respectively)[19]; and (3) 50 positive and 50 negative controls of naturally occurring enhancers (sequences observed to exhibit high and low enhancer activity, respectively)[10]. The second pilot library encompassed 7,500 elements in K562 cells and consisted of: (1) 6,394 cCREs, centred at DNase hypersensitivity peaks that did not overlap promoters; (2) 290 positive and 276 negative controls, identified by coupling CRISPR interference (CRISPRi) to single cell RNA-sequencing measurements to identify functional enhancer-gene pairs[20]; (3) 250 negative controls derived from dinucleotide shuffling cCREs randomly selected from our library; (4) 50 positive and 200 negative controls of naturally occurring enhancers (sequences observed to exhibit high and low enhancer activity, respectively)[18]; and (5) 24 positive and 16 negative manually selected controls in loci of interest such as α-globin, β-globin, *GATA1* and *MYC*[21] (Supplementary Table 1).

These pilot lentiMPRA libraries were used to transduce cells in triplicate and barcodes were sequenced at the DNA and RNA levels as described[16]. Activity scores for each element were calculated as the $\log_2$-transformed normalized count of RNA molecules from all barcodes corresponding to the element divided by the normalized number of DNA molecules from all barcodes corresponding to the element (Extended Data Fig. 1c and Supplementary Table 2). We observed a range of 50–250 median barcodes per element in each replicate (Supplementary Fig. 1b), providing a large number of independent measurements for each element, and activity scores (that is, $\log_2$-transformed RNA/DNA ratios) that were highly concordant across replicates (0.88–0.96 Pearson correlation; Supplementary Fig. 1c,d). Averaging across the three replicates, the distribution of element activity scores was strongly

divergent between most positive and negative controls (Supplementary Fig. 1e). An exception to this trend was observed for controls derived from CRISPRi-based screening efforts[20,21], which exhibited a slightly weaker signal than positive controls, indicating that in our reporter assays and outside their epigenetic context, they were still capable of activating transcription (Supplementary Fig. 1e).

We next analysed both lentiMPRA libraries for functional enhancers. We found that 2,740 out of the 8,960 (30.6%) cCREs were more active than negative synthetic controls[19] (in HepG2) and 3,703 out of 6,315 (58.6%) cCREs were more active than shuffled negative controls (in K562) (5% false discovery rate (FDR)). However, the differences in the proportion of active cCREs between cell types should not be directly compared because the negative controls were different in each case—one represents shuffled controls and the other does not. Given the extensive previous work characterizing regulatory elements in the β-globin locus and the inclusion of these sequences in our K562 library, we evaluated whether our MPRA results reproduced the findings of previous studies for five previously characterized cCREs, termed HS1–5 (Extended Data Fig. 1d). Consistent with previous work[22,23], we observed that HS2 strongly activated transcription relative to HS1 and HS3–5 (Extended Data Fig. 1d). In summary, these pilot experiments confirmed the ability of our revised lentiMPRA protocol to measure regulatory activity with high precision and reproducibility.

## cCRE characterization with lentiMPRA

With our pilot libraries showing reproducible and robust results, we next set out to test whether our revised lentiMPRA approach could measure more than 200,000 sequences in a single experiment, comprising a major portion of cCREs of any given human cell type. Using a similar scheme as in our pilot library[16], we sought to test a combination of all known 19,104 protein-coding gene promoters as well as potential enhancers (DNase peaks that are not near the promoter) in both orientations (Fig. 1a). In HepG2 cells, we tested all promoters and 66,017 potential enhancers; in K562 cells, we tested all promoters and 87,618 potential enhancers; in WTC11 cells, owing to their reduced transduction efficiency, we tested 7,500 promoters and 30,121 of 83,201 potential enhancers (Fig. 1b and Methods). To further interrogate whether open chromatin is required for transcriptional activation, we additionally tested 14,918 heterochromatic regions in our K562 library, nominated by the ENCODE consortium from regions 1 Mb either side of the *GATA1*, *MYC*, *HBE1*, *LMO2*, *RBM38*, *HBA2* and *BCL11A* loci, which are known human disease-associated and erythroid lineage genes. Collectively, incorporating dinucleotide shuffled negative controls and other positive and negative controls identified in previous studies[18,19,24], we designed and tested a total of 164,307 elements in HepG2 cells, 243,780 elements in K562 cells and 75,542 elements in WTC11 cells (Fig. 1b and Supplementary Table 3).

We observed 20–50 median barcodes per element in each replicate among all cell types (Supplementary Fig. 2a); elements supported with at least 10 barcodes (our minimum threshold) exhibited a substantially reduced standard deviation across replicates (Supplementary Fig. 2b). Element activity scores were also strongly concordant across replicate pairs, with Pearson correlations of 0.94 (HepG2), 0.76 (K562) and 0.76 (WTC11) (Supplementary Table 4 and Supplementary Fig. 2c–e). Averaging across the three replicates, we also observed strong correlations among element activity scores between cCREs common to our pilot and large-scale libraries (Pearson $r = 0.94$ in HepG2 cells and $r = 0.81$ in K562 cells; Supplementary Fig. 2f). Similarly, visualizing the large-scale K562 library in the *HBE1* locus (Extended Data Fig. 1d) and the other six disease-associated loci (Extended Data Fig. 2 and Supplementary Fig. 3) confirmed strong agreement with the K562 pilot library, with the large-scale library having greater density and highlighting additional functional regulatory elements. The inter-replicate correlations for both large-scale libraries were lower than for the pilot libraries owing

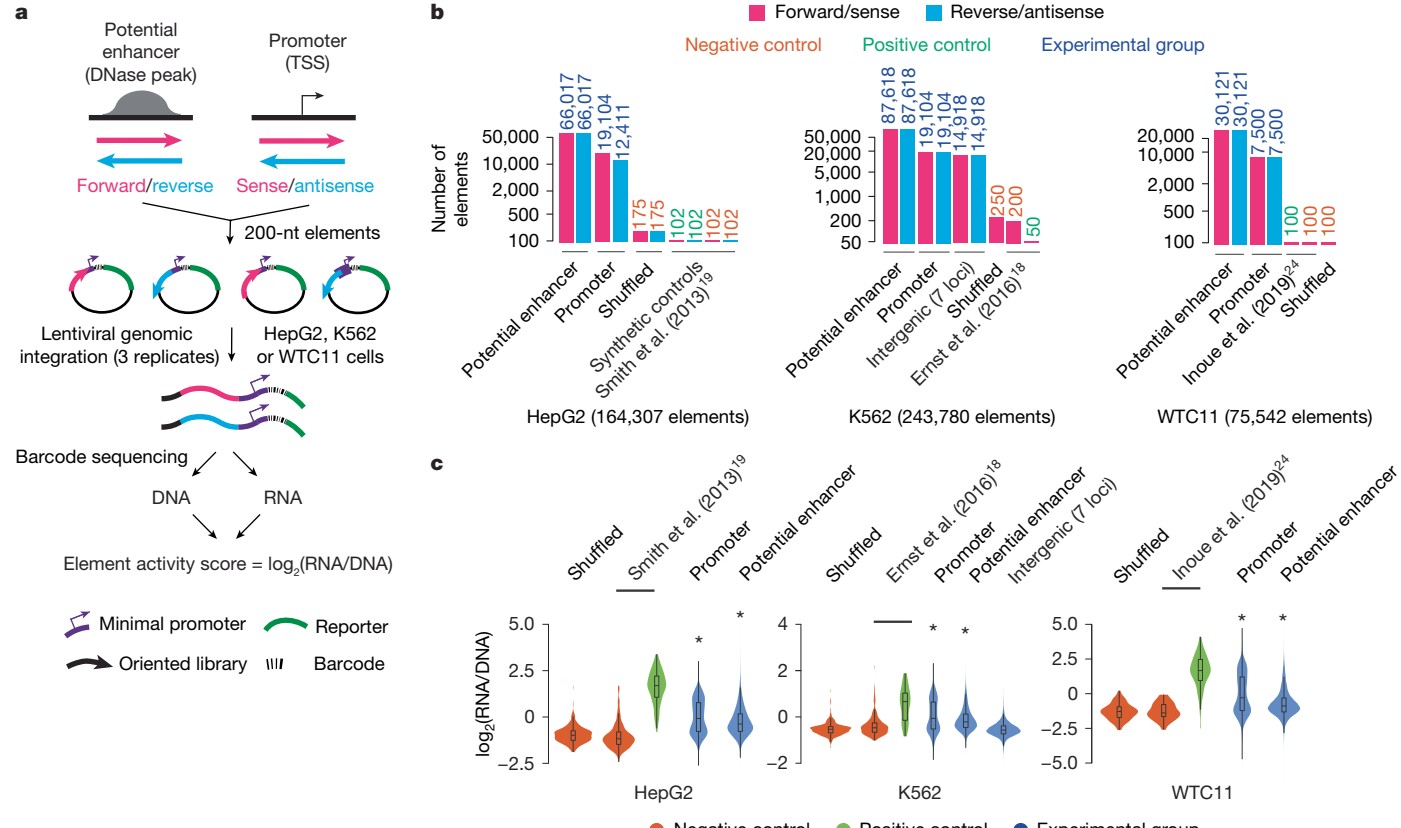

**Fig. 1 | lentiMPRA in three cell types. a**, lentiMPRA strategy for large-scale libraries. Thousands of cCREs including potential enhancers (marked by DNase peaks) and promoters (centred at the transcription start site (TSS)) are inserted into a reporter plasmid in both orientations together with barcodes. The libraries are infected into cells using lentivirus and the integrated DNA and transcribed RNA barcodes are sequenced to quantify cCRE activity. **b**, Composition of the HepG2, K562 and WTC11 libraries. Thousands of potential enhancers and promoters, negative controls and positive controls are included in each library[18,19,24]. Bars are coloured according to orientation tested and the number of tested elements is shown above the bars and coloured according to element type. **c**, Violin plots of element activity, measured as log$_2$-transformed RNA/DNA ratios for cCREs and negative and positive controls. Promoter and enhancer distributions were compared against the shuffled category using a one-sided Wilcoxon rank-sum test, followed by Bonferroni correction (*$P < 10^{-8}$).

to the trade-off between library size and per element sequencing depth. To further investigate this trade-off, we downsampled barcodes. Downsampling to 90% of the barcodes led to a near-perfect Spearman correlation with respect to element activity scores relative to those derived from the full dataset, whereas smaller proportions of the barcodes degraded this correlation (Supplementary Fig. 2g). The distributions of standard deviations for element activity scores across barcodes became tighter when considering larger barcode downsampling proportions (Supplementary Fig. 2h) and more replicates (Supplementary Fig. 2i).

The distribution of our fully processed element activity scores was strongly divergent between positive and negative controls in each cell type, with the majority of promoters and potential enhancers spanning the range in between the maximum positive control and minimum negative control scores (Fig. 1c). Promoters exhibited, on average, higher activity scores and a bimodal distribution compared with potential enhancers, which exhibited a right-skewed distribution in all cell types (Fig. 1c). This bimodal distribution was likely to be caused by inactive promoters exhibiting little to no activity in the MPRA. We next analysed all libraries to empirically measure the proportion of functional cCREs among each element type and cell type. Using shuffled controls as a background set in each cell type and both orientations of measured cCREs as a foreground set, we found that more than 50% of all promoter sequences had regulatory activity (HepG2: 11,367 out of 20,816 (54.6%); K562: 15,362 out of 29,376 (52.3%); WTC11: 5,038 out of 9,964 (50.6%); 5% FDR). For potential enhancers, we found as many as 42% to be active (HepG2: 50,714 out of 118,433 (42.8%); K562: 69,820 out of 169,260 (41.3%); WTC11: 11,861 out of 45,942 (25.8%); 5% FDR). An additional

power analysis indicated that our ability to detect functional regulatory elements was plateauing after barcode downsampling, suggesting that additional sequencing depth would be unlikely to lead to substantially altered estimates (Supplementary Fig. 2j).

To assess whether our potential enhancers could be used to validate significant enhancer–promoter interactions and/or predict CRISPRi results, we intersected this set against those tested in three different CRISPRi perturbation studies[20,25,26]. We examined the proportion of our binned activity scores that were associated with significantly regulating a promoter. We observed that the bins with the highest activity had nearly a twofold increase in the proportion of validated enhancers relative to bins with low activity (Supplementary Fig. 4a), suggesting that our MPRA datasets could be used to predict sequences with larger CRISPRi effects. Considering our activity scores alongside activity-by-contact (ABC) scores[25], we observed that our scores only subtly improved performance in the task of discriminating significant from non-significant enhancer–promoter interactions in two of these three studies (Supplementary Fig. 4b). This small improvement in performance may be partially explained by the observation that the H3K27ac signal, which is already considered in the ABC model, has the advantage of integrating local and distal regulatory information, which may overshadow the consideration of an element's local activity alone.

## Promoter properties and orientation effect

Following procedures established in previous studies[14], we utilized our substantially expanded MPRA data to examine the relative orientation

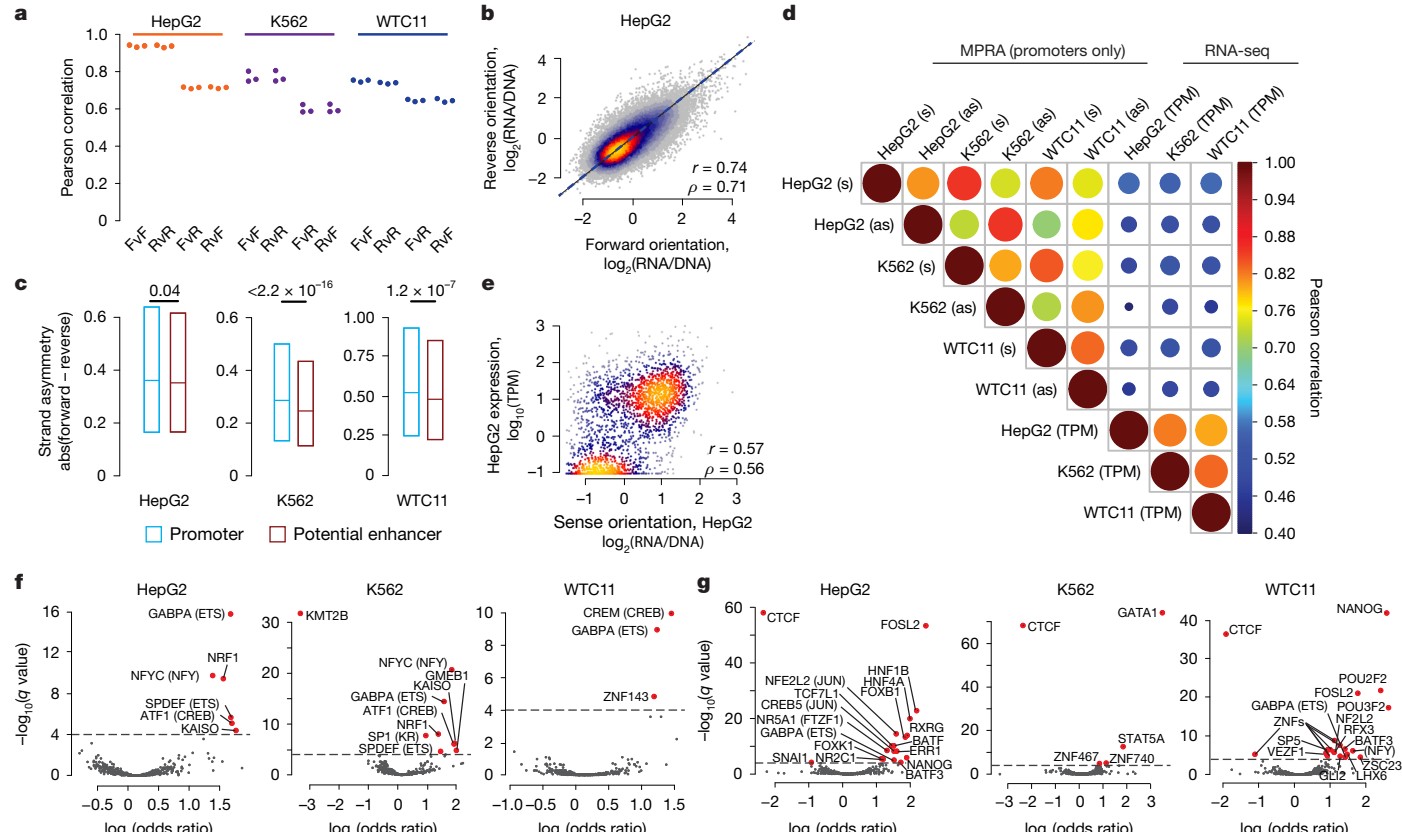

**Fig. 2 | Properties and orientation dependence of promoters and potential enhancers. a**, Beeswarm plot of the Pearson correlation values corresponding to each of the three pairwise comparisons among the three replicates. The correlations are computed between observed cCRE activity values for elements positioned either in the same (forward versus forward (FvF) and reverse versus reverse (RvR)) or opposite (forward versus reverse (FvR) and reverse versus forward (RvF)) orientations. **b**, Scatter plot of the average activity score for each cCRE in the forward versus reverse orientation. Regions are coloured according to the density of data from light blue (low) to yellow (high). Pearson ($r$) and Spearman ($\rho$) correlation values are shown. **c**, Box plots showing the distribution of strand asymmetries for promoters and potential enhancers for each cell type. The centre line is the median residual value and box edges delineate 25th and 75th percentiles, evaluated with a one-sided Wilcoxon rank-sum test. **d**, The heat map indicates the correlation between the sense (s) and antisense (as) orientations of promoters and endogenous gene expression levels measured in transcripts per million (TPM) using RNA sequencing. The sizes of the circles are proportional to the Pearson correlations. **e**, Scatter plot of activity scores for sense-oriented promoters and endogenous gene expression levels for HepG2 cells. **f,g**, Volcano plots indicating the enrichment of HOCOMOCO v.12[30] transcription factor families in the top 1,000 versus bottom 1,000 promoters (**f**) and potential enhancers (**g**), as ranked by MPRA activity. Enrichment (measured as an odds ratio) and Benjamini–Hochberg corrected $q$ values computed using Fisher's exact test. Significant families above the $P$ value acceptance threshold (dashed horizontal line) are labelled with a representative transcription factor family member with the general transcription factor family in parentheses.

dependence of promoters and enhancers. We quantified the degree to which cCREs exhibited observable orientation dependence. In all cell types examined, cCREs cloned in the same orientation exhibited approximately 0.2 greater correlation among replicate pairs than cCREs cloned in the opposite orientation with respect to the reporter (Fig. 2a). Averaging among replicates in HepG2 (the cell type with the highest technical reproducibility among replicates), we detected a substantial number of cCREs that exhibited greater activity in one orientation relative to the other (Fig. 2b). These findings suggest that the activities of cCREs are largely, but not entirely, independent of orientation.

To further compare the properties of strand asymmetry between promoters and enhancers, we analysed strand asymmetry distributions, defined as the absolute deviation between activity scores from one orientation to the other. Consistent with previous studies[27,28], we observed that promoters displayed slightly stronger strand asymmetry effects relative to potential enhancers in all cell types examined (Fig. 2c), supporting the conclusion that they can contain transcription factor binding sites (TFBSs) that promote transcription unidirectionally (or at least more unidirectionally than potential enhancers).

Given the slightly enhanced orientation dependence of promoters, we sought to evaluate the relationship between orientation-specific promoter activity as measured by lentiMPRA and RNA sequencing of endogenous genes. Across all pairs of cell types, MPRA measurements from the same orientation displayed greater correlation than measurements from the opposite orientation (Fig. 2d and Extended Data Fig. 3d). Furthermore, when comparing against endogenous expression levels, we observed that: (1) MPRA measurements from the matched cell type displayed nearly the same correlations as those from a different cell type; and (2) for each cell type, MPRA measurements from the sense orientation displayed greater correlation to endogenous gene expression levels than promoters tested in the antisense orientation (Fig. 2d and Extended Data Fig. 3d). To further evaluate whether our promoter measurements explained cell-type-specific gene expression, we computed the deviations of promoter activity from their mean activity across cell types, as well as the corresponding deviations for endogenous gene expression levels across the same cell types. All cell types exhibited a Pearson correlation of less than or equal to 0.11 between the two sets of measurements, supporting our previous conclusions that enhancers, super-enhancers and polycomb targeting are more likely to explain deviations of endogenous transcriptional activity from core promoter activity[29]. Collectively, these results suggest that core promoters possess weak

orientation dependence and when tested individually have little cell-type specificity.

Although they correspond to a very short region of the promoter, the 200-bp core of promoters centred at the TSS strongly recapitulated endogenous gene expression levels (Pearson $r \approx 0.55$; Fig. 2e and Extended Data Fig. 3a,b). Owing to the switch-like (that is, on/off) state of promoters, expression values fell into a bimodal distribution, which slightly inflated these correlations. Removing all non-expressed genes led to a reduction in the correlation between MPRA measurements of promoter activity and endogenous expression levels (Pearson $r \approx 0.43$; Extended Data Fig. 3c–g). Notably, in WTC11 cells, we found a larger cohort of transcriptionally active genes whose promoters were inactive in our MPRA (Extended Data Fig. 3b,g). Additional analysis revealed that this observation could largely be explained by the use of alternative promoters in WTC11 cells, as the cap analysis of gene expression (CAGE-seq) signals in the precise promoters tested were congruent to MPRA activity in a similar manner among all three cell types (Pearson $r \approx 0.60$; Extended Data Fig. 3h–j). We performed two complementary analyses to gain further insight into which transcription factor families might bind to promoters exhibiting high expression versus those promoting low expression and potential enhancers: (1) an enrichment analysis using motifs annotated by HOCOMOCO v.12[30] (Fig. 2f,g, Supplementary Methods and Supplementary Table 5); and (2) de novo motif discovery (Extended Data Fig. 4). Together, these analyses primarily identified CpG-rich motifs as well as TFBSs for the ETV/ETS-related, KLF-related, NFYA/B/C and THAP11 transcription factors as being associated with active promoters (Fig. 2f and Extended Data Fig. 4a). These motifs were in many cases different from those detected in high- versus low-activating potential enhancers, for which factors such as HNF1B, HNF4A, GATA1/2/6 and POU5F1–SOX2 emerged as cell-type-specific activating factors and CTCF emerged as a general repressive factor (Fig. 2g and Extended Data Fig. 4b). Although we did not anticipate such short 200-bp promoter fragments to reflect endogenous expression levels, collectively, these findings are consistent with previous models that showed that CpG-rich promoters are associated with increased gene expression; and that core promoters centred at the TSS possess weak cell-type specificity, are information dense and strongly predict gene expression levels[29].

## Sequence-based models predict activity

We next set out to train regression models to predict regulatory activity. We began with a biochemical model (Supplementary Results) that used a compilation of thousands of biochemical features extracted from the three cell types (Supplementary Table 6). This model was able to predict enhancer activities with high accuracy (Pearson $r \approx 0.72$) in all three cell types (Supplementary Fig. 5a) using a tenfold cross-validation approach on our data. Many biochemical features were strongly associated with element activity (Supplementary Fig. 5b–e). The variable feature count associated with each cell type led to the possibility of biasing performance. However, training models that considered a 'universal' feature set, merging features from all cell types, only weakly improved performance (Supplementary Fig. 5f).

Sequence-based deep learning models[31] have demonstrated strong performance relative to biochemical models[32] and have been used to predict MPRA data[33,34]. We benchmarked the performance of four sequence-based models, trained on our MPRA data for each of the three cell types: (1) MPRAnn, a standard convolutional neural network (CNN) (Supplementary Fig. 6); (2) MPRALegNet, a CNN based on the LegNet architecture[35], which uses EfficientNetV2-like convolutional blocks (Fig. 3a); (3) EnformerMPRA, which uses the CNN-transformer architecture Enformer[36] to generate a set of 5,313 predicted biochemical features and then fits a lasso regression to the MPRA data using these features; and (4) SeiMPRA, which fits a similar lasso model considering 21,907 biochemical features predicted by Sei[37]. Both MPRAnn

and MPRALegNet underwent optimization procedures to detect hyperparameter and data augmentation settings that improve model performance (Extended Data Fig. 5a,b, Supplementary Methods and Supplementary Table 1). Comparing the performance to our biochemical lasso regression model on the identical ten folds of held-out data, we observed that all sequence-based models outperformed the biochemical model, with MPRALegNet achieving the best performance in two of the three cell types (Fig. 3b and Supplementary Table 8). Although we include EnformerMPRA and SeiMPRA for comparison purposes, we caution that they may have inflated performance because: (1) they have a more than eightfold larger feature set than the biochemical models and were trained on additional cell types and biochemical marks; and (2) having been trained on nearly the entire genome, they had the opportunity to observe biochemical marks associated with elements in the test set. Moreover, sequence-based models probably performed better than biochemical models because they have access to the precise 200-bp sequence being tested, whereas biochemical signals lack this degree of spatial resolution. Combining the folds of data, our best model, MPRALegNet, achieved a performance (Pearson $r \approx 0.83$; Fig. 3c) that was comparable to the technical noise of the assay itself (that is, the replication of replicates; Supplementary Fig. 2c–e). Nevertheless, downsampling analysis indicated that the model lies in a regime that could still benefit from additional training data, given the log-linear improvement in performance as a function of training set size for each cell type (Extended Data Fig. 5c).

Given the favourable performance of MPRALegNet, we sought to examine the principles it had learned. We performed in silico mutagenesis (ISM) on the full set of MPRA data and then used TF-MoDISco-lite[38] to identify motifs at variants with a large predicted effect size. This strategy identified many housekeeping factors that are predicted to activate transcription in all cell types, including NRF1, USF1/2, TFEB and TFE3, JUN and FOS-related, KLF-related (KLF/SP), C/EBP-related and ETS-related transcription factor families; additionally, we discovered a motif for REST, a known transcriptional repressor[39] (Extended Data Fig. 6). Of note, CTCF was associated with both transcriptional activation and repression, suggesting that it may impart differential responses depending on sequence context. The top three TFBSs most frequently associated with transcriptional activation among all cell types were KLF-related, ETS-related and CTCF motifs; by contrast, the top cell-type-specific TFBS were HNF4A/G in HepG2 cells, GATA–TAL1 dimer in K562 cells and POU5F1–SOX2 composite element in WTC11 cells (Fig. 3d). Overall, many motifs, including several unknown TFBSs, were discovered that were overlooked by a classic motif-enrichment analysis (Extended Data Fig. 4), supporting the complementarity of the TF-MoDISco approach. To validate the implicated transcription factors, we examined an MPRA dataset in which elements were tested in the context of transcription factor knockdown via CRISPR inhibition in K562 cells[40]. We were able to validate GATA1, NRF1, SP1 and FOSL1 as regulatory factors, with insufficient evidence for STAT1 and ATF4, potentially owing to limited knockdown efficiency or compensatory effects conferred by other transcription factor family members (Extended Data Fig. 7a).

## MPRALegNet predicts TFBS combinations

To gain insight into the nature of combinatorial TFBS effects learned by MPRALegNet, we examined the top ten most abundant activating TFBS motifs detected in each cell type. First, we tested the effect of the number of copies of homotypic (same) TFBSs on reporter expression. In each cell type, MPRALegNet could accurately predict the activation profile for elements containing between one and five sites of the indicated TFBS (Fig. 3e and Extended Data Fig. 7b–e). In most cases, transcription factors displayed close to a multiplicative (that is, log-additive) pattern with respect to TFBS dosage[41] (Extended Data Fig. 7c). However, several transcription factor families, such as STAT (K562) and ETS-related

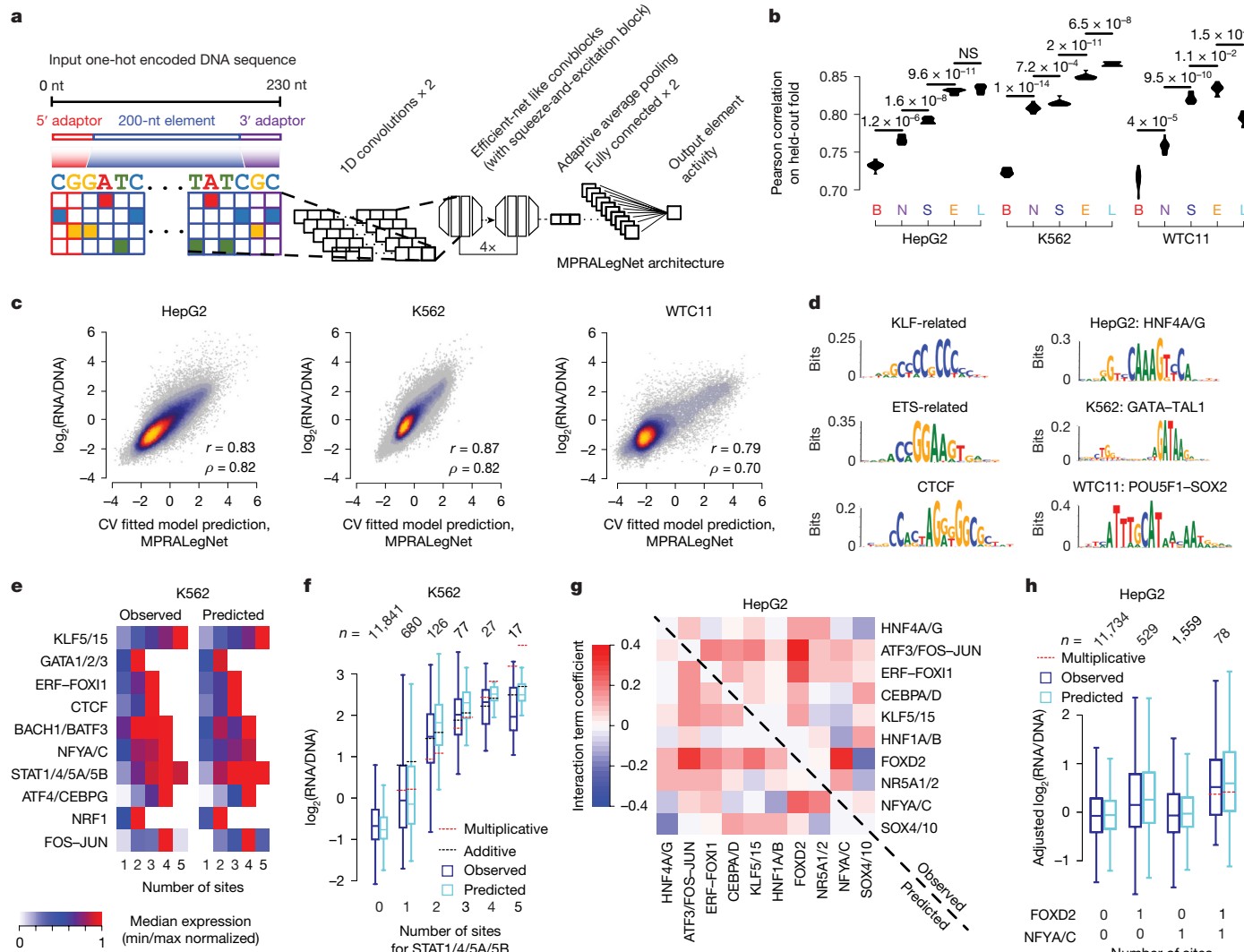

**Fig. 3 | Sequence-based models predict regulatory element activity.**
**a**, MPRALegNet is a deep CNN trained to predict cCRE activity from an input sequence of the tested element. **b**, Violin plots showing the performances of sequence-based and biochemical models on ten cross-validation (CV) folds, with improvement relative to another model evaluated using a one-sided, paired *t*-test. B, biochemical; E, EnformerMPRA; L, MPRALegNet; N, MPRAnn; S, SeiMPRA; NS, not significant. **c**, Scatter plots indicating relationship between MPRALegNet predictions and observed element activity scores for each cell type. **d**, Set of enriched motifs discovered by TF-MoDISco-lite[38]. Left, top three motifs detected across multiple cell types. Right, top motif detected for each cell type. **e**, Heat map indicating the relationship between homotypic TFBS dosage ($n = 1$ to 5 TFBSs) and the observed MPRALegNet-predicted response in K562 cells. **f**, Box plots showing the full dosage-dependent distributions for the STAT1/4/5A/5B transcription factor family, along with the expected effect

in the scenario of either a multiplicative or additive model. The number of elements represented in each group is indicated above the plot. **g**, Heat map indicating interaction term coefficients reflecting super-multiplicative (red) and sub-multiplicative effects (blue), for elements possessing the indicated pair of heterotypic TFBSs in HepG2 cells. Coefficients fit to observed and predicted values are shown in the top right and bottom left halves of the heat map, respectively. **h**, Relationship of the distributions of observed and MPRALegNet-predicted activity scores in HepG2 cells for the subset of elements possessing either zero or one TFBSs corresponding to the NFYA/C or FOXD2 families, or one TFBS for both transcription factor families, adjusted for potential confounding effects induced by the presence of other transcription factors (Methods). The red horizontal line is the expected activity score under a multiplicative model. Box centre lines and edges as in Fig. 2. Whiskers show the most extreme data point up to a maximum of 1.5× the interquartile range.

(WTC11), displayed sub-multiplicative patterns at the highest dosages (Fig. 3f and Extended Data Fig. 7e), indicative of a saturating expression effect. Super-multiplicative (that is, cooperative) effects were also observed for certain dosages, such as the increase observed from one to two sites for the STAT transcription factor family (Fig. 3f).

Next, we evaluated deviations from multiplicative effects for heterotypic (different) TFBS pairs, as quantified by an interaction term when considering the subset of elements with: (1) a single site to either of the two transcription factors; or (2) co-occurring instances of both TFBS[42]. Adjusting for possible confounding effects induced by the presence of other transcription factors (Methods), we observed both super-multiplicative and sub-multiplicative effects for different TFBS

pairs in HepG2 cells, as indicated by positive or negative interaction term coefficients, respectively (Fig. 3g). The magnitude of these terms was strongly correlated between the predictions and observations ($r = 0.92$), suggesting that MPRALegNet learned more complex combinatorial properties among TFBS pairs (Extended Data Fig. 7f). For example, the co-occurrence of ATF3/FOS–JUN and FOXD2 sites led to the strongest super-multiplicative effect (Fig. 3h); conversely, the co-occurrence of HNF4A/G and NFYA/C sites led to a sub-multiplicative effect (Extended Data Fig. 7g). Similar findings were observed in both K562 and WTC11 cells (Extended Data Fig. 7h–l). Collectively, MPRALegNet was able to model nonlinear interdependencies between TFBS combinations in all cell types.

## Predicting fine-mapping and variant effects

We next examined MPRALegNet utility for genetic fine-mapping and variant effect prediction. We examined all single nucleotide polymorphisms (SNPs) in linkage disequilibrium (LD) with lead SNPs derived from the GWAS catalogue, initially intersecting the seven tiled disease loci in our study (Extended Data Figs. 1d,2 and Supplementary Fig. 3). For every SNP in LD, we predicted the difference between the reference and alternative allele using our K562 MPRALegNet model. We found several instances in which the predicted effect size was exceptionally large. For example, the model predicted both gain-of-function and loss-of-function (LOF) SNPs around *RBM38* (rs2426715, rs376911010 and rs737092; Extended Data Fig. 8a) and a LOF of a potential enhancer within an active lentiMPRA sequence in the intron of *LMO2* (rs75395676; Extended Data Fig. 8b).

To further evaluate this fine-mapping prediction strategy, we benchmarked model predictions against two complementary tasks. First, we verified performance on variant effect data using six sets of allele-specific variants (ASVs) found in chromatin accessibility (assay for transposase-accessible chromatin with sequencing (ATAC-seq) and DNase-seq) and transcription factor binding data (chromatin immuno-precipitation with sequencing (ChIP–seq)) for HepG2 and K562 cells available in the UDACHA[43] and ADASTRA databases[44]. The significant ASVs provide information on variant effects, including the preferential transcription factor binding or chromatin accessibility as allelic imbalance towards the reference or alternative allele. For all six tested combinations of ASV sources and cell types, we observed significant associations between the observed and predicted scores both before and after excluding cases in which the ASV was non-significant or model predictions were too uncertain (Fisher's exact test odds ratios > 1.5 and $P < 0.05$; Fig. 4a and Supplementary Table 9). We conclude MPRALegNet successfully recognizes allele-specific regulatory SNP effects in matched cell types.

Next, we sought to further validate the accuracy of our variant effect predictions by generating ISM scores for promoters (*F9*, *LDLR* and *PKLR*) and an enhancer (*SORT1*) for which we previously performed MPRA saturation mutagenesis in HepG2 (*F9*, *LDLR* and *SORT1*) or K562 (*PKLR*) cells[45]. Comparing MPRALegNet predictions for the *PKLR* promoter to MPRA data revealed that most of the relevant TFBSs (GATA3, KLF9, SP5 and NFIB) could be detected, although the predicted effect sizes were relatively smaller for KLF4 and GATA2 (Fig. 4b). Collectively, we observed a correlation of 0.49 for *SORT1*, 0.65 for *PKLR*, 0.66 for *LDLR* and 0.51 for *F9* between model predictions and observed data (Extended Data Fig. 9), confirming that MPRALegNet, despite being trained on cCRE activity, could partially model the regulatory effects of individual genetic variants. These results were comparable to those from Enformer[36] (0.63 for *SORT1*, 0.83 for *PKLR*, 0.62 for *LDLR* and 0.28 for *F9*). Combined, our results show how our models can be used for the prediction of regulatory variant effects.

## Characterization of cell-specific factors

Although our large-scale MPRAs focused on element activity within each cell type, they did not directly evaluate the cell-type-specific activity of each element. We therefore designed a lentiMPRA library to test a common set of elements in all three cell types. This library consisted of around 19,000 potential enhancers from each of the three cell lines, sampled uniformly from previous large-scale MPRA experiments to span a wide range of activity; a subset of promoters that exhibit high expression variance as well as a wide range of average expression among cell types from our previous large-scale MPRA experiments; dinucleotide shuffled controls; and a set of positive and negative controls using synthetic elements previously tested in HepG2 cells[19], or natural elements with evidence to exhibit K562-specific activity[18] (Fig. 5a, Supplementary Table 10 and Methods). Elements were largely tested in a single orientation (sense orientation for promoters and forward orientation for potential enhancers).

We observed 10–70 median barcodes per element in each replicate among all cell types (Supplementary Fig. 7a). Element activity scores were strongly concordant across replicate pairs (Pearson correlations of 0.98 (HepG2), 0.97 (K562) and 0.96 (WTC11); Supplementary Table 11 and Supplementary Fig. 7b–d). Averaging across the three replicates, we observed strong agreement among element activity scores between cCREs common to both our joint and large-scale libraries (Pearson $r = 0.90$ (HepG2), $r = 0.88$ (K562), $r = 0.83$ (WTC11); Supplementary Fig. 7e). We observed the distribution of element activity scores to be strongly divergent and weakly cell-type-specific between positive and negative controls in each cell type (Supplementary Fig. 7f). Although promoters and potential enhancers displayed significant activity in all cell types, the distribution of activities for potential enhancers derived from the matched cell type was greater than those from unmatched cell types (Supplementary Fig. 7f).

To further examine cell-type specificity, we evaluated the behaviour of each element category in each pair of cell types. Promoters exhibited the strongest correlation among cell-type pairs (mean Pearson $r = 0.82$); by contrast, potential enhancers displayed weaker correlations when comparing the activity scores from the cell type from which the enhancer was derived to those from a different cell type (Pearson $r = 0.32–0.51$ (HepG2), $r = 0.51–0.65$ (K562) and $r = 0.64–0.65$ (WTC11); Supplementary Fig. 8). Next, we evaluated the relationship between DNase accessibility relative to MPRA activity in all three cell lines. We observed that strong DNase accessibility signals were not a prerequisite for MPRA activity. For instance, we found that sequences with nearly absent DNase signal in K562 cells, but high DNase signal in both HepG2 and WTC11 cells, could still lead to high MPRA activity in K562 cells (Fig. 5b and Supplementary Fig. 9a). Further reinforcing this observation, we observed that 15–25% of elements that lack DNase signal were more active than shuffled negative controls (5% FDR), although increased DNase signal was clearly associated with an increased proportion of active elements (Supplementary Fig. 9b). Collectively, our results show that promoters are less cell-type-specific, whereas potential enhancers show stronger cell-type specificity, in line with their presumed cell-type-specific functions[46].

We next sought to interrogate the cell-type-specific activity of each element. We performed a principal components analysis using our matrix of element activity scores in three cell types and removed the dominant principal component, which represented the 'universal' signal of element activity among cell types. An analysis of principal components 2 and 3 (PC2 and PC3) revealed that promoters have a slight bias towards expression in WTC11 cells, and both controls and potential enhancers have a stronger bias towards greater expression in the cell type from which they were derived (Fig. 5c). We computed an element specificity score, which measures the deviation of each element from its mean activity across cell types. These scores recapitulated the expected patterns of enrichment or depletion of element activity for different element categories, with HepG2 and K562 controls showing strong relative activity in their respective cell types; potential enhancers showing strong relative activity in their respective cell types; and promoters and negative controls showing weakly stronger activity in WTC11 cells relative to others (Fig. 5d and Supplementary Fig. 10). A possible explanation for the stronger activity of negative controls in WTC11 could be that stem cells tend to exhibit a more globally open chromatin state[47], making them susceptible to greater levels of background transcription relative to other cell types.

We next benchmarked the performance of biochemical and sequence-based models in predicting our lentiMPRA element specificity scores. Consistent with previous results, a multi-task version of MPRALegNet outperformed the biochemical model and EnformerMPRA outperformed both MPRALegNet and the biochemical model for each of the three cell types (Pearson $r \approx 0.81$ for EnformerMPRA; Fig. 5e

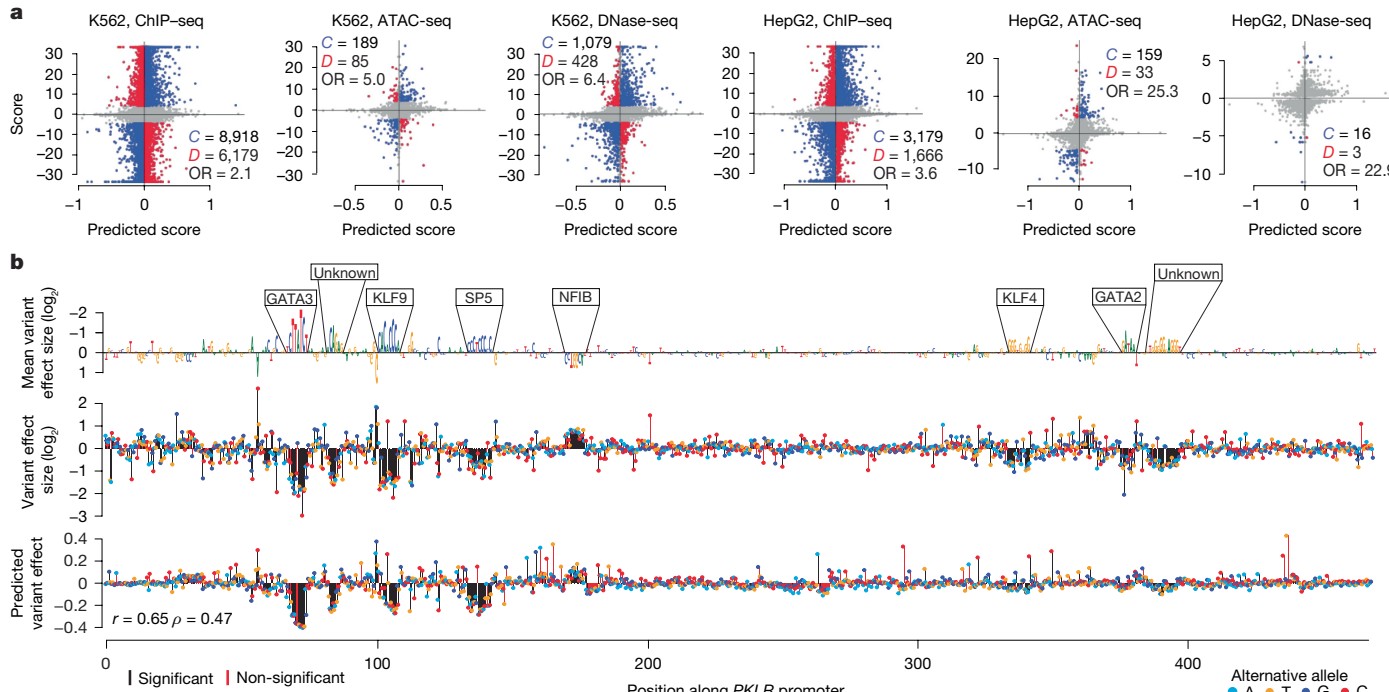

**Fig. 4 | MPRALegNet variant effect prediction. a**, Scatter plots of predicted variant effects and observed allele-specific differences detected in ChIP–seq, ATAC-seq and DNase-seq data in K562 and HepG2 cells. Indicated are the number of cases in which the predictions and observations are concordant (*C*; blue), discordant (*D*; red), or not considered (grey) because either the ASV FDR > 0.05 or the model predictions are too uncertain (*P* value > 0.05; Supplementary Methods). The corresponding odds ratio (OR) is also indicated (Fisher's exact test using the 2 × 2 contingency tables provided in Supplementary Table 9). **b**, Saturation mutagenesis data from the *PKLR* enhancer[45]. Top, the reference sequence scaled to the mean effect size among all alternative mutations, annotated by 6 out of 8 significant TFBSs that match known motifs[54]. Middle, measured effect sizes of individual variants. Bottom, MPRALegNet predictions with corresponding Pearson (*r*) and Spearman (*ρ*) correlation values.

and Supplementary Table 12). Using TF-MoDISco-lite[38], we identified cell-type-specific motifs learned by MPRALegNet, detecting 21 motifs to be associated with cell-type-specific activity in HepG2 and K562 cells and 12 motifs in WTC11 cells (Extended Data Fig. 10). Individual transcription factors linked to the top three ranked cell-type-specific binding motifs from each cell type also exhibited strong cell-type-specific expression in the expected cell types; additionally, CTCF showed weakly enriched expression in WTC11 (Supplementary Fig. 11). It is important to note that the transcription factors tested here do not represent a comprehensive set of transcription factor family members that recognize the same motif, and that other untested family members might further explain the cell-type specificity of the observed motif.

## Discussion

Large-scale MPRA datasets are available for other cell lines[6–9]. However, they are primarily tested via episomal STARR-seq, require a very large number of cells, provide an episomal readout and tend to use a strong promoter to increase the ability to detect activity[7]. By contrast, our modified lentiMPRA provides large functional datasets with an 'in genome' readout. The ability to systematically test thousands of cCREs in an unbiased manner for a given cell type allowed us to identify predictive biochemical and sequence-based features for each cell type with high confidence. However, although the number of tested sequences was high, many additional cCREs may have been omitted in our annotations and subsequently in our assays, owing to our selection criteria, additional cCRE annotation assays, marks or tools that were not used, technical issues of the biochemical assays used to select sequences and other factors.

We tested all 19,104 known promoters of protein-coding genes in both orientations in HepG2 and K562 cells and 7,500 in WTC11 cells. In addition to observing promoter activity strand-orientation bias in line with

previous studies[27,28], we extensively characterized the sequence-based information needed to generate these on/off switches. We found that 200-bp blocks centred at the TSS have sufficient sequence data to provide this switch and are sufficient to drive expression in a similar manner to their associated gene. Sequencing of these active core promoters shows that they are enriched for CpG-rich motifs that are known to have ubiquitous function[29] (Extended Data Fig. 4a). They also include the KLF-related transcription factor family that are known to interact with the transcription initiation complex and additional transcription factors that provide ubiquitous promoter activity[48] and the ETS-related family which is enriched in ubiquitously expressed promoters[49]. We also observed an enrichment for the NF-Y family that is known to interact with the CCAAT box and TATA-less eukaryotic promoters[50]. Of note, our lentiMPRA design tested promoters together with a minimal promoter that is 32 bp long, which could affect promoter activity. However, this approach enabled us to test hundreds of thousands of enhancers and thousands of promoters and compare them in the same assay. Our results were similar to previous reports[27,28], showing orientation biases for promoters and motif enrichment that is known to provide ubiquitous promoter expression, supporting the idea that the addition of this 32-bp sequence to our assayed promoters probably did not affect our findings.

In line with previous work[10,14,51], we show that sequence-based models provide superior ability to predict functional sequences from MPRA. MPRALegNet enabled us to tease out many motifs that are important for these predictions, both universal and cell-type specific, and model their combinatorial effects. Of note, one of the main enriched transcription factor motifs among the promoters and enhancers in all three cell types are the stripe KLF-related transcription factors. Stripe transcription factors are thought to provide co-accessibility and increase residence time for other transcription-associated factors in promoters and enhancers[52] and were also found to be enriched in active regulatory elements in a

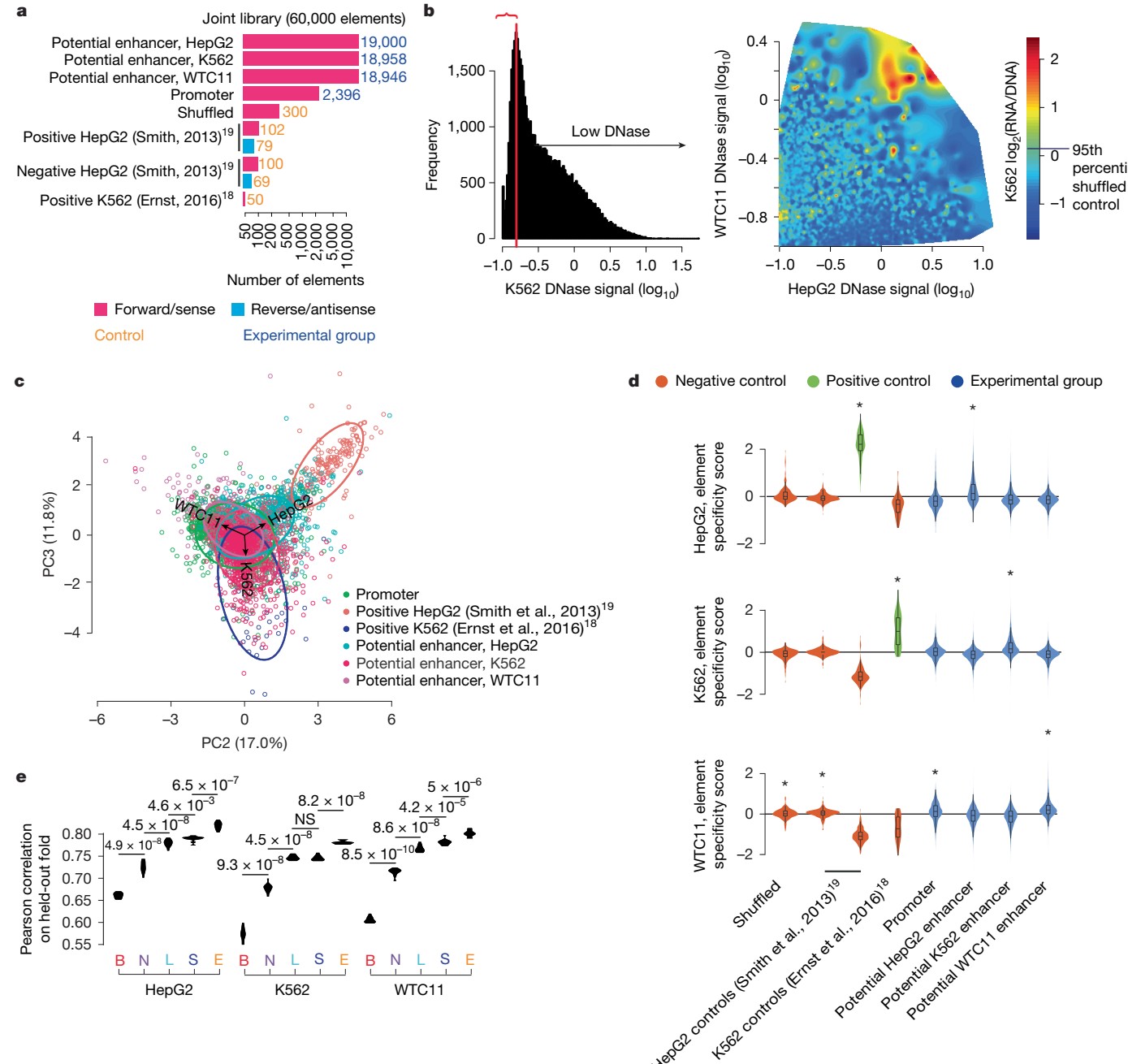

**Fig. 5 | Assessment of cCRE cell-type-specific activity. a**, Composition of the joint library tested in all three cells. **b**, Detection of active elements in K562 cells lacking DNase signal. We evaluated the DNase signal for the subset of elements with low K562 DNase accessibility (left of vertical red line, indicated by red bracket) in the other two cell types and then quantified a smoothed kernel density estimate of MPRA activity in K562 cells. **c**, Principal components analysis biplot indicating the second (PC2) and third (PC3) principal components, with a random sample of up to 1,000 data points from each element category plotted. The loading vectors (corresponding to each cell type) as well as ellipses fitting the regions of highest density for each element category are also shown. **d**, Violin plots showing the distribution of element specificity scores for each element category, alongside information about which distributions show a median significantly greater than zero (one-sided Wilcoxon signed-rank test, *$P < 0.05$). Inner boxplots are plotted as described for Fig. 3f,h. **e**, Performance of trained MPRAnn, MPRALegNet, SeiMPRA and EnformerMPRA models on each of ten cross-validation folds of held-out data, relative to the corresponding performance of lasso regression models trained on biochemical features, with improvement relative to another model evaluated with a one-sided, paired *t*-test.

recent large-scale lentiMPRA[11], in line with their generalizable function. Although MPRALegNet was trained on three cell types, the similar performance of cell-type-agnostic models and cell-type-specific models in the variant effect prediction task[36] and observation of similar measured effect sizes of the same variants in multiple cell types[45] support its use in additional cell types. Although MPRALegNet only performs competitively with Enformer, its roughly 200-fold reduction of parametric complexity from Enformer's approximately 249 million to around

1.3 million parameters provide a computationally efficient and practical way to rapidly predict variant effects on a genome-wide scale. We discuss several limitations of this work in Supplementary Discussion.

In summary, our work provides a large catalogue of functional regulatory elements in three established cell lines accompanied by machine learning-based tools that provide a valuable resource for prediction of regulatory activity. We provide systematic support for the following generalizations about mammalian regulatory elements: (1) enhancer

activity is largely independent of orientation, in line with the original definition of enhancers[53]; (2) enhancers have more inherent cell-type specificity than promoters; and (3) cell-type specificity is driven by a small number of cell-type-specific TFBSs. These datasets will also improve our understanding of the regulatory code, variant effects and regulatory element design for therapeutic delivery.

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

## Methods

### Design of Agilent oligonucleotide library

**HepG2 pilot library.** For the HepG2 pilot library, we collected two replicates of DNase I hypersensitivity data derived from HepG2 cells (ENCODE narrowPeak BED files: ENCFF505SRS and ENCFF268DTI, hg19 human genome build)[55]. For each replicate, we collapsed overlapping peaks using bedtools merge (parameters "-o collapse -c 2,3,7"). Then, we identified peaks that intersected between the two replicates, merged these peaks, and removed the subset of merged peaks that overlap promoters (defined as regions ±2,500 nt around any annotated TSS). The resulting distribution of peak sizes was such that 97% of peaks were ≤200 bp in length. We therefore centred the designed oligonucleotides at each merged DNase peak, consistent with the known region of maximal regulatory activity[18], and added ±100 bp to either side. This procedure resulted in a set of 66,017 cCREs. For this pilot library, we sought to evaluate cCREs which overlapped a wide range of putative transcription factor binding sites. We therefore intersected these potential enhancers with wgEncodeRegTfbsClusteredWithCellsV3.bed.gz[56] in order to count the number of putative HepG2 transcription factor binding sites intersecting these cCREs. We uniformly and randomly sampled ~1,834 cCREs with 0-1, 1–5, 5–10, 10–20, and >20 TFBSs for a total of 9,172 elements. Including 50 positive and 50 negative controls from each of two previous studies[10,19] resulted in a total of 9,372 elements. These 171-bp controls from previous work were linked downstream of a 29-nt random sequence GGTGCTCGATTTAATTTCG CCGACGTGAT to match the 200-bp sequence length of cCREs. For the final oligonucleotide library, each element was linked to the 5′ adaptor AGGACCGGATCAACT and 3′ adaptor CATTGCGTGAACCGA, designing two 230-bp oligonucleotides per element to minimize the impact of oligonucleotide synthesis errors.

**K562 pilot library.** An analogous procedure was followed for the K562 pilot library as in 'HepG2 pilot library', with the following modifications: (1) ENCODE narrowPeak BED files ENCFF027HKR and ENCFF154JCN (hg38 human genome build)[55] were used; (2) merging these peaks resulted in 34,367 potential enhancers; (3) after intersecting the peaks with K562 transcription factor binding sites, we sampled ~1,278 enhancers from each transcription factor binding site bin to test a total of 6,394 cCREs; (4) 250 additional negative controls were chosen by dinucleotide shuffling 250 random potential enhancers possessing 1–5 TFBSs; (5) positive and negative controls were chosen from CRISPRi screens[20,21], a previous MPRA[18], and select loci of interest such as α-globin and β-globin; (6) a total of 7,500 elements were tested; and (7) controls were already 200 bp in length, requiring no addition of a random sequence.

**HepG2 large-scale library.** Following the procedures outlined in 'HepG2 pilot library', we tested all 66,017 previously identified cCREs in both orientations. For human protein-coding gene promoters, we extracted the average signal across cell types in TPM for each CAGE peak listed in hg19.cage_peak_phase1and2combined_tpm_ann.osc.txt.gz from the FANTOM5 consortium[57,58]. The first exons of all protein-coding gene transcripts were collected from Ensembl v.83 (hg38 genome build)[59], transformed into hg19 coordinates using liftOver[60], and then intersected with the CAGE peaks to identify a single promoter per gene corresponding to the promoter with the maximal average TPM. To select the final 200-bp oligonucleotide for testing, we identified the centre of the promoter DNase peak on the basis of the HepG2 DNase peaks merged across replicates (described in 'HepG2 pilot library'). In the scenario in which no DNase peak overlapped the promoter, we centred on the midpoint of the CAGE peak. In the scenario in which neither a DNase nor CAGE peak existed, we centred on the TSS defined by the Ensembl annotation. This resulted in a total of 19,104 protein-coding gene promoters, of which 6,181 were centred on a DNase peak, 9,735 were centred on a CAGE peak and 3,188 were centred on a Ensembl TSS definition. The oligonucleotide tested included the ±100-bp window around this central position in the sense orientation with respect to the gene. A random subset of 12,411 promoters were also tested in the antisense orientation. We tested 102 positive and 102 negative controls from a previous study[19] as well as 175 dinucleotide shuffled negative controls in both orientations. These shuffled controls were derived from shuffling a random subset of 175 DNase peaks. This resulted in a library consisting of 164,307 elements, for which we ordered one 230-bp oligonucleotide per element.

**K562 large-scale library.** To acquire a set of DNase peaks for testing, we used the 'optimal peak' calls derived from processing ENCODE experiment ID: ENCSR000EOY through the ENCODE DCC Irreproducible Discovery Rate (IDR) pipeline, available at https://github.com/ENCODE-DCC/atac-seq-pipeline (generously provided by A. Kundaje). Removing DNase peaks overlapping human promoters resulted in 87,618 potential enhancers tested in both orientations. The promoters tested were identical to those described in 'HepG2 large-scale library', except that it included all 19,104 promoters tested in both orientations. We tested 50 positive and 200 negative controls from a previous MPRA study[18] as well as the same 250 dinucleotide shuffled negative controls as tested in 'K562 pilot library'. Finally, 14,918 tiles not overlapping DNase peaks, and subsampled from the ±1 Mb region around the following 7 genetic loci: *GATA1*, *MYC*, *HBE1*, *LMO2*, *RBM38*, *HBA2*, and *BCL11A*, were chosen using our representative subset selection approach (described in the 'Representative subset selection' section below) and tested in both orientations. This resulted in a library consisting of 243,780 elements, for which we ordered one 230-bp oligonucleotide per element.

**WTC11 large-scale library.** To acquire a set of DNase peaks for testing, we used the peak calls derived from applying the hotspot2 pipeline (https://github.com/Altius/hotspot2) at FDR = 0.05 to ENCODE experiment ID: ENCSR785ZUI (generously provided by R.Sandstrom)[61]. This resulted in two independent replicates, which were merged into a unified set using the procedures described in 'HepG2 pilot library'. Removing DNase peaks overlapping human promoters resulted in 83,201 potential enhancers. Together with the 19,104 promoters described in 'HepG2 large-scale library', these elements were subsampled to select 30,121 potential enhancers and 7,500 promoters using our representative subset selection approach described below, and tested in both orientations. We also tested 100 positive and 100 negative controls from a previous study[24] as well as 100 dinucleotide shuffled negative controls, which were derived from shuffling 100 random sequences from our set of 30,121 potential enhancers. This resulted in a library consisting of 75,542 cCREs, for which we ordered one 230-bp oligonucleotide per element.

**Joint library tested in HepG2, K562, and WTC11 cells.** Given the measured potential enhancers from the forward orientations in each of the HepG2, K562 and WTC11 large-scale libraries, we binned each set of cCREs into ten equally sized bins spanning the range of measured $\log_2$(RNA/DNA) values in the selected cell type. We randomly sampled an approximately equal number from each bin, resulting in 19,000 HepG2, 18,958 K562 and 18,946 WTC11 potential enhancers. A similar procedure was followed with sense-oriented promoters, except that the ten bins were established on the basis of the mean $\log_2$(RNA/DNA) across all three cell types (that is, instead of performing the procedure independently in each cell type as before), and the top 1,000 promoters exhibiting the greatest variance across three cell lines were also selected for testing. This resulted in the selection of 2,396 out of 19,104 promoters. We also tested 181 positive and 169 negative HepG2 controls from a previous study[19], 50 positive K562 controls from a previous study[18],

and 300 dinucleotide shuffled negative controls. The shuffled controls originated from selecting 100 shuffled controls from each of the three cell types. This resulted in a library consisting of 60,000 cCREs, for which we ordered one 230-bp oligonucleotide per element.

**Representative subset selection.** Given the limited number of testable elements in the large-scale K562 and WTC11 libraries, we designed a subset selection procedure to more optimally sample a non-redundant subset of elements associated with diverse biochemical features. For K562 cells, we used ground sets of non-overlapping 200-bp windows uniformly covering each of 7 genetic loci; for WTC11 cells, we used ground sets of 83,201 potential enhancers and 19,104 promoters. To perform representative subset selection with these ground sets, we utilized an objective function called facility location. This submodular set function can be optimized using a greedy algorithm, and yields a subset of elements that covers the epigenetic diversity of the ground set[62]. The facility location function is given as:

$$f(A) = \sum_{v \in V} \max_{a \in A} \varphi(v, a)$$

where $V$ is the ground set, $A$ is a subset of $V$ with $k$ elements and $\varphi$ is a nonnegative similarity function. Optimization of the facility function was performed using the Python package apricot (https://github.com/jmschrei/apricot/)[63]. For this study, we set $k = 2,231$ for each of the 7 loci in K562 cells, $k = 30,121$ for WTC11 potential enhancers, and $k = 7,500$ for promoters in WTC11. From the 7 loci, we then filtered out the tiles that overlapped DNase peaks as they had already been tested, and then subsampled to ~2,131 tiles per locus to retrieve 14,918 tiles among all loci.

To assess the pairwise similarity of each element, we utilized hundreds of ENCODE histone and transcription factor ChIP–seq experiments derived from K562 and WTC11-H1 embryonic stem cells (Supplementary Table 6). For each 200-bp tile in the ground set, we computed the mean signal for each ChIP–seq dataset, resulting in a vector of biochemical measurements for each 200-bp tile. We used the Pearson correlation coefficient as a similarity function given these ChIP–seq features.

## Generation of MPRA libraries

The MPRA libraries were generated as previously described[16]. In brief, the Agilent oligonucleotide pool was amplified by 5-cycle PCR using forward primer (pLSmP-enh-f, Supplementary Table 13) and reverse primer (minP-enh-r, Supplementary Table 13) that adds a minimal promoter and spacer sequences downstream of the cCRE. The amplified fragments were purified with 0.8x AMPure XP (Beckman coulter), and amplified for 15 additional cycles using the forward primer (pLSmP-enh-f) and reverse primer (pLSmP-bc-primer-r, Supplementary Table 13) to add 15 bp of random sequence that serves as a barcode. The amplified fragments were then inserted into *SbfI*/*AgeI* site of the pLS-SceI vector (Addgene, 137725) using NEBuilder HiFi DNA Assembly mix (NEB), followed by transformation into 10-beta competent cells (NEB, C3020) using the Gemini X2 machine (BTX). Colonies were allowed to grow up overnight on carbenicillin plates and midiprepped (Qiagen, 12945). For HepG2 and K562 pilot libraries, we collected approximately 1 million and 1.3 million colonies, so that on average 50 and 100 barcodes were associated with each cCRE, respectively. For HepG2, K562 and WTC11 large-scale libraries, we collected approximately 8 million, 12 million and 3 million colonies aiming to associate approximately 50, 50 and 40 barcodes per cCRE, respectively. For the joint library, we collected approximately 3.3 million colonies, aiming to associate approximately 55 barcodes per cCRE. To determine the sequences of the random barcodes and their association to each cCRE, the cCRE-mP-barcodes fragment was amplified from each plasmid library using primers that contain flowcell adapters (P7-pLSmP-ass-gfp and P5-pLSmP-ass-i17, Supplementary Table 13). The fragment was then sequenced with a NextSeq mid-output 300 cycle kit using custom primers (Read 1, pLSmP-ass-seq-R1; Index read, pLSmP-ass-seq-ind1; Read 2, pLSmP-ass-seq-R2, Supplementary Table 13).

## Cell culture, lentivirus packaging and titration

HepG2 (ATCC, HB-8065) and K562 (ATCC, CCL-243) cell culture were performed as previously described[10]. WTC11 human iPS cells (RRID:CVCL_Y803) were cultured in mTeSR plus medium (Stemcell technologies, 100-0276) and passaged using ReLeSR (Stemcell technologies, 100-0484), according to the manufacturer's instructions. WTC11 cells were used for the MPRA experiments at passage 43–49. Cells were not authenticated or checked for mycoplasma contamination. Lentivirus packaging was performed using HEK293T (ATCC, CRL-3216), as previously described with modifications[16]. In brief, 50,000 cells per cm² HEK293T cells were seeded in T175 flasks and cultured for 48 h. The cells were co-transfected with 7.5 μg per flask of plasmid libraries, 2.5 μg per flask of pMD2.G (Addgene 12259), and 5 μg per flask of psPAX2 (Addgene 12260) using EndoFectin Lenti transfection reagent (GeneCopoeia) according to the manufacturer's instructions. After 8 h, cell culture media was refreshed and ViralBoost reagent (Alstem) was added. The transfected cells were cultured for 2 days to complete lentivirus packaging. The lentivirus libraries in the culture media were separated from the HEK293T cells and concentrated using the Lenti-X concentrator (Takara) according to the manufacturer's protocol. To measure DNA titre for the lentiviral libraries in HepG2, K562, or WTC11, cells were seeded at $1 \times 10^5$ cells per well in 24-well plates and incubated for 24 h. Serial volume (0, 2, 4, 8, 16 and 32 μl) of the lentivirus was added along with Polybrene at a final concentration of 8 μg ml⁻¹. The infected cells were cultured for three days and then washed with PBS three times. Genomic DNA was extracted using the Wizard SV genomic DNA purification kit (Promega). Multiplicity of infection (MOI) was measured as relative amount of viral DNA (WPRE region, forward; 5′-TACGCTGCTTTAATGCCTTTG-3′, reverse; 5′-GGGCCACAACTCCTCATAAAG-3′) over that of genomic DNA (intronic region of LIPC gene, forward; 5′-TCCTCCGGAGTTATTCTTGGCA-3′, reverse; 5′-CCCCCCATCTGATCTGTTTCAC-3′) by quantitative PCR using SsoFast EvaGreen Supermix (Bio-Rad), according to the manufacturer's protocol.

## Lentiviral infections and DNA and RNA barcode sequencing

For the HepG2 and K562 pilot libraries, 2.4 M HepG2 or 10 M K562 cells per replicate were seeded in 10 cm dishes or T75 flasks, respectively, and incubated for 24 h. The HepG2 and K562 cells were infected with the lentiviral libraries along with 8 μg ml⁻¹ Polybrene, with an estimated MOI of 50 or 10, respectively. The higher MOI in HepG2 is due to these cells being adherent compared to K562 that grow in suspension. For the large-scale HepG2 library, 15 M HepG2 cells per replicate were seeded in 3×15 cm dishes (5 million per dish), incubated for 24 h, and infected with the library along with 8 μg ml⁻¹ Polybrene, with an estimated MOI of 50. For the large-scale K562 library, 85 million K562 cells per replicate were seeded in 3 T225 flasks (28.3 M per flask), incubated for 24 h, and infected with the library along with 8 μg ml⁻¹ Polybrene, with an estimated MOI of 10. For the large-scale WTC11 library, 38.4 million WTC11 cells per replicate were seeded in four 10 cm dishes (9.6 M per dish), incubated for 24 h, and infected with the library along with 8 μg ml⁻¹ Polybrene, with an estimated MOI of 10, due to higher MOIs being lethal for these cells. For the joint library, 5 million HepG2, 28 million K562 and 38.4 million WTC11 cells were infected with the estimated MOI of 50, 10 and 10, respectively. For each experiment, three independent infections were performed to obtain three biological replicates. After three days of culture, genomic DNA and total RNA were extracted from the infected cells using AllPrep DNA/RNA mini kit (Qiagen), and sequencing library preparations were performed as previously described[16]. The libraries were then sequenced with a NextSeq high-output 75 cycle kit using custom primers (Read 1, pLSmP-5bc-seq-R1; Index1 (unique molecular

idenitifier (UMI) read), pLSmP-UMI-seq; Index2, pLSmP-5bc-seq-R2; Read 2, pLSmP-bc-seq; Supplementary Table 13)[16].

## MPRA processing pipeline

**Associating barcodes to designed elements.** For each of the barcode association libraries, we generated FASTQ files with bcl2fastq v.2.20 (parameters "--no-lane-splitting --create-fastq-for-index-reads --use-bases-mask Y*,I*,I*,Y*"), splitting the sequencing data into paired-end index files delineating the barcodes (I1 and I2) and paired-end read files delineating the corresponding element linked to the barcode (R1 and R2). These files were used to associated barcodes to elements using the association utility of MPRAflow 1.0[16] (run as: nextflow run association.nf --fastq-insert "R1.fastq.gz" --fastq-insertPE "R2.fastq.gz" --fastq-bc "I1.fastq.gz" --fastq-bcPE "I2.fastq.gz" --aligner "bt2_strand" --design "designed_sequences.fa"). Here, designed_sequences.fa was a FASTA file incorporating all of the element sequences that had been ordered from the corresponding Agilent library, and bt2_strand was used to map elements in a strand-aware fashion to accommodate the existence of elements tested in both orientations. The final output of this utility was a filtered_coords_to_barcodes.pickle file mapping barcodes to elements.

**Replicates, normalization and RNA/DNA activity scores.** For each of the indexed DNA and RNA libraries, we demultiplexed the sequencing run and generated Fastq files with bcl2fastq v.2.20 (parameters "--barcode-mismatches 2 --sample-sheet SampleSheet.csv --use-bases-mask Y*,Y*,I*,Y* --no-lane-splitting --minimum-trimmed-read-length 0 --mask-short-adapter-reads 0"), where SampleSheet.csv catalogued the correspondence between the index sequence and DNA or RNA replicate sample of origin. In several instances, the "--barcode-mismatches 2" resulted in an index assignment clash, requiring us to instead use "--barcode-mismatches 1". These commands split the sequencing data into paired-end read files delineating the barcodes (R1 and R3) and a file indicating the UMI (R2) for each DNA or RNA replicate sample. We compiled a table of these files to indicate the 3 RNA and 3 DNA files for each of the three replicates in the file experiment.csv. Finally, we used the count utility of MPRA-flow 1.0[16] (run as: nextflow run count.nf -e "experiment.csv" --design "designed_sequences.fa" --association "filtered_coords_to_barcodes. pickle") to compute activity scores for each element and replicate as $\log_2$(RNA/DNA). Elements with which were measured with fewer than 10 independent barcodes were removed to reduce the impact of measurement noise in downstream analysis. This filter led to the following number of retained elements: (1) HepG2 pilot library, 9,153/9,372 (97.7%); (2) K562 pilot library, 7,323/7,500 (97.6%); (3) HepG2 large-scale library, 139,886/164,307 (85.1%); (4) K562 large-scale library, 226,255/243,780 (92.8%); (5) WTC11 large-scale library, 56,093/75,542 (74.2%); (6) HepG2 joint library, 56,018/60,000 (93.4%); (7) K562 joint library, 56,008/60,000 (93.3%); and (8) WTC11 joint library, 55,983/60,000 (93.3%). To combine the data from all three replicates, the distribution of activity values was normalized to the median activity value within each replicate, and then the activity values were averaged across the three replicates.

## Regression modelling

**Biochemical model features.** We extracted all transcription factor ChIP–seq, histone ChIP–seq, DNase-seq, and ATAC-seq bigWig files available for HepG2, K562, and WTC11 cells for the hg38 human genome assembly under 'released' ENCODE status[56]. To account for the lack of WTC11 data available, we also collected all such datasets for H1-ESCs for inclusion in the predictive model. This resulted in 1,506 bigWig files for HepG2 cells; 1,206 files for K562 cells; and 277 files for WTC11/H1-ESCs (Supplementary Table 6). For each candidate element aside from controls, we computed the mean bigWig signal extracted from the corresponding region of the human genome using bigWigAverageOverBed[60].

All data was right-skewed, and was therefore log-transformed (that is, after adding a pseudocount of 0.1) to approximate a normal distribution. Finally, for each cell type, multiple replicates corresponding to the same 'experiment target' (Supplementary Table 6) were averaged to compute the consensus signal for each target in each cell type. This led to a total of 655 HepG2 features, 447 K562 features and 122 WTC11/H1-ESC features considered by the models.

**Sei and EnformerMPRA model features.** For the large-scale libraries, Sei[37] and Enformer[36] were used to predict element activity in both orientations (that is, including adaptors in a fixed orientation to simulate the MPRA experiment). The resulting 21,907 Sei and 5,313 Enformer predictions for each of the two orientations were averaged. For the joint library, Sei and Enformer were used to predict element activity in only the forward/sense orientation, and the resulting human predictions were carried forward as features. As Sei requires an input sequence length of 4,000 bp and Enformer requires one of 196,608 bp, all elements were extended with "N" padding in both directions while centring on the element sequence.

**Data pre-processing and model training.** For each of the three large-scale libraries, the $\log_2$(RNA/DNA) scores for each element were averaged among both orientations in which the element was tested, and then randomly assigned to one of ten cross-validation folds (Supplementary Table 8). All predictive features (that is, biochemical features from the matched cell type, or all Enformer features) were $z$-score-normalized to scale the features similarly. This enabled a direct comparison of coefficients among features derived from the resulting linear models. As described before[10,14,32], for each regression task we optimized the $\lambda$ regularization hyperparameter using tenfold cross-validation, and then used the optimal value for $\lambda$ to train 10 lasso regression models, each on 9 of the 10 folds of data, to evaluate the performance of each model on the held-out fold. To evaluate the most relevant features selected, we trained a lasso regression model on the full dataset and visualized the 30 coefficients with the greatest magnitude. A similar strategy was used for data from the joint library tested in all three cell types, ensuring that the same element measured in different cell types was always assigned to the same fold (Supplementary Table 12).

**Training MPRALegNet.** The LegNet architecture[35] was adapted to the training data in the following ways: (1) to account for longer sequences but smaller training set size compared to the original LegNet, we added max pooling layers after each local block; (2) the kernel size and number of blocks were selected to match the model's receptive field to the sequence length; (3) weight decay during training was increased to prevent model overfitting; (4) gradient clipping was used to avoid gradient explosion during the one-cycle learning rate policy (see Supplementary Methods for more details). For the training-time augmentation, each sequence was provided twice, both in the forward and reverse complement orientations, along with their corresponding measured element activity scores. At test time, the model predicted four scores: (1) the scores for the forward and reverse element orientations relative to the fixed flanking (adaptor) regions; and (2) as a further augmentation, the scores for the full reverse complement sequences (that is, obtaining the reverse complementary sequence of the element and adaptor regions together) from step (1). The final prediction represented an average of these four values. The reproducible code, including implementation and complete parameter settings, is available on GitHub (https://github.com/autosome-ru/human_legnet) and Zenodo (https://zenodo.org/records/10558183 and https://zenodo.org/records/13908857).

**Interpreting motifs identified by MPRALegNet.** As a step towards motif interpretation, ISM was performed for all possible single

nucleotide variants on each 200-bp sequence. Owing to the nature of our cross-validation strategy, for each sequence there were nine models for which the sequence was held out during training. ISM scores were generated for every sequence by averaging the predictions from these nine models. The average reference sequence prediction was then compared with that of the alternative sequence[32]. We then interrogated our ISM scores to identify the most pertinent motifs associated with changes in variant activity using TF-MoDISco-lite v.2.0.4 (https://github.com/jmschrei/tfmodisco-lite), a more efficient version of TF-MoDISco[38]. The TF-MoDISco-lite algorithm was used with default settings and similar seqlet patterns were matched against JASPAR 2022 CORE vertebrate non-redundant database[64] using Tomtom[65].

**Modelling dose-dependent and combinatorial motif effects learned by MPRALegNet.** A non-redundant set of positional weight matrices (PWMs) from each cell type, as ranked by TF-MoDISco-lite (Extended Data Fig. 6), were extracted and scanned (using FIMO 5.5.4[66], parameters "--text --thresh 0.001") along each promoter and potential enhancer that was tested bidirectionally. The motif scans were summarized into a matrix of counts for each transcription factor and element tested, as well as log-likelihood (sum of the log(probabilities)), reflecting the likelihood of a given transcription factor binding the element while considering all TFBS instances in both orientations and their respective binding affinities. We then performed an analysis of homotypic (that is, dose-dependent effects for a single transcription factor) as well as heterotypic (that is, combinatorial effects among pairs of transcription factors) for the top 10 activating transcription factors of each cell type.

For homotypic analysis, we plotted the median element activity (that is, both predicted and observed) for elements possessing 0, 1, 2, 3, 4, or 5 motifs, filtering away elements with ≥1 site to any of the other top 10 transcription factors to reduce the chances of a confounding effect. Groups with a sample size of <10 were also filtered out to minimize the impact of noise. The expected dose-dependent responses (for example, dashed lines in Fig. 3f) were computed using linear regression models examining the relationship between either the observed or MPRALegNet-predicted MPRA activity and the number of TFBSs, given log-transformed and untransformed space to model either multiplicative or additive effects, respectively. The expected trend for multiple sites was extrapolated on the basis of the slope and intercept terms of these linear models.

For heterotypic analysis, we evaluated every pair of the 10 activating motifs, isolating cases in which the element possessed 0 counts of both transcription factors, 1 count of one transcription factor or the other, or 1 count each of the first and second transcription factor. Again, all elements were filtered to those with ≥1 site to any of the other top 10 transcription factors other than the transcription factor pair considered. To further account for confounding effects that could be attributable to all other transcription factors (that is, including those beyond the top 10), we computed the residuals from a linear model which considered the log-likelihood values for all other transcription factors besides the pair of transcription factors under consideration. We call these 'adjusted $\log_2$(RNA/DNA)' (for example, $y$ axis in Fig. 3h) because they removed variability explained by the binding affinities and occurrences of other transcription factors. Finally, a regression model was fit independently to the predicted and observed activity scores. This model sought to predict activity as a function of the presence of TF1, TF2 or an interaction term (TF1 × TF2). The coefficient for the interaction term represented the strength of the super-multiplicative effect (that is, if the coefficient was positive) or the sub-multiplicative effect (that is, if the coefficient was negative)[41,42].

**Prediction using MPRALegNet.** To generate predictions on an arbitrary sequence, we recommend generating predictions using all 90 pretrained models (considering test-time sequence augmentations such as orientation and shifting for extra precision), and then averaging the predictions to achieve the final prediction. We recommend replacing the fixed 15-bp adaptors with the surrounding natural genomic sequence context whenever available, to reduce the chances that artefactual motifs occurring at the adaptor-sequence boundaries could bias the results.

**Calculation of element specificity scores.** To compute element specificity scores (ESSs) using activity scores from the joint library, $\log_2$(RNA/DNA) values from each cell line were first $z$-score-transformed. Then an ESS for each element was computed by subtracting the element's score in each cell type by the mean element score across cell types. A full table of ESSs is provided (Supplementary Table 10).

### Reporting summary

Further information on research design is available in the Nature Portfolio Reporting Summary linked to this article.

## Data availability

Raw sequencing data and processed files generated in this study are available in the ENCODE portal for the pilot libraries (HepG2: ENCSR463IRX; K562: ENCSR460LZI), large-scale libraries (HepG2: ENCSR022GQD; K562: ENCSR382BVV; WTC11: ENCSR244FWB), and joint libraries (HepG2: ENCSR405QCT; K562: ENCSR203UFY; WTC11: ENCSR336MKI).

## Code availability

Code to train and interpret MPRAnn and MPRALegNet is available at https://github.com/visze/sequence_cnn_models and https://github.com/autosome-ru/human_legnet. Pretrained models and code have also been deposited at Zenodo (https://zenodo.org/records/10558183 (ref. 67) and https://zenodo.org/records/13908857 (ref. 68)).

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

**Acknowledgements** We thank members of the ENCODE4 Consortium who have provided critical feedback throughout this project; the groups of R. Tewhey, J. Stamatoyannopoulos and A. Kundaje for sharing MPRA data, DNase data and peak calls, respectively; I. Vorontsov for assistance with HOCOMOCO motif clustering; D. Nogina, E. Aristova and N. Gryzunov for assistance with MPRALegNet finalization; and J. Engreitz for assistance and input on scoring of the ABC model. Research was supported in part on work supported under an NRSA NIH fellowship 5T32HL007093 (to V.A.); ASHBi, sponsored by the World Premier International Research Center Initiative (WPI), MEXT, Japan (to F.I.); DFG grant 464313370 (to M.S. and P.M.D.); MPRALegNet tuning (I.V.K. and D.P.) was supported by RSF grant 20-74-10075 (to I.V.K.); ASV analysis (I.V.K.) was supported by MSHERF grant 075-15-2021-1344; grants from

the National Human Genome Research Institute (NHGRI) U24 HG009446 (to W.S.N.), and UM1HG009408 (to J.S. and N.A.) and UM1HG011966 (to M.K., J.S. and N.A.). J.S. is an Investigator of the Howard Hughes Medical Institute.

**Author contributions** V.A. designed experiments, performed most computational analyses and generated figures and tables. F.I. performed most MPRA experiments and wrote the associated methods sections. B.K.M., A.S. and Z.Z. helped with cloning and sequencing samples. M.S. trained MPRAnn and helped to interpret deep learning models. P.M.D. helped generate Enformer, Sei and MPRALegNet predictions as well as STREME results. P.K. performed barcode downsampling analysis. P.M.D. and P.K. performed their work with supervision from M.K. and M.S. V.A., M.S., P.M.D. and M.K. interpreted modelling and STREME results. D.P. trained MPRALegNet, performed performance optimization and ASV analysis under the supervision of I.V.K. I.G.-S. helped to guide MPRALegNet interpretation with respect to combinatorial TFBS effects. V.A. and J.Z. deposited data into the ENCODE portal. G.G.Y. and W.S.N. developed and performed the representative subset selection strategy. V.A. and N.A. wrote most of the paper with additional help from all authors. J.S. and N.A. supervised the study.

**Competing interests** V.A. is an employee of Sanofi, but performed this work independently of Sanofi. J.S. is a scientific advisory board member, consultant and/or co-founder of Cajal Neuroscience, Guardant Health, Maze Therapeutics, Camp4 Therapeutics, Phase Genomics, Adaptive Biotechnologies, Scale Biosciences, Somite Therapeutics, Sixth Street Capital and Pacific Biosciences. N.A. is a co-founder and on the scientific advisory board of Regel Therapeutics and receives funding from BioMarin Pharmaceutical Incorporated. All other authors declare no competing interests.

**Additional information**
**Correspondence and requests for materials** should be addressed to Vikram Agarwal, Jay Shendure or Nadav Ahituv.

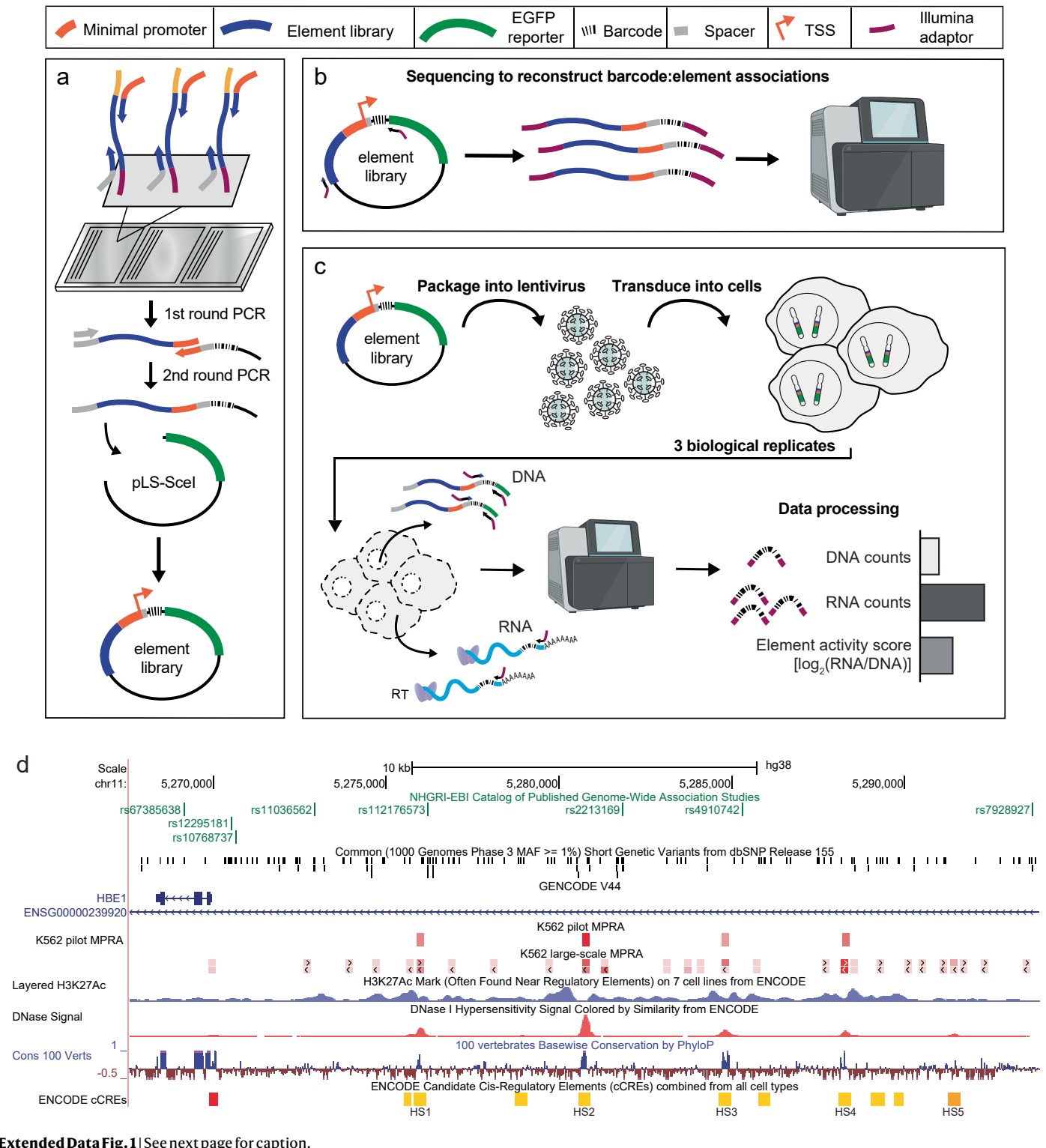

**Extended Data Fig. 1** | See next page for caption.

**Extended Data Fig. 1 | A next-generation lentiviral massively parallel reporter assay (lentiMPRA) strategy to measure the transcriptional regulatory activity of > 6,000–240,000 enhancers simultaneously.** **a**, Designed 230nt oligos corresponding to thousands of cCREs are synthesized on an Agilent array. The 1st round of PCR adds on a minimal promoter, while the 2nd round of PCR adds random barcodes to these sequences. The library is then cloned into a pLS-SceI vector harboring an EGFP reporter to generate the final element library. **b**, The element-barcode fragments within the library are amplified by PCR and sequenced using an Illumina NextSeq instrument. This enables reconstruction of element-barcode pairings. **c**, The element library is packaged into lentiviruses and transduced into HepG2, K562, or WTC11 cells in a series of three replicates. Cells are grown in cultured medium for three days prior to the harvesting of RNA and DNA. Each RNA and DNA sample from each replicate is extracted, and barcodes are sequenced on an Illumina NextSeq instrument. Finally, DNA and RNA-derived barcodes are counted to compute a normalized activity score for each element in each replicate. **d**, UCSC genome browser tracks annotating, from top to bottom: i) Lead single nucleotide polymorphisms (SNPs) from published Genome-wide Association Studies (GWAS); ii) Common SNPs from the 1000 Genomes Phase 3 dataset; iii) GENCODE gene track; iv) MPRA activity scores from the pilot K562 MPRA library for each of the five enhancers tested, with stronger red indicative of higher activity; v) MPRA scores corresponding to the large-scale K562 MPRA library, tested in both orientations; vi) H3K27Ac; vii) DNase I hypersensitivity signal in K562 cells; viii) base conservation among 100 vertebrate species; and ix) the five enhancers (HS1-HS5) of the globin locus tested in the pilot and large-scale K562 MPRA libraries. Image of DNA sequencer created with BioRender.com.

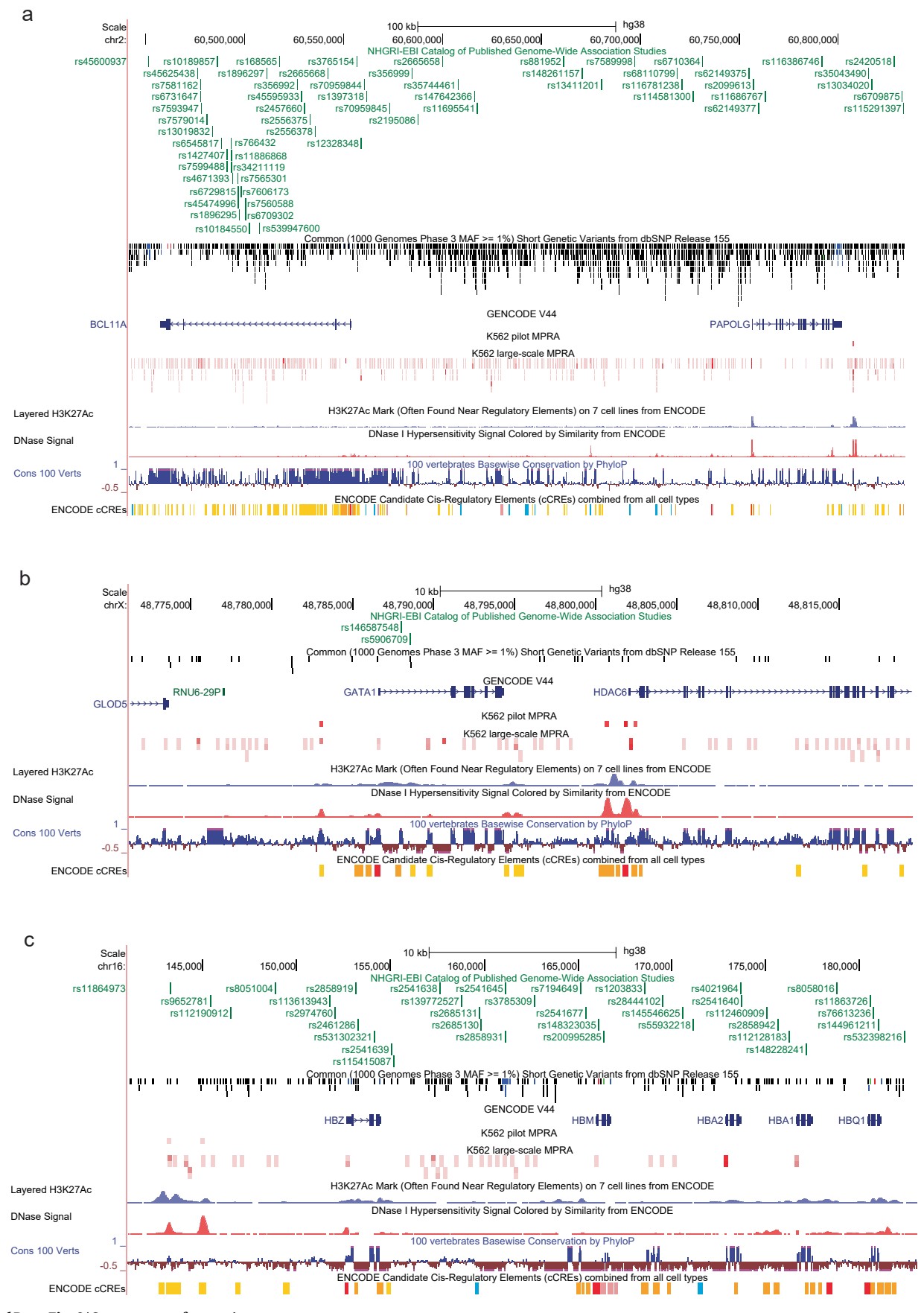

**Extended Data Fig. 2** | See next page for caption.

**Extended Data Fig. 2 | MPRA activity in selected disease loci. a-f**, UCSC genome browser tracks annotating, from top to bottom: i) Lead single nucleotide polymorphisms (SNPs) from published Genome-wide Association Studies (GWAS); ii) Common SNPs from the 1000 Genomes Phase 3 dataset; iii) GENCODE gene track; iv) MPRA activity scores from the pilot K562 MPRA library for each of the five enhancers tested, with stronger red indicative of higher activity; v) MPRA scores corresponding to the large-scale K562 MPRA library, tested in both orientations; vi) H3K27Ac and vii) DNase I hypersensitivity signal in K562 cells; viii) base conservation among 100 vertebrate species. Snapshots provided for *BCL11A* (**a**), *GATA1* (**b**), and *HBA2* (**c**). Additional loci are shown in Supplementary Fig. 3.

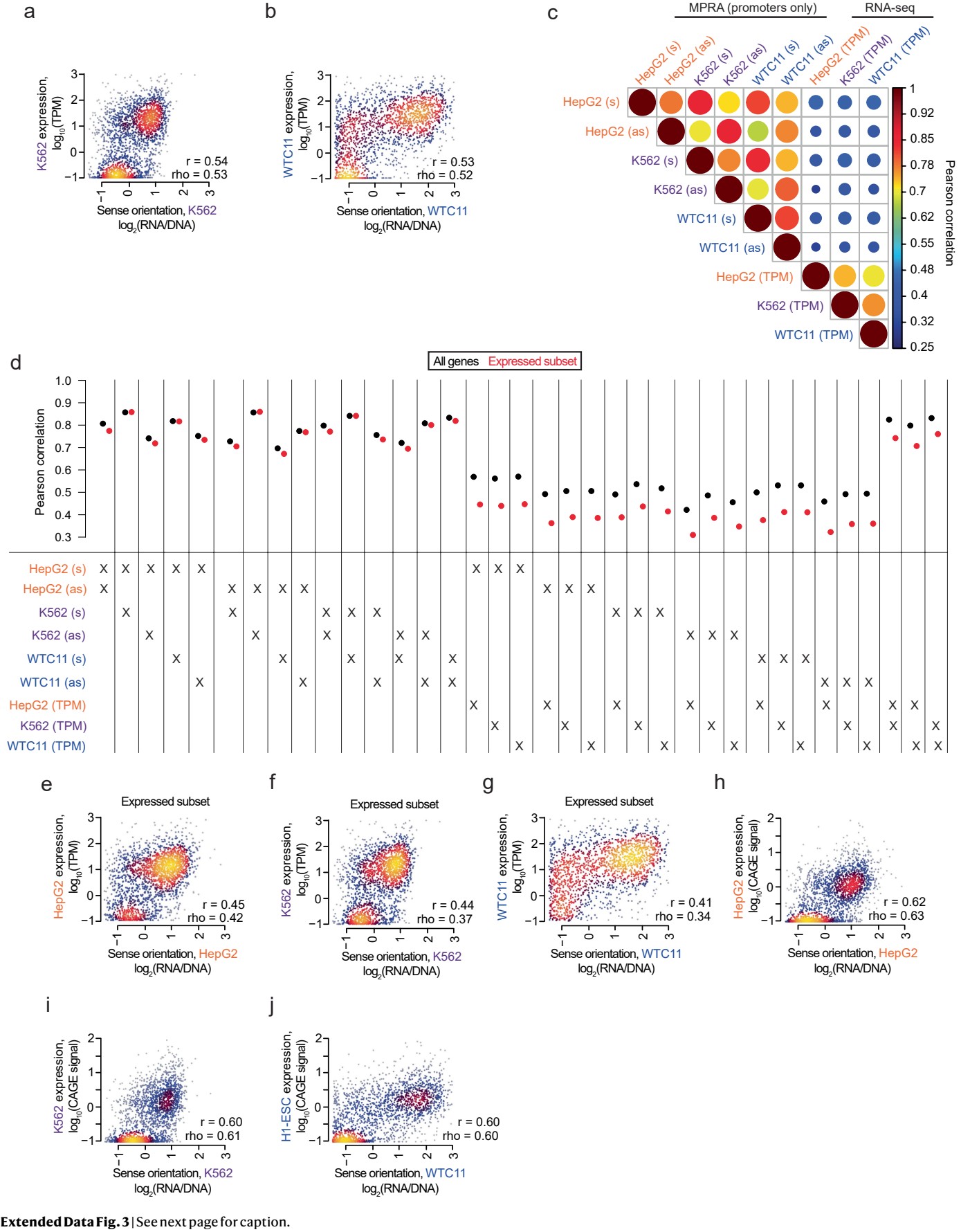

**Extended Data Fig. 3** | See next page for caption.

**Extended Data Fig. 3 | Properties of promoter activity in three cell types.**
**a-b**, Scatter plots of activity scores for sense-oriented promoters tested in the MPRA and endogenous gene expression levels for **(a)** K562 and **(b)** WTC11 cells. Expression levels follow a bimodal distribution. Also indicated are the Pearson (r) and Spearman (rho) correlation values. **c**, Upper triangular heatmap indicating the correlation between the sense (s) and antisense (as) orientations of promoters tested in the MPRA as well as endogenous gene expression levels measured in transcripts per million (TPM) using RNA-seq, filtered for the set of genes with detectable expression (*i.e.*, > 0 TPM). The sizes of the circles are proportional to the Pearson correlations. **d**, Alternative representation of the data shown in Fig. 2d and panel (c), showing the Pearson correlation between each pair of measurements indicated below the horizontal line. Black points represent all genes (*i.e.*, akin to Fig. 2d) and red points represent the expressed subset of genes [*i.e.*, akin to panel (c)]. **e-g**, Scatter plots of activity scores for sense-oriented promoters tested in the MPRA and endogenous gene expression levels for **(e)** HepG2, **(f)** K562, and **(g)** WTC11 cells, filtered for the set of genes with detectable expression (*i.e.*, > 0 TPM). **h-j**, Scatter plots of activity scores for sense-oriented promoters tested in the MPRA and endogenous gene expression levels for HepG2 (**h**), K562 (**i**), and WTC11 (**j**) cells, as measured by CAGE-seq signal[58] in the precise promoter tested. Due to lack of availability of processed CAGE-seq for WTC11, we instead used H1-ESCs, a transcriptionally similar embryonic stem cell line.

**a**

| HepG2 | Description | K562 | Description | WTC11 | Description |
|---|---|---|---|---|---|
| (logo) | ETV5/7 | (logo) | KLF-related | (logo) | KLF-related |
| (logo) | CpG-rich | (logo) | ETS-related | (logo) | ETS-related |
| (logo) | CpG-rich | (logo) | NFYA/B/C | (logo) | CpG-rich |
| (logo) | CpG-rich | (logo) | NRF1 | (logo) | THAP11 |
| (logo) | CpG-rich | (logo) | CpG-rich | (logo) | CpG-rich |
| (logo) | CpG-rich | (logo) | CpG-rich | (logo) | CpG-rich |
| (logo) | NFYA/B/C | (logo) | ATF1/FOSL1::JUND | (logo) | CpG-rich |
| (logo) | NFYA/B/C | (logo) | THAP11 | (logo) | CpG-rich |
| (logo) | NFYB | (logo) | NRF1 | (logo) | NFYA/B/C |
| (logo) | THAP11 | (logo) | NFYA/B/C | (logo) | TFE3/USF2/MLX |
| (logo) | CpG-rich | (logo) | CpG-rich | (logo) | YY1 |
| (logo) | KLF-related | | | (logo) | ZNF680 |
| | | | | (logo) | ELF2/4 |

| HepG2 | Description | K562 | Description | WTC11 | Description |
|---|---|---|---|---|---|
| (logo) | FOXD3 | (logo) | NFIB | (logo) | ZIC3 |
| (logo) | SCRT2 | (logo) | PLAGL2 | (logo) | NFIB |
| | | (logo) | ZEB1 | (logo) | THAP1 |
| | | | | (logo) | EBF1 |
| | | | | (logo) | HNF4A |

**b**

| HepG2 | Description | K562 | Description | WTC11 | Description |
|---|---|---|---|---|---|
| (logo) | FOSB::JUNB | (logo) | GATA2/3/5/6 | (logo) | FOSB::JUNB |
| (logo) | FOXB1/FOXC1 | (logo) | GATA1/2/6 | (logo) | POU5F1::SOX2 |
| (logo) | JUND | (logo) | FOSB::JUNB | (logo) | NFYA/C |
| (logo) | TCF7/TCF7L2 | (logo) | ETS-related | (logo) | GATA3 |
| (logo) | HNF4A | (logo) | STAT1/4/5A/5B | (logo) | SOX4 |

| HepG2 | Description | K562 | Description | WTC11 | Description |
|---|---|---|---|---|---|
| (logo) | CTCF | (logo) | TP53 | (logo) | ELF1/3 |
| (logo) | ZFP335 | | | (logo) | GABPA |
| (logo) | KLF-related | | | (logo) | POU2F2 |
| (logo) | ZNF460 | | | | |

| K562 | Description |
|---|---|
| (logo) | CTCF |
| (logo) | TCF3/4 |
| (logo) | TCFL5 |
| (logo) | THAP11 |
| (logo) | ZNF460 |

| WTC11 | Description |
|---|---|
| (logo) | CTCF |

**Extended Data Fig. 4 | Enriched motifs detected in three cell types. a-b**, Set of motifs enriched in the top 1,000 most active vs. bottom 1,000 least active promoters (**a**) or potential enhancers (**b**) (*i.e.*, as measured by large-scale MPRAs). Motifs were discovered by STREME[67] for each of the three cell types evaluated, and matched against the JASPAR 2022 CORE vertebrate non-redundant database[64] using Tomtom[65] (*i.e.*, other than the set of CpG-rich motifs). Motifs above the horizontal line in each panel are those associated with gene activation, while motifs below the line are those associated with repression.

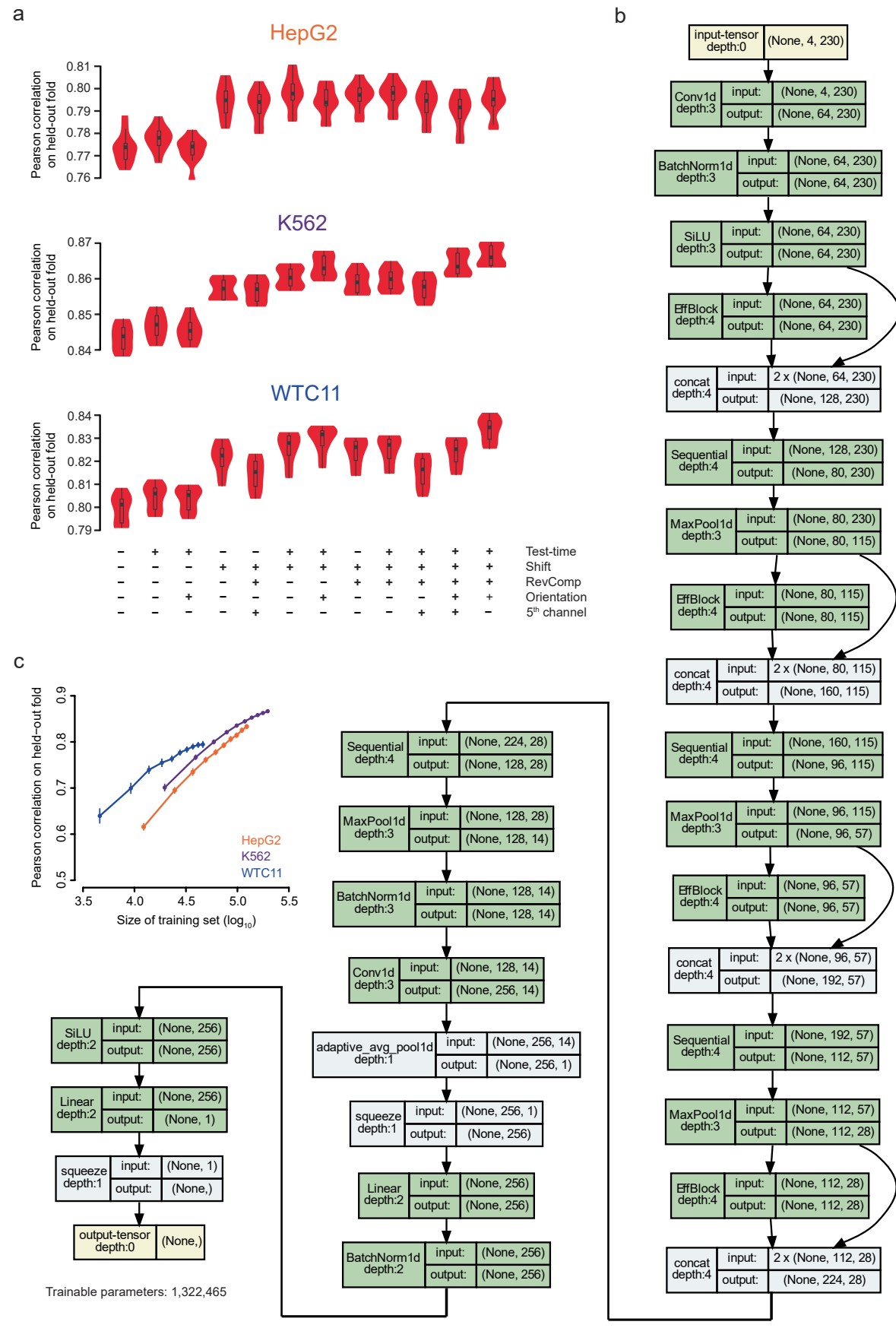

**Extended Data Fig. 5** | See next page for caption.

**Extended Data Fig. 5 | Architecture and performance of MPRALegNet.**
**a**, Violin plots showing the performances of different variations of MPRALegNet on each of the ten cross-validation folds of held-out data, for different types of augmentations. "-" and "+" indicate removal or usage, respectively, of the following augmentations: i) "test-time", whereby the mean prediction is computed for various augmentations of the test sequence; ii) "shift", whereby a sequence was randomly shifted by 0 to +21 bp; iii) "RevComp", whereby a sequence was randomly reverse complemented; iv) "Orientation", whereby measured element activity scores were considered for each orientation tested, instead of the mean across both orientations; and v) "5th channel", whereby a 5th channel was considered alongside the one-hot encoded sequence (*i.e.*, the first 4 channels) to indicate the sequence's orientation. **b**, Complete architecture of the MPRALegNet model. Indicated for each layer is the layer name and dimensionality of the input and output matrices. 'None' refers to the batch size used during model training. **c**, Impact of the size of the training set on model performance. Data from each cell type were downsampled to every 10th percentile (*i.e.*, from 10 to 100%). Error bars represent the standard deviation of the Pearson correlations across 90 models (10 held-out folds of data x 9 trained models varying by the choice of validation set).

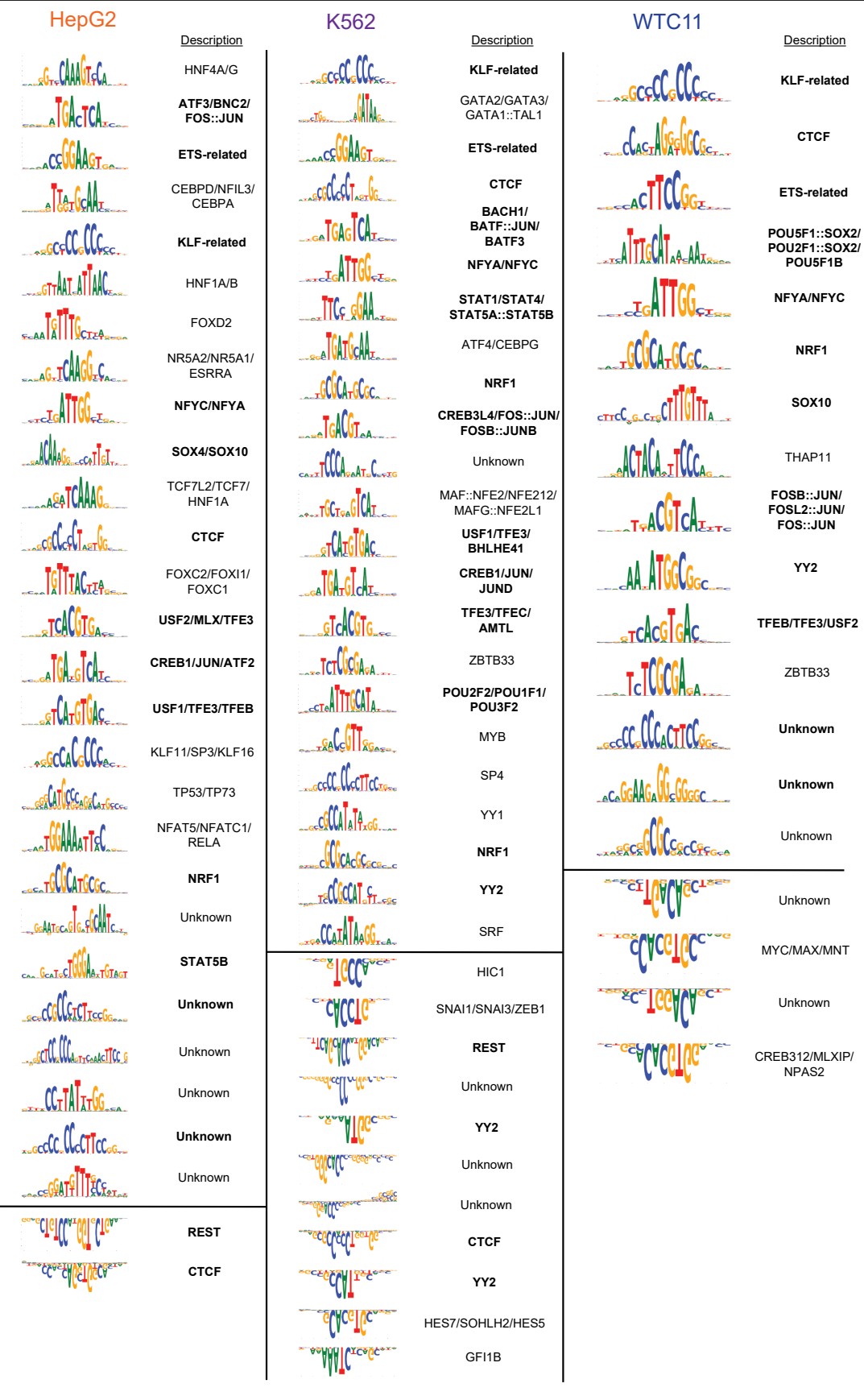

**Extended Data Fig. 6 | Motifs detected by MPRALegNet.** Set of enriched motifs discovered by TF-MoDISco-lite[38] for each of the three cell types evaluated. Motifs shown are rank-ordered according to their "seqlet"[38] count. TFBSs associated with transcriptional inhibition (*e.g.*, REST) are oriented upside down and shown below the horizontal lines. TFBSs detected in at least two cell types (*i.e.*, likely bound to housekeeping TFs) are shown in bold.

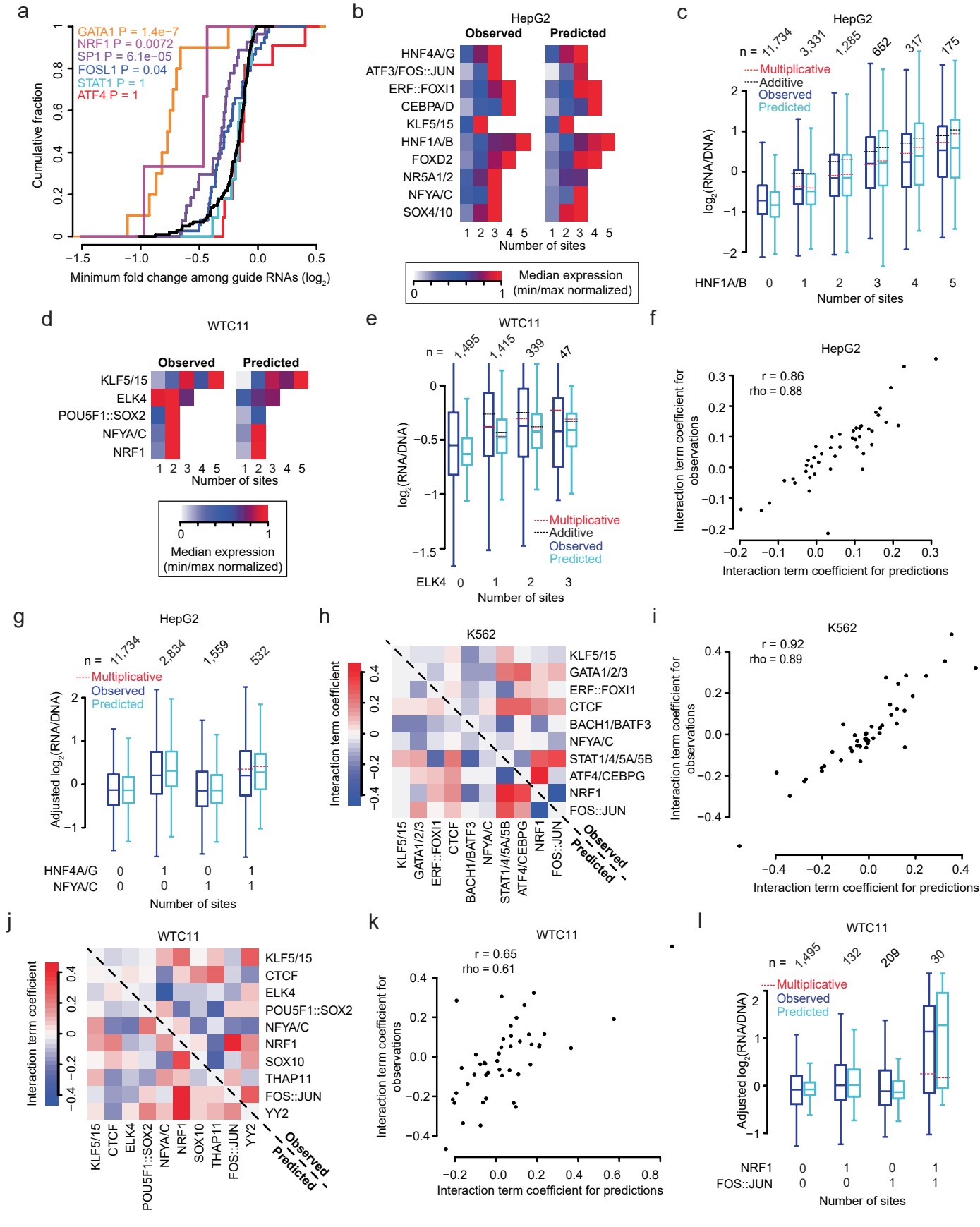

**Extended Data Fig. 7 |** See next page for caption.

**Extended Data Fig. 7 | Combinatorial TFBS effects learned by MPRALegNet.**
**a**, Effect of TF knockdown on loss of regulatory element activity, from a reanalysis of a prior MPRA[40]. Shown are cumulative density plots for the subset of elements possessing TF binding sites to the corresponding knocked down TF, relative to a control (*i.e.*, non-targeting guide RNA) shown in black. The x-axis indicates the minimum (*i.e.*, most negative) fold change among three guide RNAs targeting the TF. Data shown is from the high MOI condition sampled at day 10[40]. P-values indicate a shift in the distribution as assessed by a one-sided Kolmogorov-Smirnov (K-S) test, followed by a Bonferroni multiple hypothesis testing correction. **b-e**, These panels are arranged in the same scheme as Fig. 3e,f, except display results for homotypic TFBSs in HepG2 (**b-c**) and WTC11 (**d-e**) cells

for the indicated TF families. **f**, Scatter plot of interaction terms fit to predicted and observed values for TFBS pairs in HepG2 cells. The data is the same as that presented in Fig. 3g, but also includes Pearson (r) and Spearman (rho) correlation values. **g**, This panel is similar to that shown in Fig. 3h, but shows an example of a pair of heterotypic TFBSs that exhibit a sub-multiplicative effect when co-occurring. **h-k**, These panels are arranged in the same scheme as Fig. 3g and panel (**f**), except display results for K562 (**h-i**) and WTC11 (**j-k**) cells for the indicated TF families. **l**, This panel is similar to that shown in Fig. 3h, but shows an example of a pair of heterotypic TFBSs that exhibit a super-multiplicative effect when co-occurring in WTC11 cells.

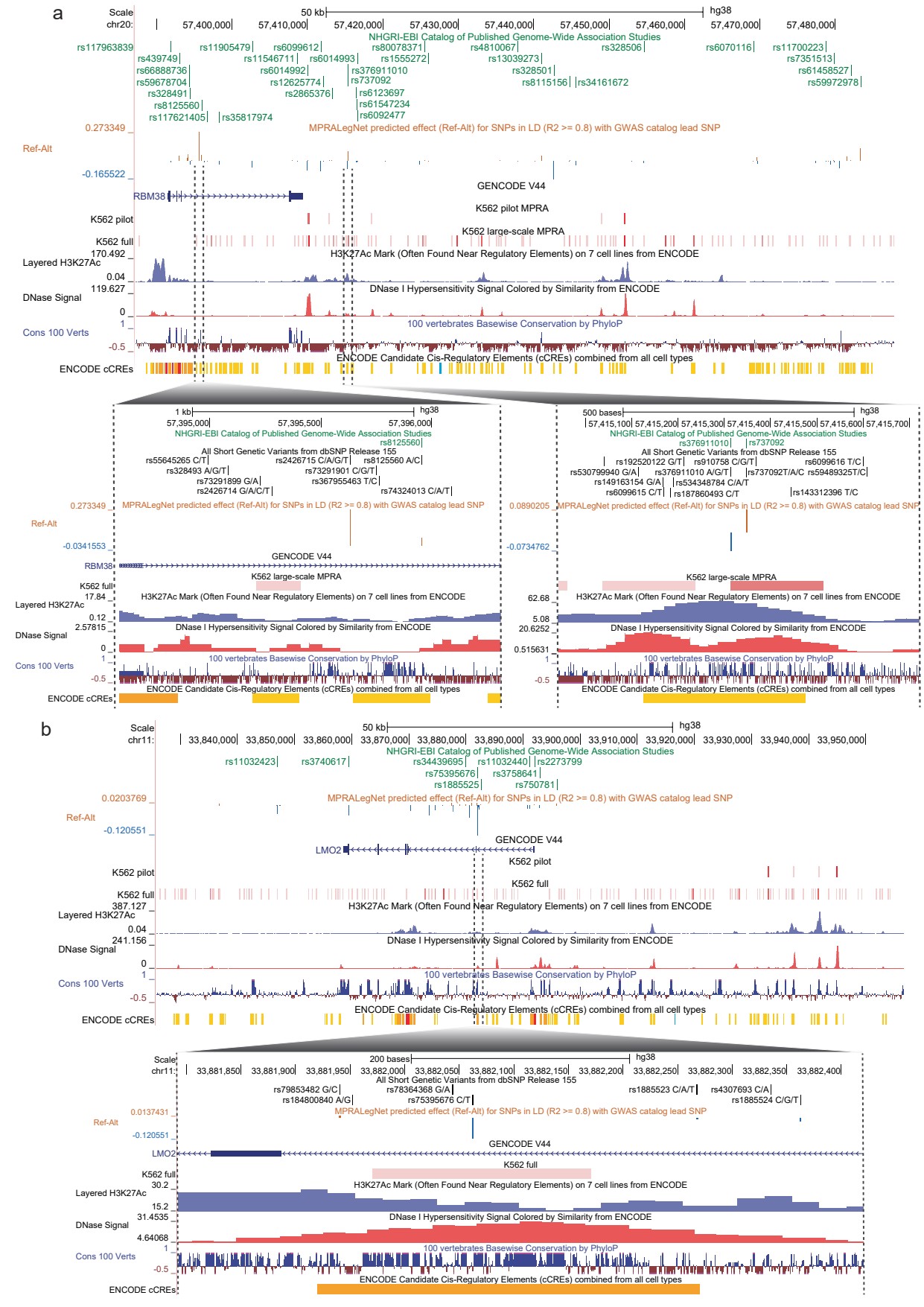

**Extended Data Fig. 8** | See next page for caption.

**Extended Data Fig. 8 | Variant effect predictions in the *RBM38* and *LMO2* loci. a**, UCSC genome browser snapshot of the *RBM38* locus, showing from top to bottom: i) Lead SNPs from published GWAS; ii) variant effect predictions derived from MPRALegNet for LD variants with GWAS lead SNPs ($R^2 \geq 0.8$); iii) MPRA activity scores from the pilot K562 MPRA library for each of the five enhancers tested, with stronger red indicative of higher activity; iv) MPRA scores corresponding to the large-scale K562 MPRA library, tested in both orientations; v) H3K27Ac and vi) DNase I hypersensitivity signal in K562 cells; vii) base conservation among 100 vertebrate species; viii) ENCODE cCRE track. The bottom panel shows two zoomed-in regions of the implicated causal SNPs (*i.e.*, expanded from the vertical dashed lines), several of which are located within a DNase I site. **b**, This panel follows the same scheme, except displays the *LMO2* locus and one zoomed in region showing the implicated causal SNP located within a DNase I site having the strongest predicted effect size.

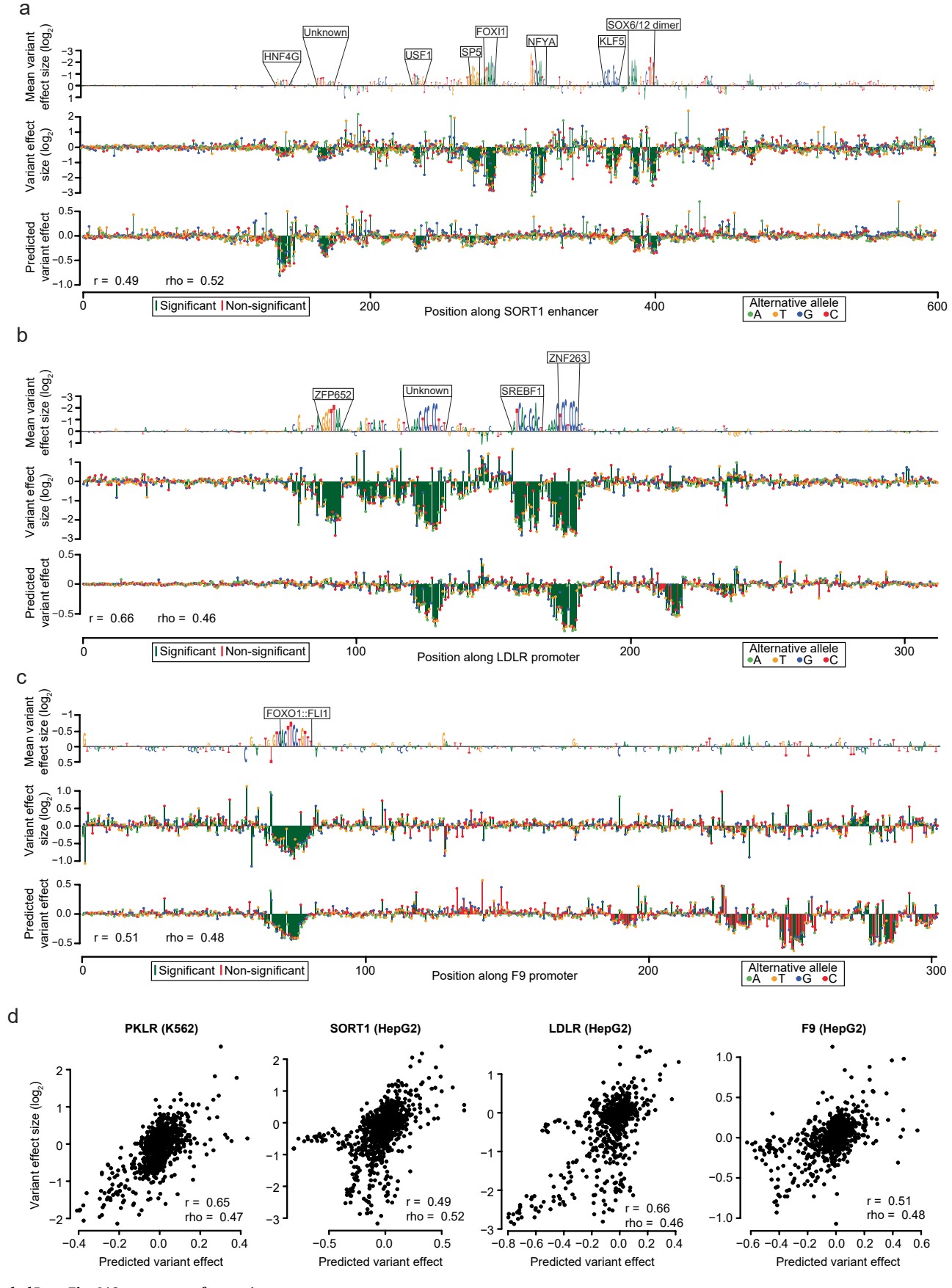

**Extended Data Fig. 9** | See next page for caption.

**Extended Data Fig. 9 | Performance of MPRALegNet in saturation mutagenesis prediction task. a-c**, Saturation mutagenesis data from the *SORT1* enhancer (**a**), *LDLR* promoter (**b**), and *F9* promoter (**c**)[45]. Shown in the top row is the reference sequence scaled to the mean effect size among all alternative mutations, annotated by significant TFBSs that match known motifs[54]. Measured effect sizes of individual variants are displayed in the second row. The bottom row shows MPRALegNet predictions as well as corresponding Pearson (r) and Spearman (rho) correlation values to the observed data. **d**, Scatter plots showing the correlation between predicted genetic variant effects by MPRALegNet and observed variant effects, as detected in a saturation mutagenesis MPRA experiment testing the *PKLR* promoter, *SORT1* enhancer, *LDLR* promoter, and *F9* promoter[45].

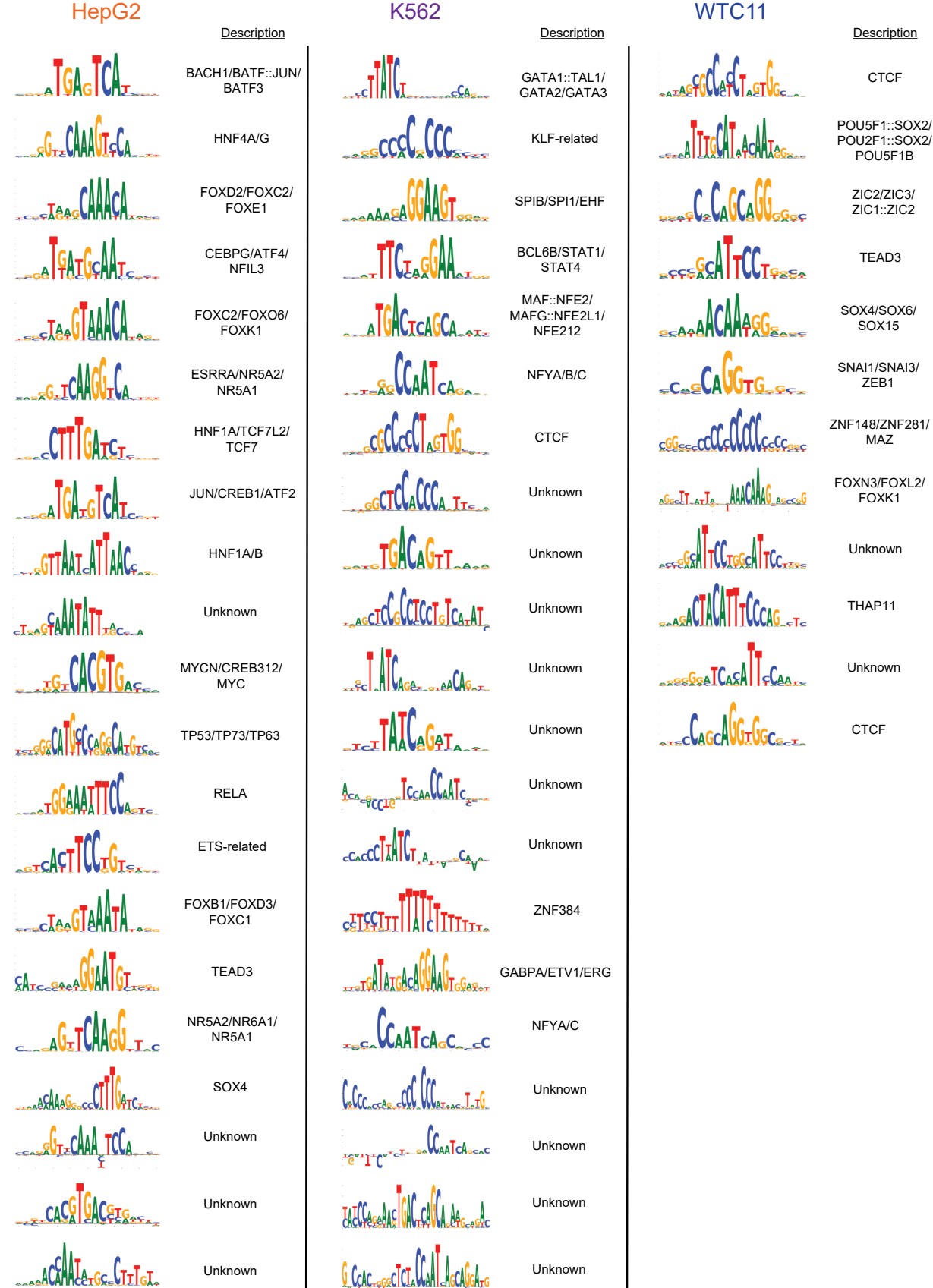

**Extended Data Fig. 10 | Interpretation of cell-type-specific motifs detected by MPRALegNet.** Set of enriched cell-type-specific motifs discovered by TF-MoDISco-lite[38] for each of the three cell types evaluated on the joint MPRA library. Motifs shown are rank-ordered according to their "seqlet"[38] count.

# Reporting Summary

## Statistics

For all statistical analyses, confirm that the following items are present in the figure legend, table legend, main text, or Methods section.

| n/a | Confirmed | |
|---|---|---|
| ☐ | ☒ | The exact sample size (*n*) for each experimental group/condition, given as a discrete number and unit of measurement |
| ☐ | ☒ | A statement on whether measurements were taken from distinct samples or whether the same sample was measured repeatedly |
| ☐ | ☒ | The statistical test(s) used AND whether they are one- or two-sided *Only common tests should be described solely by name; describe more complex techniques in the Methods section.* |
| ☐ | ☒ | A description of all covariates tested |
| ☐ | ☒ | A description of any assumptions or corrections, such as tests of normality and adjustment for multiple comparisons |
| ☐ | ☒ | A full description of the statistical parameters including central tendency (e.g. means) or other basic estimates (e.g. regression coefficient) AND variation (e.g. standard deviation) or associated estimates of uncertainty (e.g. confidence intervals) |
| ☐ | ☒ | For null hypothesis testing, the test statistic (e.g. *F*, *t*, *r*) with confidence intervals, effect sizes, degrees of freedom and *P* value noted *Give P values as exact values whenever suitable.* |
| ☒ | ☐ | For Bayesian analysis, information on the choice of priors and Markov chain Monte Carlo settings |
| ☒ | ☐ | For hierarchical and complex designs, identification of the appropriate level for tests and full reporting of outcomes |
| ☐ | ☒ | Estimates of effect sizes (e.g. Cohen's *d*, Pearson's *r*), indicating how they were calculated |

*Our web collection on statistics for biologists contains articles on many of the points above.*

## Software and code

Policy information about availability of computer code

| Data collection | https://github.com/Altius/hotspot2, https://github.com/ENCODE-DCC/atac-seq-pipeline |
|---|---|
| Data analysis | bcl2fastq v2.20, MPRAflow 1.0, Enformer v1.0, https://github.com/visze/sequence_cnn_models, https://github.com/autosome-ru/human_legnet, TF-MoDISco-lite v2.0.4, Tomtom, STREME, https://github.com/jmschrei/apricot/ |

For manuscripts utilizing custom algorithms or software that are central to the research but not yet described in published literature, software must be made available to editors and reviewers. We strongly encourage code deposition in a community repository (e.g. GitHub). See the Nature Portfolio guidelines for submitting code & software for further information.

## Data

Policy information about availability of data

All manuscripts must include a data availability statement. This statement should provide the following information, where applicable:
- Accession codes, unique identifiers, or web links for publicly available datasets
- A description of any restrictions on data availability
- For clinical datasets or third party data, please ensure that the statement adheres to our policy

Raw sequencing data and processed files generated in this study are available in the ENCODE portal for the pilot libraries (HepG2: ENCSR463IRX; K562: ENCSR460LZI), large-scale libraries (HepG2: ENCSR022GQD; K562: ENCSR382BVV; WTC11: ENCSR244FWB), and joint libraries (HepG2: ENCSR405QCT; K562: ENCSR203UFY; WTC11: ENCSR336MKI). Code to train and interpret MPRAnn and MPRALegNet is available at https://github.com/visze/sequence_cnn_models and https://github.com/autosome-ru/human_legnet. Pretrained models and code have also been deposited in Zenodo (https://zenodo.org/record/8219231). All

summarized processed data is included in supplementary tables.

## Human research participants

Policy information about <u>studies involving human research participants and Sex and Gender in Research.</u>

| Reporting on sex and gender | We did not perform sex- and gender-based analysis in the study, because the data were collected using established cell lines. Gender has no impact on the lentiMPRA results. |
|---|---|
| Population characteristics | Population characteristics are not relevant to this study. |
| Recruitment | Recruitment is not relevant to this study. |
| Ethics oversight | N/A |

Note that full information on the approval of the study protocol must also be provided in the manuscript.

# Field-specific reporting

Please select the one below that is the best fit for your research. If you are not sure, read the appropriate sections before making your selection.

☒ Life sciences  ☐ Behavioural & social sciences  ☐ Ecological, evolutionary & environmental sciences

For a reference copy of the document with all sections, see <u>nature.com/documents/nr-reporting-summary-flat.pdf</u>

# Life sciences study design

All studies must disclose on these points even when the disclosure is negative.

| Sample size | Sample sizes are noted in each of the figures and resulted from the number of elements designed that were successfully measured after using our processing pipeline. |
|---|---|
| Data exclusions | Low quality data excluded from the analysis are clearly indicated in the Methods sections with described filtering procedures |
| Replication | LentiMPRA were done with three biological replicates for each of three cell types (HepG2, K562, WTC11). |
| Randomization | N/A |
| Blinding | Blinding was not possible, as we needed to know the identity of each library for the subsequent analyses. |

# Reporting for specific materials, systems and methods

We require information from authors about some types of materials, experimental systems and methods used in many studies. Here, indicate whether each material, system or method listed is relevant to your study. If you are not sure if a list item applies to your research, read the appropriate section before selecting a response.

## Materials & experimental systems

| n/a | Involved in the study |
|---|---|
| ☒ | ☐ Antibodies |
| ☐ | ☒ Eukaryotic cell lines |
| ☒ | ☐ Palaeontology and archaeology |
| ☒ | ☐ Animals and other organisms |
| ☒ | ☐ Clinical data |
| ☒ | ☐ Dual use research of concern |

## Methods

| n/a | Involved in the study |
|---|---|
| ☒ | ☐ ChIP-seq |
| ☒ | ☐ Flow cytometry |
| ☒ | ☐ MRI-based neuroimaging |

# Eukaryotic cell lines

Policy information about cell lines and Sex and Gender in Research

| | |
|---|---|
| Cell line source(s) | 293T (CRL-3216, ATCC), HepG2 (HB-8065, ATCC), K562 (CCL-243, ATCC), WTC11 (GM25256, Coriell Institute). |
| Authentication | 293T, HepG2, K562 and WTC11 cells were not authenticated. |
| Mycoplasma contamination | 293T, HepG2, K562 and WTC11 cells were not tested for mycoplasma contamination. |
| Commonly misidentified lines (See ICLAC register) | No commonly misidentified cell lines were used in the study. |

