## [Peer Review File · Nature]

Massively parallel characterization of transcriptional regulatory elements

Corresponding Author: Professor Nadav Ahituv

Version 0:

Reviewer comments:

Referee #1

(Remarks to the Author)

In "Massively parallel characterization of transcriptional regulatory elements in three diverse human cell types", Agarwal, Inoue et al present lentiMPRA experiments focused on DNase sites for three human cell types HepG2, K562, and WTC11.

Overall, I see value in this work and the dataset produced, but I also found this work of a relatively more incremental nature than what I generally see published in Nature given the other papers that have been published over the past ~10 years using MPRA technology. The main novelty of this paper is the combination of the scale of the number of sequences and testing based on integrating into the genome, where prior work either tested fewer sequences integrating into the genome or tested a similar or greater number of sequences on plasmids. However, the authors integration approach still has limitations and it wasn't clear the distinction between integrating into the genome and testing on a plasmid would have a substantial impact on the conclusions reached in this paper. My overall impression is this is a more technical distinction that would be appreciated by a more specialized audience.

I have a number of other concerns described in my specific comments below.

Specific Comments/Questions

1. The authors are making claims of being comprehensive in testing functional CRE. For instance, the abstract states 'Collectively, our work provides an exhaustive catalog of functional CREs in three widely used cell types' while only testing a relatively small fraction of genomic sequences (i.e. called DNase peaks in a cell type). They are at same time claiming to achieve what I consider a relatively high validation rate (e.g. abstract states '41.7% to be functional') given the limitations of the assay, not requiring other marks supporting enhancer activity, and only testing for activation. I am however skeptical that both the comprehensive coverage and high validation rate can be true in a meaningful way. I think it is quite likely that many regions that the authors did not test would also show activity in the assay at the level the authors are currently using to define functional CREs. The background distribution was only estimated relative to dinucleotide shuffles. The authors did not include a random sampling of non-DNase peaks in a cell type which would be needed to confirm that there are not a substantial number of non-DNase peaks in the genome achieving the same activity levels to define functional DNase peaks. Also, even for regions tested claiming an exhaustive catalog of functional CREs is an overstatement given the limitations of the assay (e.g. only testing 200bp of sequence).

2. Related to the above, the authors did test a set of non-DNase peaks in K562 from seven loci within 1MB of associated with erythroid lineage genes. However, this creates a bias and limits what can be concluded about non-DNase peaks and is inconsistent with their focus on testing DNase peaks genomewide. In particular, I suspect that the authors selection criteria might be less likely to pick DNase sites in other cell types. DNase sites in other cell types might be more likely to show signal in their assay than sites that are not DNase in any cell type. I found the decision to restrict to these seven loci also odd given the authors emphasis on using a representative subset selection algorithm in the methods.

3. Related to the prior point presumably for the jointly tested libraries they tested many locations that are not DNase in the cell type tested but are DNase in other cell types. Do those sites show activity relative to the dinucleotide shuffled controls? Extended data fig 8 and 9 suggest substantial activity of 'putative enhancers' selected based on one cell type in the other, but it is not clear whether those sites are also a putative enhancer in the other cell type or not.

4. Also related to the above the authors should consider evaluating the validation rate as a function of where the DNase peak falls relative to the DNase peak calling threshold. If the authors still are validating functional elements near the cutoff, that could argue against their set being comprehensive.

5. In some cases, the authors tested loci based on a representative subset location method where it is not uniform the number of loci selected is representing. If the authors reweighted the distribution based on the number of loci each selected loci is representing would that cause any meaningful changes to the output distribution?

6. How did the authors select which sequences to create dinucleotide shuffles for? Are they representative of the dinucleotide content of the sequences tested if not a random sample?

7. The author emphasize that their method can be used for 'hard to transfect cell lines' and are critical of prior work doing similar assays claiming they 'only work in a limited number of established cell types', but this was not compelling in the context of the results they present in the paper. In the paper only present results for HepG2, K562, and WTC11. HepG2 and K562 are relatively easy to transfect and have been used in a number of previous MPRA papers without requiring lentiMPRA. For WTC11 the authors noted transfection efficiency limited the number of sequences they could test, while there are other papers have reported MPRA/STARR-seq experiments for human ESC (e.g. Barak et al, Cell Stem Cell 2018).

8. The authors place emphasis as a main contribution of their work relative to prior MPRA designs is that they are integrating in the genome as opposed to on a plasmid. However, it is unclear that this difference makes a meaningful impact on any conclusions on the paper. The author in some cases use as positive and negative controls sequences that were tested on a plasmid and showed relatively good agreement. Even when integrating in the genome it is the average activity from those integrations may not represent activity in the native loci, which the authors do in effect acknowledge with the results of extended data fig 2e.

Also compared to some prior MPRA/STARR-seq designs the authors are only able to test one 200bp window per regulatory element, while other designs get more information based on capturing multiple different sequence positions per regulatory element.

9. The authors highlight at the end of the abstract as main contribution of the paper "and showcases how large-scale functional measurements can be used to dissect regulatory grammar", but there have been previous approaches also using deep learning and MPRA measurements to attempt to dissect regulatory sequences as the authors do here which was not acknowledged or cited (e.g. Movva et al, PLoS one 2018).

10. I felt the introduction gave a misimpression about the contents of the manuscript. It presented various problems and limitations of some approaches that weren't actually being addressed by the authors work. One example of this was problem of linking regulatory regulatory elements to genes as the author approach does not provide information about target genes of regulatory elements. Another example was DNase I hypersensitivity not differentiating enhancers from promoters as the author assay can show activity for both types of regions. A third example was measurements of the influence of genetic variants on activity. Note while the authors do make predictive models of based on sequence and MPRA data in this paper those have also been made for biochemical assays. The authors are also critical of ChIP-seq in terms of boundaries, but the results in this paper are also limited by only considering one 200bp window per element.

11. Can the author's DNase sites having a repressive role based on the data they or is their assay only powered to detect activation?

12. I would suggest the authors reconsider the terminology 'putative enhancer' to describe any promoter-distal DNase site. DNase is known to be associated with insulators and silencers. Often the 'putative enhancer' terminology is used with additional evidence such as histone modifications or in some cases based on histone marks without necessarily requiring a DNase peak. Also it wasn't explicit until the methods section that this is how 'putative enhancer' was defined for the scale-up experiment, which made the claim of testing all 'putative enhancers' confusing. Additionally in some places in the text the authors dropped the 'putative' word.

13. I would also suggest the author use the DNase peak terminology instead of the CRE terminology where that is a more accurate description of what is actually being tested.

14. In the abstract the authors state "we used lentivirus-based massively parallel reporter assays (lentiMPRAs) to test the regulatory activity of over 680,000 sequences representing a nearly comprehensive set of all annotated CREs among three cell types (HepG2, K562, and WTC11)," even if one assumes annotated CREs=called DNase peaks, I don't feel the statement is true for WTC11 since only a minority of distal DNase peaks were tested.

15. The authors include 30bp adapter sequence as part of the input to their sequence based prediction models. I find that difficult to justify since improvement gains from that information would be artificial and not reflect the ability to predict activity of the actual regulatory sequence of interest.

16. The comparison between using enformer based features and biochemical features is confounded in that the authors are using a different and larger number of features for enformer.

17. Are the authors applying enformer at some of the same loci enformer was trained? If so, some of its features could represent memorization of biochemical features it was trained on than a true reflection of the predictability from sequence.
18. Related to the above points, some of the improvement of sequence predictions over biochemical features might be an artifact of the assay being limited to only test 200bp sequences. Presumably regulatory sequences that need more than 200bp to function will have biochemical markings but not be active in the assay, while the sequence models can make a prediction that is more specific to the actual 200bp being tested.
19. The authors in the main text are making the point that they are using a transformer/enformer architecture. However, they are not taking advantage of likely a key aspect of the transformer architecture in the ability to handle long range sequences as they are only using ~200bp sequences. Is there any advantage to using the enformer/transformer architecture with the extensive N padding relative to prior non-enformer/transformer sequence models focused on shorter sequences?
20. The authors implicate specific TF/motif by applying their convolutional neural network (MPRAnn) followed by ISM followed by TF-MoDISco-lite followed by Tomtom applied to the JASPAR database. However, I felt for implicating specific TF/motifs a simpler and more biologically informative analysis approach could have been to conduct standard TF motif scanning and enrichment analyses. For instance, in some of the figures I see a TF motif shown for two of the cell types but not a third one (e.g. REST extended data fig 6), and it is unclear if that is because the TF/motif is biologically different or the stochasticity of their procedure. Do the TF/motifs implicated by MPRAnn+ISM+TF-MoDISco-lite analysis differ than those that would be implicated by standard motif scanning and enrichment? If they do differ is there additional evidence to support the TF/motif predictions from the MPRAnn+ISM+TF-MoDISco-lite+Tomtom being better?
21. Related to the above, in Extended Data Fig. 4 the authors present de novo motif results based on STREME, while other places (Fig. 4d/5e; Extended Data Fig. 6b/11) they present them based on MPRAnn+ISM+TF-MoDISco-lite. Is there a reason they used one approach in one place and another in the other place? In general, do the authors see an advantage for de novo motif discovery by MPRAnn+ISM+TF-MoDISco-lite than more traditional approaches such as STREME?
22. Extended Fig. 6a at the bottom lists 1,700 trainable parameters and 3,957,751 non-trainable parameters. It wasn't clearly exactly what were all the trainable vs. non-trainable parameters and how the non-trainable parameters set. Also it wasn't described what the 'None' corresponded to in the figure.
23. The methods section on Train MPRAnn could provide information on how the hyper-parameters/architecture were decided upon.
24. 'we next set out to test whether our revised lentiMPRA approach could assay the majority of CREs of any given human cell type, i.e. over 200,000 sequences' – the authors tested more than 200K sequences in an experiment but not over 200K CREs as might be implied by the sentence
25. "This analysis revealed a large suite of TFs whose activity was strongly correlated to the top features, and identified other relevant components of cohesin, including SMC3 and STAG1 (Extended Data Fig. 5a)." – I don't see SMC3 or STAG1 in Extended Data Fig. 5a
26. The section "Cell culture, lentivirus packaging, and titration" mentions HEK293T but that wasn't mentioned elsewhere in the paper. Were experiments also done in HEK293T and if so why aren't they also being shown in the paper?
27. I would suggest adding some type of figure/table associated with the TF expression analysis described in the last sentence of the results.
28. The authors made the point about limitation of sample size in training their previous models, but did not actually quantify how important the increased sample size for training was to the predictive models, which could have been done by downsampling their data points.
29. The authors present results for four regulatory elements saturation mutagenesis data set for which a number of other methods have been evaluated including recently the enformer method though it is not clear if the exact same evaluation procedures are applied. How does MPRAnn compare to existing method that have been previously evaluated with this data? Note it is a limitation compared to existing methods that MPRAnn cannot make predictions for all elements from the dataset.
30. In the introduction the authors state "Previous work has utilized self-transcribing active regulatory region sequencing (STARR-seq) to test a large number of sequences for regulatory activity in human cells^{11–14} – not all of the cited references used STARR-seq. Some used non-STARR-seq MPRA variants. Also note there have been various papers which applying STARR-seq after using a capture step based on ATAC or ChIP which among STARR-seq based papers might be closest to the author's design.
31. In the model training description a held-out fold is mentioned for evaluating model performance, but from my reading of the description that fold could have an influence the selection of the lambda hyperparameter though it will likely be small.
32. Some of the terminology for the categories in the supplementary tables did match the categories given in the main text of

the manuscript.

33. WTC11 data has not been deposited

Referee #2

(Remarks to the Author)

In their manuscript entitled "Massively parallel characterization of transcriptional regulatory elements in three diverse human cell types", Agarwal et al. apply lentiMPRAs (lentivirus-based massively parallel reporter assays) to test the regulatory activity of 680,000 sequences containing candidate cis-regulatory elements (CREs) across three commonly used cell types (HepG2, K562, and WTC11). This is an ambitious analysis of a near comprehensive set of candidate regulatory sequences across three cell types. They report 41.7% of the tested sequences having functional activity, with promoters having significant strand orientation effects and enhancers having weaker orientation effects but increases cell-type specific characteristics. Their lentiMPRA results yielded a sequence-based prediction model of CRE function. Overall, their work provides: (1) an improvement to their previous lentiMPRA protocol, expanding the number of sequences/variants tested in a single experiment from ~14,000 to over 200,000, and (2) an extensive and highly valuable catalog of functional CREs for the field. The manuscript should be worthy of publication in Nature following revisions to replicate the cell-type-specific analyses across isogenic comparisons and add functional validation of top CRS, eGenes and/or networks.

Major Comments:

1. LentiMPRA technology – Although the authors refer to an “Optimization of lentiMPRA technology” in scaling up their methodology, it is unclear precisely what has been changed from their previous report in Nature Protocols. Please detail how optimization was undertaken and precisely how methods have been updated.
 - a. The median barcodes for WTC11 cell lines are below 20 in all replicates (extended fig 3), does using fewer than 20 barcodes increase variation? In the methods, the authors note removal of CREs with fewer than 10 barcodes. Please state the percentage CREs removed and provide actual number of tested CREs.
 - b. What was the power to detect significant transcriptionally active elements with 20-50 median barcodes? Might this have missed active promoters/enhancers – especially cell-type specific effects that could manifest as more subtle shifts in transcription?
 - c. Given the size of the library analyzed and the number of cells required, can the authors assess if three replicates per condition was necessary or sufficient.
2. Cell-type specificity versus donor specificity: The cell types compared are not donor matched, limiting the comparisons that can be made across cell types. HepG2 (human hepatoma 15yo white male), K561 (lymphoblast 53yo female), and WTC11 (pluripotent stem cells, 30-34yo male Japanese) are distinct cell types each derived from different donors, of different ages and sexes, introducing possible transcriptional activity due to genotype.
 - a. For example, in the pilot MPRAs, HepG2 cells revealed 30.6% putative enhancers to be more active than shuffled negative controls, while K562 showed 58.6%; since both libraries were prioritized with DNase peaks, what might cause the two-times activity difference between the two cell lines? Notably, activity differences across cell types diminished in the larger dataset.
 - b. The authors should address inter-donor and sex effects in their limitations. Can they determine what percent variability between cell-type activity may be attributed to genotype-specific (and sex-specific) regulation?
 - c. The authors should include two or more differentiated cell types from the WTC11 hiPSCs, even if only across a subset of predicted cell type specific CREs.
3. Biochemical features predicting regulatory activity - The authors note the bias of the previous regression model towards a strong H3K27ac signal. Does their new model including MPRA signals have feature preferences. Across cell types, do they observe any difference/bias in prediction based on the number of features included? If including more features or datapoints increases prediction, does combining data from all three cell types give a more general and statistically powerful regression model, particularly since their data suggested little cell type-specific promoter activity?
4. Functional validation: It would be informative to test the extent that a catalog of thousands of functional CREs advances our ability to resolve genotype-to-phenotype associations.
 - a. Can the cis- and trans-target genes of these CREs be predicted by incorporating Hi-C and other epigenetic datasets, in order to apply cell-type specific activity-by-contact and activity-by conformation models?
 - b. What percentage of top CREs can be further functionally validated by CRISPRi and/or prime editing? To what extent can their proximal and distal target genes be correctly predicted?
 - c. Likewise, can cell-type specific driver TFs be validated by manipulating their expression in the other cell types (e.g. WTC11) and exploring the impact on cell type identity?
 - d. Disease relevance: Can the authors apply their validated set of CREs and/or their novel sequence-based models to predict CRE function to improve fine-mapping of disease GWAS?

Referee #3/4

(Remarks to the Author)

This paper presents a scaled-up version of lentiMPRA assay, which involves the parallelized insertion of sequence elements of interest into random positions in the genome multiple times, followed by a reporter gene expression readout to assess sequence element function. The authors performed this assay on three cell lines and analyzed the data. The results confirmed that human promoters are more orientation-specific than enhancers. The authors developed a custom neural network, MPRAnn, and built upon a previous neural network, EnformerMPRA, and a lasso regression model based on biochemical features to predict element functional activity. In the final section, the authors tested a common set of regulatory elements in the three cell lines.

The updated lentiMPRA version represents an advancement over its low-throughput predecessor, demonstrating impressive consistency and reproducibility across replicates. However, the biological questions it investigates appear limited in their capacity to reveal novel insights. It remains unclear what unique biological information this assay can offer that has not already been discovered. Moreover, the computational analysis is shallow with minimal insights provided; a similar result could be achieved through standard motif analysis of accessible chromatin sites.

Major issues:

- The experimental method utilized in this study is an interventional experiment, which has the potential to reveal causal relationships. However, due to the insertion of CREs at random positions within the genome, the context within a CRE grammar is not replicated as in native sequences. As a result, the primary question being investigated is how short sequence elements function outside of their original context. This approach essentially measures the functional effect of sequence elements averaged across different random contexts, which should be taken into account when drawing conclusions from the results. It is important to note that this aspect is not explicitly stated in the text. Therefore, it is necessary to consider that the effect size is measured across random contexts when interpreting the results.
- It is unclear what the technical innovation of the revised lentiMPRA is compared to their previous version. Could the authors add a clear explanation of differences and the current contribution?
- Although the loss of sequence context presents challenges, there remain several intriguing biological inquiries to explore. Nevertheless, interpreting the current results is not a straightforward task. For example, the experiment seeks to determine how well enhancers function as promoters by placing them upstream of a minimal promoter in a cross-comparison with promoter elements. The results reveal that enhancers exhibit low activity across all cell lines (as depicted in Figure 1d). The authors describe enhancers as exhibiting a “right skew” distribution, but this is also observed for negative controls, like K562.
- In terms of the orientation dependence results, the difference in directional preference between promoters versus enhancers is not a big effect (albeit statistically significant). However, the language used throughout the text is quite strong. The way in which the results are plotted in Fig 2, is difficult to assess. For instance, the Pearson’s r for K562 FvF and RvR in Fig 2a seems to be under 0.8 on average. But, the correlation between replicate pairs is “0.88-0.96”. Where is this dropoff coming from? I think a clearer description of the analysis could be helpful.
- The pilot experiment shows high replicability but the main CRE characterization results are drawn from the larger scale experiments which have a much lower replicability. The authors should provide a better explanation about these differences. Based on the scatter plot (Fig. 2b), it appears quite symmetric. It appears that the reverse strand preference has just as much preference in many sequence elements as the forward strand. It could be beneficial to color promoters and enhancers using a different coloring scheme. The language about Fig 2c conclusions about TFBS promote transcription unidirectionally is much stronger than the evidence.
- When placing promoters in different orientations, how does this affect promoter competition with the minimal promoter? Fig 2 overall is quite complex to interpret and would benefit from a better description of the takeaways. For instance, it seems that the lentiMPRA assay leads to the same correlation for RNA-seq across cell types for HepG2. Why is this and is there a better way to stratify the results to achieve a more precise conclusion from this analysis (eg. strong vs weak promoters, highly vs lowly expressed genes or housekeeping vs differential cell type specific)?
- The computational modelling provides minimal insights. It falls short of highlighting the benefits of this dataset and the power of deep learning modelling. The main insights are motifs learned by MPRAnn via attribution map clustering with TF-MoDISCo-lite. In principle, this insight could likely be accomplished with standard motif analysis. MPRAnn and other neural networks can learn more complex relationships between motifs, which could potentially reveal deeper insights into cis-regulatory grammars. But that was not done here, even though more thorough in silico interrogation of DNNs is now standard protocol for model interpretation.
- Moreover, the hyperparameter choices in MPRAnn are not standard; softmax hidden activations for conv layers and sigmoid for dense layers. It seems like this model was purposefully designed to be inhibited to make Enformer-MPRA appear better. A simple hyperparameter search would lead to significantly improved performance.
- In terms of the biochemical model, a change in regularization strength in the lasso model can alter the list of salient biochemical features. So modelling itself is only meaningful if generalizable knowledge is extracted from the analysis. But there is minimal effort to validate the computational modelling; it is descriptive with no depth.

- Also, the lentiMPRA elements are embedded randomly in the genome, so it's not quite clear how the biochemical features make sense for this analysis; they correspond to the original location of the sequence element.
- The authors claim Enformer is not used for the motif analysis because of multicollinearity of features. But how about the colinearity of biochemical features used in the biochemical model? Similar to the biochemical model, Enformer can still be used for interpretability compared to MPRAAnn or a better explanation should be given for not using it.
- More critically, there is no attempt to connect the insights (whether motifs or biochemical features) across the 3 models. What is driving the improved predictive performance?

Technical concerns

- Where are the lentivirus' integrating into the genome? Does the location have an effect on readout? What is the variance of the same CRE injected into different loci in the same cell (across the 20-50 elements)? The raw data should be provided (pre-averaging the elements).
- Element specificity score in Fig 5c is a confusing metric, the authors should directly plot distributions. HepG2 seems to have a much higher replicate consistency compared to K562 and WCT112. Comment on this would be helpful.
- Data availability: WCT112 data is missing and K562 is mislabelled -- ENCSR022GQD, according to ENCODE portal, is HepG2. A clearer description of these files and how the dataset was processed up to data splits would be greatly beneficial for reproducibility.
- There is a gap between the major issues in the field and what is addressed in this paper. The presentation should be a bit more clear on what is addressed and those speculations of what could be addressed with this technology should be in discussion. For instance, in the introductory paragraphs, they highlight a major open issue in the field is our poor understanding of disease associated variants. The article does not address this issue directly.
- The authors should deposit all "processed data" for the training, validation, testing, along with the model weights on zenodo. Also a static snapshot of the code.

Minor issues:

- I don't follow the rationale for why WTC112 cells had a cohort of transcriptionally active genes whose promoters were inactive in our MPRA (extended Data fig 4b), suggestive of a shift towards a euchromatic promoter state in this cell line. Could this also be that these promoters are not a sufficient size for certain genes (i.e. not as compact)?
- I don't agree with this statement that "putative enhancers span the range in between both extrema" as a description of Fig 1d. I only agree with the promoters, not the enhancers.
- Table references are not all correct.
- It would be nice to get quantitative values in table form of the performance of each model.
- Can the authors add analysis similar to that in Figure 4 for other elements in the saturation mutagenesis dataset (corresponding to K562)?
- Why are negative controls in Fig 5c for WTC112 significantly different from 0?
- The written presentation is a bit repetitive, especially with the pilot experiment, CRE characterization experiment, and joint library experiment.

Version 1:

Reviewer comments:

Referee #1

(Remarks to the Author)

The authors made revisions that addressed some of my previous comments, but my overall assessment of this manuscript that I gave in my original review has not changed substantially.

The authors toned down their claims in some places about providing an exhaustive catalog of CREs, but I still do not think the authors results are consistent with what they are stating as their overall conclusion:

“In summary, notwithstanding these caveats, our work provides a nearly comprehensive catalog of functional regulatory elements in three established cell lines that are widely used in biological research: HepG2, K562, and iPSCs.”

Considering the authors only tested a small fraction of genomic sequences and extrapolating from the figures they show in response to points 3 and 4, I suspect if they tested more of these sequences the authors would call far more functional CREs in a cell type than they are currently reporting or is generally appreciated.

Given their results are challenging current dogma in terms of validating sites without DNase signal in the cell type, as the authors note in response to point 3, I am questioning why the authors are choosing to relegate this dogma challenging result to a supplementary figure. This also raises questions of whether the result of validating 41.7% of cCREs to be functional in the abstract might not be as meaningful as might be interpreted.

Given the authors are still validating a substantial number of sites with relatively low DNase signal particularly in K562 and HepG2 and there are many more sites below the calling threshold not being tested I think this suggests many additional sites if tested would likely be considered functional based on their procedures. I think these results should be disclosed to the readers and preferably also with a quantitative statistic associated with them. With respect to the authors point about a 5% false discovery rate, they could do a more local FDR estimate based on the DNase signal.

In response to my comment on not applying the approach to hard to transfect cell lines, the authors mention other MPRA papers for hard to transfect samples, but the novelty of this paper is the scaled up-version of the assay which still hasn't been demonstrated in hard to transfect cell lines as far as I understand.

I agree there were technical differences between the author's assay and other approaches that were reported in their previous paper. I am less clear if there is a main result being reported in this paper that was specifically enabled because of one of those differences.

I still think the introduction could give a misimpression of the contents manuscript since it mentions a set of limitations, a number aren't addressed in the manuscript and is not explicit in the introduction on which ones are being addressed in the manuscript.

The authors misinterpreted my suggestion for doing standard TF motif-scanning enrichment analysis. For this I meant using a set of known motifs for motif scanning and enrichment and not performing de novo motif discovery as the authors did in response. A TF motif-scanning enrichment analysis approach would be simpler and allow a more direct comparison across cell types not confounded by the stochasticity of motif discovery procedures. Furthermore, it could give a more quantitative interpretation to the motif results. As currently it is hard to interpret the relative importance of individual motifs among the large set of discovered motifs the authors are presenting.

The authors variant/GWAS analysis is similar to what has been done previously with sequence models trained to predict other biochemical assays. There is a disadvantage here in that the analysis is only provides cell type specific information for a limited number of cell types with their MPRA data compared to what has been done before for some other biochemical assays. Based on the comparison with Enformer results there wasn't a clear advantage even with cell types for which the MPRA data was available (2 loci were better and 2 loci were worse).

The authors are misinterpreting the results presented in the section “MPRALegNet predicts combinatorial TFBS effects” and reaching conclusions about the ‘nature of combinatorial TFBS effects learned by MPRALegNet’ that aren't justified by what they presented. In principle one could have a linear model with many features that could lead to the same types of results the authors are presenting even though a linear model does not learn combinatorial feature interactions. The issue is that the authors are making predictions with many features, but then evaluating the combinatorial effects with only two features at a time. It thus could appear the combinatorial term is making a contribution, but might not be in the context of all the features. Also from a biological perspective the biological results about the pairwise association of features with activity or correlation with the number of copies could have been produced directly from analyzing the observed data so it is not clear the added value of MPRALegNet in this analysis.

The authors added a sentence ‘Considering our activity scores alongside Activity-by-contact (ABC) scores³⁴, we observed that our scores only subtly improved performance in the task of discriminating significant from non-significant enhancer–promoter interactions in these three studies (data not shown).’ – I think the authors should show these results so a reader can directly assess this subtle improvement being claimed.

Referee #2

(Remarks to the Author)

I commend the authors on their thoughtful and detailed response to the feedback of myself and the other reviewers. They now apply a new neural network to improve regulatory element identification, interrogate cooperative and dose-dependent regulatory behavior, and improve fine-mapping of seven different human diseases. Moreover, they cleverly apply existing CRISPRi perturbation studies to validate significant enhancer–promoter interactions identified as well as transcription factors implicated by their study. Additionally, the textual revisions clarify the methods and analyses applied throughout, and also discuss in detail the limitations of the results presented. This manuscript is a technical tour de force and a joy to read! I

am happy to recommend this manuscript for publication at Nature at this time.

Referee #3/4

(Remarks to the Author)

Major concerns:

- The biochemical features predict regulatory activity section doesn't seem to add much to the overall story. It's a statement that yes we can do it, but we don't gain much insights (due to colinearity). The additional experiments using a universal set don't seem to be so modest (Supp Fig 4D) and thus the conclusions drawn from it seem to be reaching. Since no insights can be garnered from the biochemical regression model, I suggest removing this section from the main text and only including the model's performance within the section "Sequence-based models predict regulatory activity with higher precision" as a baseline for comparison with the deep learning models.

- Regarding the section "MPRALegNet predicts combinatorial TFBS effects" details on the observed values should be provided in the main text to better appreciate what is being compared in the figures. Importantly, the combinatorial TFBS effects analysis is based on observational analysis of motifs in the sequences and is thus riddled with confounding factors. The analysis is not measuring dosage or combinatorial features learned by the model because of the other factors that can influence a sequence beyond the motifs that were scanned by FIMO. Moreover, each motif scanned by FIMO has a range of affinities, when treated as a binary value for presence or not, it introduces another confounder to this analysis. This analysis is simply showing prediction agreements for subsets of the sequences, whose correspondence was already shown as part of the test performance. Uncovering TFBS combinatorial rules requires more carefully designed interventional experiments. Without properly controlling for confounding, the statements about combinatorial TFBS effects cannot be made from this analysis.

- Previously the following issue was stated in our review (in major point 1): "This approach essentially measures the functional effect of sequence elements averaged across different random contexts, which should be taken into account when drawing conclusions from the results. It is important to note that this aspect is not explicitly stated in the text. "The point is not whether or not a constant integration site through a safe harbor (single integration site) is better or not compared to random integration. The point is rather that the authors have to take this into account and make sure they acknowledge that the results do not necessarily agree with those expected from sequences in their native contexts (and promoters). The experiment marginalizes out the effects of the genomic context and it is unclear how this is different from episomal assays in essence. This is not sufficiently addressed in the revised text.

Minor concerns:

- The introduction is still misleading. The last paragraph is relevant to the study but the rest of the introduction is a broad overview. It seems like the text could benefit from shrinking the 'review of the field' part (i.e. the first 2 paragraphs of the introduction) and making it clear what limitations the paper is going to address and what it discovers. It would help to guide the reader with the exact goals of the study, e.g. test orientation effects, characterize cell type specific factors in promoters and enhancers (more similar to the text in the last paragraph of the introduction).

- the new addition to paragraph 1 on page 10 seems to indicate that lentiMPRA assay does not capture the cell type specific trans factor effects and essentially measures the promoter activity directly, marginalizing out context effects and not incorporating the trans effects. If this is the case the authors should make it clear that the conclusion that promoters are not cell type specific refers to their intrinsic activity.

- On page 10 paragraph 2 -- a correlation drop from 0.55 to 0.43 does not seem 'modest'. This indicates that if we marginalize promoter context and remove trans effects the transcriptional activity will only have a correlation of 0.43; it does not strongly predict gene expression levels.

- The format of figure 2d is still confusing. Visually it would be easier to understand this as a beeswarm plot (similar to Fig 2a), as it is now the reader has to decipher the pearson correlation values from the size and color of the circles which would be more straightforward to do with a y axis scale. Although I can see a pattern of alternating big and small circles it is unclear how big this difference is (in relation to the other effect sizes discussed in the paper).

- With regards to the promoters having more strand specificity than enhancers. Figure 2c is the only evidence of such a difference and it is based on absolute values. If anything, both enhancers and promoters show comparable strand asymmetry.

- "Both MPRAnn and MPRALegNet underwent optimization procedures to detect hyperparameter and data augmentation settings that improve model performance " Only description of MPRAnn is in the supplementary methods.

- A statistical test to compare enhancer or promoter distributions to negative controls for Fig 1D should be performed.

- For figures 4f and extended data fig 7e it would be helpful to plot out the theoretical values of purely multiplicative effects

based on the single motif effects. Also, the multiplicative pattern should be supported with a linear fit. It seems to be plateauing. A direct comparison with an additive model would also make it more compelling. It seems that all dosage curves saturate, indicating a more sigmoidal trend. Also, the number of sequences for each number of sites should be presented, I presume sequences with 5 of the same motifs occurs much less than the number of sequences with 1 motif.

- there seems to be a typo in figure 5b OR labels (extra decimal place dot)

- in supplementary fig 7 what do the top right corner numbers indicate?

- The term modest seems to range in effect sizes across different examples depending on whether it supports the results or not. There is a lot of work done here and stating the results as is could be more beneficial than to obfuscate it with murky, imprecise language.

Version 2:

Reviewer comments:

Referee #1

(Remarks to the Author)

I first note that since my last review, the paper "Massively parallel characterization of regulatory elements in the developing human cortex" by Deng et al (with an overlapping co-corresponding author of this paper) was published in Science (previously an earlier version was only a pre-print). This in my view compromises much of what was already the relatively modest novelty of this paper for Nature since it also applies lentiMPRA on a large scale and does similar types of analyses (e.g. deep learning sequence analyses).

Previously I raised issue with the authors reaching this conclusion:

"In summary, notwithstanding these caveats, our work provides a nearly comprehensive catalog of functional regulatory elements in three established cell lines that are widely used in biological research: HepG2, K562, and iPSCs"

In response the authors noted they added this sentence to the discussion instead:

'It is important to note that while the number of tested sequences are high, there could be several additional CREs that were missed in our annotations and subsequently in our assays,...'

I do not consider this conclusion to be fundamentally different than the author's previous conclusion. The word 'several' implies that not many CREs were missed, but I still think there are many sequences that would show activity according to their thresholds if actually tested in the assay.

The authors may have misinterpreted my point with respect to the 41.7% statistic in the abstract. My concern with the 41.7% statistic given in the abstract ('...and found 41.7% to be functional.') is that people might interpret 'functional' to have a stronger meaning here than it traditionally has, particularly given the questions around why the authors assay also considers many sequences to be 'functional' that do not have DNase support in the tested cell type. The authors response was pointing out the added sentence about how they may have missed some additional CREs.

As part of the authors response to my previous point about not showing that the technical differences of the authors assay compared to other assays impacted the main results in this paper, the authors respond:

'The major difference is scale, allowing us to test a significantly large number of sequences using this assay'

However, it is not the case that the authors assay was significantly larger in terms of the number sequences tested than all previous applications of plasmid based MPRA/STARR assays.

While the authors made some improvements to the introduction there are still issues with it.

In particular, the authors state:

"For example, DNase I hypersensitivity will not distinguish between spurious and functional binding events of a given TF, nor does it inform the functional identity of a regulatory element (e.g., active vs. repressive), its magnitude of activity, or the gene(s) that it regulates."

and then claim:

"Massively parallel reporter assays (MPRAs) overcome these limitations by testing thousands of sequences/variants for their regulatory activity in a multiplex fashion."

However, it is not the case that MPRA determines the target gene. Also MPRAs do not directly determine spurious vs. functional binding of a given TF directly, as it is testing a CRE sequence as a whole.

In response to my previous point about the authors not actually showing an overall improvement for variant effect applications relative to previous sequence-based models trained on other assay types which are already available in more cell and tissue types, the authors respond about the reduced model complexity of their model relative to Enformer. I did not consider this to be a satisfying response, even if one accepts that model complexity should be a consideration here, since the authors haven't shown that the LegNet architecture or other existing sequence models that have far fewer parameters than Enformer trained on individual chromatin datasets do not achieve comparable performance to MPRALegNet in their evaluations.

With respect to my points and those of Reviewer 3/4 about reaching conclusions about the combinatorial effects of TFs that aren't justified by their analyses, the authors attempted to control for some potential confounders. However, there still exists many potential confounders in terms of additional information MPRALegNet had access to that haven't been controlled for and that could explain the observations instead of true combinatorial effects.

For the residual analysis that the authors added, I am somewhat surprised that conducting the analysis on the residuals had essentially no effect. Have the authors verified that the model they fit is able to get meaningful predictions? In particular, I am concerned the authors appeared to have used the 'log-likelihood values' for TFs for determining residuals, but then switched to a binary representation for the model on the TF values, which could be confounding this analysis.

Also, in the author's response they acknowledge that the point of the analysis is not to show that MPRALegNet was adding value in the biological interpretation to directly analyzing the observed data, which makes me question why they are highlighting this analysis so prominently. From the perspective of validating a methodological contribution, this analysis is also not directly comparing/benchmarking to competing sequence-based prediction models, so the value from that perspective also seems limited.

Referee #3

(Remarks to the Author)

The authors have addressed all of my concerns.

Referee #4

(Remarks to the Author)

The authors have addressed all of my concerns.

Author Rebuttal To Initial Comments:

POINT-BY-POINT RESPONSE TO REVIEWERS

We thank the reviewers and editor for their encouraging comments as well as their valuable constructive comments, which we believe has helped us significantly strengthen our revised manuscript. Please find below our detailed responses to each of the points raised alongside our revised manuscript. Reviewer and editor comments are replicated in full in black, with our inline responses in blue.

Editor comments

Your manuscript, "Massively parallel characterization of transcriptional regulatory elements in three diverse human cell types", has now been seen by 4 referees (referees #3 and 4 co-reviewed and submitted a joint report), whose comments are attached below. While they find your work of potential interest, as do we, they have raised important concerns that in our view need to be addressed before we can consider publication in Nature.

You will see that while the referees appreciate the amount of work and data that are being presented and the value of the resource, they have expressed their disappointment in the relative lack of novel biological insight that this has led to. Adding further analyses that *demonstrate* the power of this resource to gain novel insights would certainly strengthen the case for eventual publication in Nature.

We thank the editor for the encouraging comments and suggestions. To address the 'relative lack of novel biological insight' from our work, we have added the following in the revised version of our article:

1. We showcase how our MPRA results that extensively tiled seven different human disease loci can identify functional regulatory elements targeting these disease-associated genes (**Fig 1a, Extended Data Fig 2**), providing candidate sequences for disease diagnostics and therapeutics.
2. A new neural network, MPRALegNet, that achieves state-of-the-art performance on our dataset (**Fig 4a-d, Extended Data Fig 5**), with clearer distinction between TFBSs learned by classical motif search methods vs neural network interpretation methods (**Extended Data Fig 4, Extended Data Fig 6, Extended Data Fig 10**) allowing enhanced regulatory element identification and characterization.
3. Revised interpretation of this model to deeply interrogate more complex findings of the neural network model, such as the nonlinear/cooperative behavior and dosage-dependence of motifs on activating expression (**Fig 4e-h, Extended Data Fig 7**).
4. We show how our data can be used to select targeted sequences in CRISPRi screens that will have a large effect on gene expression and validate transcription factors engaged in regulation (**Supplementary Fig 3, Extended Data Fig 7a**).
5. Applications of our data and model to GWAS fine-mapping and genetic variant effect prediction and interpretation, specifically applied to numerous disease-associated loci (**Fig1a, Fig 5, Extended Data Fig 2, Extended Data Fig 8, Extended Data Fig 9**).

Referees' comments:

Referee #1

In “Massively parallel characterization of transcriptional regulatory elements in three diverse human cell types”, Agarwal, Inoue et al present lentiMPRA experiments focused on DNase sites for three human cell types HepG2, K562, and WTC11.

Overall, I see value in this work and the dataset produced, but I also found this work of a relatively more incremental nature than what I generally see published in Nature given the other papers that have been published over the past ~10 years using MPRA technology. The main novelty of this paper is the combination of the scale of the number of sequences and testing based on integrating into the genome, where prior work either tested fewer sequences integrating into the genome or tested a similar or greater number of sequences on plasmids. However, the authors integration approach still has limitations and it wasn't clear the distinction between integrating into the genome and testing on a plasmid would have a substantial impact on the conclusions reached in this paper. My overall impression is this is a more technical distinction that would be appreciated by a more specialized audience.

Please see our response to **questions 7 and 8** below regarding differences between plasmid-based and lentiviral assays. To address the perceived incremental nature of the work, we have added the following in the revised version of our article to further strengthen the biological findings:

1. We showcase how our MPRA results that extensively tiled seven different human disease loci can identify functional regulatory elements targeting these disease-associated genes (**Fig 1a, Extended Data Fig 2**), providing candidate sequences for disease diagnostics and therapeutics.
2. A new neural network, MPRALegNet, that achieves state-of-the-art performance on our dataset (**Fig 4a-d, Extended Data Fig 5**), with clearer distinction between TFBSs learned by classical motif search methods vs neural network interpretation methods (**Extended Data Fig 4, Extended Data Fig 6, Extended Data Fig 10**) allowing enhanced regulatory element identification and characterization.
3. Revised interpretation of this model to deeply interrogate more complex findings of the neural network model, such as the nonlinear/cooperative behavior and dosage-dependence of motifs on activating expression (**Fig 4e-h, Extended Data Fig 7**).
4. We show how our data can be used to select targeted sequences in CRISPRi screens that will have a large effect on gene expression and validate transcription factors engaged in regulation (**Supplementary Fig 3, Extended Data Fig 7a**).
5. Applications of our data and model to GWAS fine-mapping and genetic variant effect prediction and interpretation, especially applied to numerous disease-associated loci (**Fig1a, Fig 5, Extended Data Fig 2, Extended Data Fig 8, Extended Data Fig 9**).

I have a number of other concerns described in my specific comments below.

Specific Comments/Questions

1. The authors are making claims of being comprehensive in testing functional CRE. For instance, the abstract states 'Collectively, our work provides an exhaustive catalog of functional CREs in three widely used cell types' while only testing a relatively small fraction of genomic sequences (i.e. called DNase peaks in a cell type). They are at same time claiming to achieve what I consider a relatively high validation rate (e.g. abstract states '41.7% to be functional') given the limitations of the assay, not requiring other marks supporting enhancer activity, and only testing for activation. I am however skeptical that both the comprehensive coverage and high validation rate can be true in a meaningful way. I think it is quite likely that many regions that the authors did not test would also show activity in the assay at the level the authors are currently using to define functional CREs. The background distribution was only estimated relative to dinucleotide shuffles. The authors did not include a random sampling of non-DNase peaks in a cell type which would be needed to confirm that there are not a substantial number of non-DNase peaks in the genome achieving the same activity levels to define functional DNase peaks. Also, even for regions tested claiming an exhaustive catalog of functional CREs is an overstatement given the limitations of the assay (e.g. only testing 200bp of sequence).

We thank the reviewer for this comment and completely agree that assessing whether a sequence is positive in MPRA can be done in various ways. As the reviewer mentions in point #2, we did test non-DNase peaks in K562, but these were only from seven specific loci which could lead to some bias. However, our final library tested 60,000 sequences in three different cell types with over 18,000 sequences selected due to having DNase peaks only in one cell type and could thus be used as non-DNase peaks in the other two cell types (addressed in point #3).

Regarding length, we agree with the reviewer that testing only 200bp of sequence could lead to bias, as we've previously shown that MPRA sequences centered at the same DNase peak but tested at 192, 354 and 678 lengths exhibit length-dependent differences in activity (Klein et al. *Nature Methods* 2020). We reanalyzed this Klein et al. data to estimate the number of shorter (192 bp) fragments that would have exhibited a different MPRA readout had the same element been tested as a long fragment (678 bp), with differences between the long vs short fragments displayed in the histogram below. Differences between a long vs short element were computed as the residual emerging from a robust regression fit between the two groups of data. Using one standard deviation from the mean as a threshold to call differential activity (vertical bars), we found that 10.1% of elements exhibit greater activity in the long fragment, while 3.4% exhibit reduced activity. This suggests that our estimates for functional CREs is better thought of as a lower bound because the false negative rate is ~3x greater than the false positive rate.

Histogram displaying the difference in enhancer activity between elements tested with a long (678 bp) and short (192 bp) genomic context. X-axis represents the residual retrieved after fitting a robust linear model between activity scores derived from each length class. Positive values indicate the relatively greater activity of the element tested in the long context.

In terms of length, as currently oligosynthesis is the cheapest way to obtain a large number of DNA sequences for MPRA libraries, we are limited by the length they can obtain. As the goal of this article was to test a large number of sequences for all three cell types, we had to work with this limitation.

To address this, we added an entire new paragraph to the discussion section detailing these caveats. In addition, we have toned down throughout the article, including abstract, our statement that we tested a comprehensive CRE dataset for all three cell lines. An excerpt of our discussion towards this point is provided here:

‘We note that our work has several limitations. We used a minimal promoter that provides minimal background expression but could limit the ability to detect potential repressive sequences. We did not confirm that transcripts originated from within the minimal promoter, and results might be different if another common promoter had been used. Conclusions about the properties of enhancers are also caveated by the fact that in MPRA the distance between enhancer and promoter are short relative to the endogenous genome. Due to oligo synthesis restrictions, we only tested sequence lengths of 200 bp, centered on DNase hypersensitivity peaks, which could overlook additional sequences that are important for cCRE activity. For example, significant MPRA differences were observed when the same genomic region was tested at different lengths²¹. However, it is worth mentioning that DNA accessibility was shown to provide a strong prediction of activity in a previous HepG2 and K562 MPRA that used 5 bp overlapping 145 bp tiles to test over 15,000 genomic regions, with the tile at the DNase peak center most frequently active²⁷. The sensitivity of DNase peak calling algorithms, which are subjected to sequencing depth and technical bias, could also affect our sequence selection. LentiMPRA tests sequences

outside of their genomic context, which could lead to false positives. In addition, the transgene integrates in numerous different genomic locations which could lead to site-of-integration effects. However, to correct for this, we required at least 10 integrations per sequence, had antirepressors on either side of our integrated lentiMPRA construct, and observed high correlations between technical replicates. We also tested sequences in three different cell lines originating from three different donors, which could lead to *trans* environmental effects. However, we should note that previous lentiMPRAs carried out by our lab in different donors for the same cell or organoid type⁶⁶ or even in cell lines from different species⁶⁷, chimpanzees and humans, show minor *trans* environmental effects. In addition, while we identified numerous functional cCREs, lentiMPRA cannot directly link them to their target gene.'

2. Related to the above, the authors did test a set of non-DNase peaks in K562 from seven loci within 1MB of associated with erythroid lineage genes. However, this creates a bias and limits what can be concluded about non-DNase peaks and is inconsistent with their focus on testing DNase peaks genomewide. In particular, I suspect that the authors' selection criteria might be less likely to pick DNase sites in other cell types. DNase sites in other cell types might be more likely to show signal in their assay than sites that are not DNase in any cell type. I found the decision to restrict to these seven loci also odd given the authors emphasis on using a representative subset selection algorithm in the methods.

We apologize for not being clear on how these seven loci were chosen. These seven regions/sequences were selected by the entire ENCODE consortium to be tested in various assays from several labs as part of the ENCODE consortium. We have added the following text to address this:

'v) 24 positive and 16 negative manually selected controls in loci of interest such as α -globin, β -globin, *GATA1*, and *MYC*³⁰, several of which were chosen by the ENCODE consortium (**Supplementary Table 1**).'

Regarding the point: "DNase sites in other cell types might be more likely to show signal in their assay than sites that are not DNase in any cell type.", please see a more detailed response in point 3 below.

3. Related to the prior point presumably for the jointly tested libraries they tested many locations that are not DNase in the cell type tested but are DNase in other cell types. Do those sites show activity relative to the dinucleotide shuffled controls? Extended data fig 8 and 9 suggest substantial activity of 'putative enhancers' selected based on one cell type in the other, but it is not clear whether those sites are also a putative enhancer in the other cell type or not.

This is a great suggestion! We have carried out this analysis in the revised version and also added a new figure (**Supplementary Fig. 8**), shown below, to indicate how DNase sites that experience a closed chromatin state in a given cell type are active in other cell types. Interestingly, for both HepG2 and K562 cells, we find that DNase peaks with low DNase in the matched cell type, but high DNase activity in other cell types, seem to be responsive by MPRA activity in a manner greater than the signal for shuffled controls. However, this trend did not seem to be the case for WTC11 cells. Overall, these findings are contrary to current dogma that chromatin accessibility is

a prerequisite for CRE activity, and further support the reviewer's previous comments that there could be active CREs which are below the threshold detectable by DNase accessibility.

We have added the following text in the **Results** section to expand upon this point:

'Next, we evaluated the relationship between DNase activity relative to MPRA activity in all three cell lines. We observed that strong DNase signals were not a prerequisite for MPRA activity. For instance, we found that sequences with nearly absent DNase signal in K562 cells, but high DNase signal in both HepG2 and WTC11 cells, still led to modest to high MPRA activity in K562 cells (**Supplementary Fig. 8**). Collectively, our results show that promoters are less cell-type specific, likely due to the presence of TFBSs for housekeeping TFs which can universally activate them. In contrast, potential enhancers show stronger cell-type specificity, in line with their presumed cell-type-specific functions⁵⁵.'

Supplementary Fig. 8. Activity of elements with low DNase signal. In each cell type, we quantified DNase signal and selected the subset of elements with low signal (i.e., having a signal below the threshold indicated by the vertical red line). For this subset of elements, we evaluated DNase signal in the other two cell types and then quantified a smoothed kernel density estimate of MPRA activity in the cell type being examined.

4. Also related to the above the authors should consider evaluating the validation rate as a function of where the DNase peak falls relative to the DNase peak calling threshold. If the authors still are validating functional elements near the cutoff, that could argue against their set being comprehensive.

Using our data, we have now plotted below how the DNase signal (i.e., within each annotated DNase peak in the corresponding cell type) is related to the detection of a validated enhancer (i.e., those exceeding the upper horizontal line, which reflects the 95th percentile of an observed measurement of shuffled controls). We can clearly see that even in DNase peaks possessing the lowest signal intensity, there are elements that are detected to be functional from our assay in all three cell types. However, it is important to note that with respect to the significance thresholds (horizontal lines), there is a 5% false discovery rate in detecting “functional” elements. Although the proportion of elements measured to be functional but also have among the lowest DNase signal is very low, we nevertheless acknowledge the possibility that there are additional functional elements that were simply not annotated as peaks by the ENCODE consortium. We have therefore added the following text to caution that our results are dependent on the sensitivity of DNase peak calling algorithms, which in turn depend on sequencing depth:

‘The sensitivity of DNase peak calling algorithms, which are subjected to sequencing depth and technical bias, could also affect our sequence selection.’

Relationship between DNase signal (x-axis) in the peak measured and element activity (y-axis) as measured by the MPRA for each of three cell types. Also shown are the 5th and 95th percentile intervals for dinucleotide shuffled controls (horizontal lines).

5. In some cases, the authors tested loci based on a representative subset location method where it is not uniform the number of loci selected is representing. If the authors reweighted the distribution based on the number of loci each selected loci is representing, would that cause any meaningful changes to the output distribution?

This comment applies to the set of promoters and putative enhancers of WTC11, of which there are about 4.6x the latter than the former. Because we have a large sample size, we subsampled the distributions to match in size and observed that the subsampling does not change the results. For instance, we see that randomly subsampling the putative enhancer class to match the number of promoters [purple line] nearly perfectly overlaps the distribution of the initial enhancer set [red line]. Thus, the output distribution is not impacted by the relative representation of elements.

Cumulative density function (CDF) plot showing the cumulative fraction of elements with element activity scores at or below the indicated x-axis value. Lines are colored according to the indicated groups in the figure legend.

6. How did the authors select which sequences to create dinucleotide shuffles for? Are they representative of the dinucleotide content of the sequences tested if not a random sample?

We have now added additional text to the **Methods** to be more precise about the provenance of the dinucleotide shuffled sequences:

In the *K562 pilot library* section: 'iv) 250 additional negative controls were chosen by dinucleotide shuffling 250 random potential enhancers possessing 1-5 TFBSs'

In the *HepG2 large-scale library* section: 'We tested 102 positive and 102 negative controls from a prior study²⁸ as well as 175 dinucleotide shuffled negative controls in both orientations. These shuffled controls were derived from shuffling a random subset of 175 DNase peaks.'

In the *K562 large-scale library* section: 'We tested 50 positive and 200 negative controls from a prior MPRA study²⁷ as well as the same 250 dinucleotide shuffled negative controls as tested in the "K562 pilot library".'

In the *WTC11 large-scale library* section: 'We also tested 100 positive and 100 negative controls from a prior study³³ as well as 100 dinucleotide shuffled negative controls, which were derived from shuffling 100 random sequences from our set of 30,121 potential enhancers.'

In the *Joint library tested in HepG2, K562, and WTC11 cells* section: 'We also tested 181 positive and 169 negative HepG2 controls from a prior study²⁸, 50 positive K562 controls from a prior study²⁷, and 300 dinucleotide shuffled negative controls. The shuffled controls originated from selecting 100 shuffled controls from each of the 3 cell types.'

In the plot below, we observe that, for a given cell type, the shuffled control dinucleotide frequencies are relatively well-matched, especially when comparing against genomic_random:

Frequencies for the 16 dinucleotides (computed while considering both the forward and reverse DNA strands), showing the DNase peak frequencies relative to shuffled controlled frequencies, as well as length-matched sequences sampled randomly from the genome (black).

Nevertheless, we can see that the frequencies associated with K562_DNase are modestly different than those from K562_shufcontrols, and slightly more similar to both HepG2_DNase and HepG2_shufcontrols. Our primary use of the shuffled controls in this study was simply to retrieve estimates of the background, 'negative set' distribution of MPRA scores. Insofar as this was our goal, we can utilize our joint library to observe how the 95th and 5th percentiles (which were used to estimate functional hits in the large-scale libraries) differ between the set of K562 and HepG2 shuffled controls. In the plot below, we observed that the 95th and 5th percentiles are similar, and therefore would only have a subtle effect on the estimates of functional enhancers in K562 cells:

Distributions of the shuffled controls derived from each cell type (two violin plots), along with the 5th and 95th quantiles (horizontal lines) when assessed in K562 cells (MPRA score, y-axis).

7. The authors emphasize that their method can be used for 'hard to transfect cell lines' and are critical of prior work doing similar assays claiming they 'only work in a limited number of established cell types', but this was not compelling in the context of the results they present in the paper. In the paper only present results for HepG2, K562, and WTC11. HepG2 and K562 are relatively easy to transfect and have been used in a number of previous MPRA papers without requiring lentiMPRA. For WTC11 the authors noted transfection efficiency limited the number of sequences they could test, while there are other papers that have reported MPRA/STARR-seq experiments for human ESC (e.g. Barak et al, *Cell Stem Cell* 2018).

We completely agree with the reviewer that while one of the advantages of a lentivirus-based approach for MPRA is to carry out this assay in hard to transfect cells, this was not an advantage in the cells that we tested in this article. Because this project is part of the ENCODE project we were limited to testing cells that were chosen by the consortium. We do have several other MPRA publications from our lab that have used this MPRA method in neurons, primary cells and organoids for example, which would be much harder to transfect. In the text we mention this is an advantage for the method, albeit not shown in this article:

'To address these and other limitations of episomal MPRA, we previously developed a lentivirus-based MPRA (lentiMPRA) that enables reproducibility and multiplexability, the ability to carry out these assays in 'hard to transfect cell lines', and provides an 'in genome' readout. lentiMPRA results were also shown to be more strongly correlated with ENCODE annotations and sequence-based models¹⁹, and an MPRA meta-analysis showed that lentiMPRA provides higher cell-type specificity predictions versus episomal MPRA²⁰.'

And later we revised to:

'Here, we applied an optimized lentiMPRA method and confirmed the reproducibility and reliability of this technology to test over 200,000 sequences in a single experiment, which covers the majority of cCREs of any given human cell type²⁴. We utilized the high-throughput capability of this method to substantially expand MPRA data in three major ENCODE cell types, human hepatocytes (HepG2), lymphoblasts (K562), and induced pluripotent stem cells (iPSCs; WTC11), to examine the relative orientation dependence of promoters and enhancers.'

8. The authors place emphasis as a main contribution of their work relative to prior MPRA designs is that they are integrating in the genome as opposed to on a plasmid. However, it is unclear that this difference makes a meaningful impact on any conclusions on the paper. The author in some cases used positive and negative controls sequences that were tested on a plasmid and showed relatively good agreement. Even when integrating in the genome it is the average activity from those integrations may not represent activity in the native loci, which the authors do in effect acknowledge with the results of extended data fig 2e.

In a previous publication from our labs (Inoue et al. *Genome Research* 2017), we tested the same library that integrates into the genome (standard lentivirus) and one that does not integrate into

the genome (non-integrating lentivirus), showing that the integrated version provides more reproducible and more strongly predictable results both from ENCODE annotations and sequence-based models. In addition, in an MPRA meta-analysis, 'in genome' MPRA was shown to provide higher cell-type specificity predictions versus episomal-based MPRA. We mention this in the introduction:

'provides an 'in genome' readout. lentiMPRA results were also shown to be more strongly correlated with ENCODE annotations and sequence-based models¹⁹, and an MPRA meta-analysis showed that lentiMPRA provides higher cell-type specificity predictions versus episomal MPRA²⁰.'

In addition, we tested nine different MPRA technologies, including both the one used here (lentiMPRA) and various episomal ones (Klein et al. *Nature Methods* 2020). While we observed good correlations between these different methods, in particular standard episomal, STARR-seq and lentiMPRA (Figure 2b in that article) we did observe several differences. For example, for STARR-seq, we observed that proteins involved in mRNA stability or splicing factors have a large effect on activity, likely due to the mRNA of the enhancer serving as the barcode. Comparison between lentiMPRA and episomal MPRA found factors like FOXP1, a pioneer transcription factor, to be associated with lentiMPRA, fitting with it being a 'within the genome' readout. To address this, we have added and revised the following text in the introduction:

'Furthermore, comparison of lentiMPRA to eight other MPRA designs using the same library found strong correlations with episomal MPRA and STARR-seq, but also several differences, including pioneer factors (FOXP1, RFX5) being more predictive of regulatory activity²¹. However, one previous limitation of lentiMPRA was the number of sequences/variants that could be tested in a single experiment (*i.e.*, up to 14,000)^{22,23}.'

Also compared to some prior MPRA/STARR-seq designs the authors are only able to test one 200bp window per regulatory element, while other designs get more information based on capturing multiple different sequence positions per regulatory element.

There is always a trade off with any MPRA/STARR-seq design as mentioned above. For this article, as the goal was to test numerous candidate regulatory sequences for three different cell lines, we decided to focus on different genomic regions instead of tiling specific regions to increase our breadth of tested sequences, which we clarify in our revised introduction. In addition, we should note that previous work that tiled 145bp oligos across H3K27ac dips with 5bp overlaps in HepG2 and K562, termed Sharpr-MPRA (Ernst et al *Nature Biotechnology* 2016), showed that DNase sites are a strong predictive feature of regulatory activity for these tiles. To address this, as mentioned above, we added a new caveat paragraph to the discussion section, that mentions the following:

'Due to oligo synthesis restrictions, we only tested sequence lengths of 200 bp, centered on DNase hypersensitivity peaks, which could overlook additional sequences that are important for cCRE activity. For example, significant MPRA differences were observed when the same genomic region was tested at different lengths²¹. However, it is worth mentioning that DNA accessibility was shown to provide a strong prediction of activity in a previous HepG2 and K562 MPRA that used 5 bp overlapping 145 bp tiles to test over 15,000 genomic regions, with the tile at the DNase peak center most frequently active²⁷.'

9. The authors highlight at the end of the abstract as main contribution of the paper “and showcases how large-scale functional measurements can be used to dissect regulatory grammar”, but there have been previous approaches also using deep learning and MPRA measurements to attempt to dissect regulatory sequences as the authors do here which was not acknowledged or cited (e.g. Movva et al, PloS one 2018).

Thank you for this recommendation. We have now cited Movva et al, PloS One 2018 as further justification for our deep learning approach. We have also added de Almeida et al, *Nature Genetics* 2022 as a further usage of deep learning tools to predict MPRA data.

10. I felt the introduction gave a misimpression about the contents of the manuscript. It presented various problems and limitations of some approaches that weren't actually being addressed by the authors work. One example of this was the problem of linking regulatory regulatory elements to genes as the author approach does not provide information about target genes of regulatory elements. Another example was DNase I hypersensitivity not differentiating enhancers from promoters as the author assay can show activity for both types of regions. A third example was measurements of the influence of genetic variants on activity. Note while the authors do make predictive models of based on sequence and MPRA data in this paper those have also been made for biochemical assays. The authors are also critical of ChIP-seq in terms of boundaries, but the results in this paper are also limited by only considering one 200bp window per element.

We apologize if the introduction provided a misimpression of the contents of the manuscript. We should note that the first paragraph deals with general regulatory element challenges to allow the reader to understand the difficulties the field faces in general (including linking elements to target genes), while the second provides challenges relevant to this article. We have gone through the introduction carefully, taking all these comments in mind, and significantly revised it, in particular relating to the challenges our work is trying to address. In addition, we have added an entire paragraph to the discussion section listing the limitations of our work, including several of these comments.

11. Can the author's DNase sites having a repressive role based on the data they or is their assay only powered to detect activation?

Relative to other studies which use a stronger promoter, our experimental scheme uses a weak promoter that is more optimized for active element detection, which limits its ability to identify silencing elements. While we observe in the figures presented in **point 4** that there are a number of elements in each cell type that seem to have expression levels below the 5th percentile of the shuffled controls, the hit rate in all three cell types is about the same or less than the false discovery rate itself [*for promoters* -- HepG2: 333 of 20,816 (1.6%); K562: 2,153 of 29,376 (7.3%); WTC11: 351 of 9,964 (3.5%); *for enhancers* -- HepG2: 1,541 of 118,433 (1.3%); K562: 3,634 of 169,260 (2.1%); WTC11: 1,380 of 45,942 (3.0%); 5% FDR]. Thus, we do not find sufficient statistical evidence to classify these elements as silencing elements. We added the following text to the discussion section:

'We used a minimal promoter that provides minimal background expression but could limit the ability to detect potential repressive sequences. We did not confirm that transcripts originated from within the minimal promoter, and results might be different if another common promoter had been used.'

12. I would suggest the authors reconsider the terminology 'putative enhancer' to describe any promoter-distal DNase site. DNase is known to be associated with insulators and silencers. Often the 'putative enhancer' terminology is used with additional evidence such as histone modifications or in some cases based on histone marks without necessarily requiring a DNase peak. Also it wasn't explicit until the methods section that this is how 'putative enhancer' was defined for the scale-up experiment, which made the claim of testing all 'putative enhancers' confusing. Additionally in some places in the text the authors dropped the 'putative' word.

We completely agree with the reviewer and apologize for this oversight. We have gone carefully through the article and revised accordingly, changing where relevant to 'DNase peak', candidate *cis*-regulatory element ('cCRE') or 'potential enhancer' and defining them in first mention in the text.

13. I would also suggest the author use the DNase peak terminology instead of the CRE terminology where that is a more accurate description of what is actually being tested.

As mentioned above, we changed this where relevant. However, we should note that having a DNase peak suggests that a sequence is a candidate *cis*-regulatory element (cCRE) and the term cCRE is relevant to all types of regulatory elements. Also, as this article is part of the ENCODE4 consortium and this term was adopted by it to describe these sequences (see for example: <https://screen.encodeproject.org/>) we tried to stick with the same terminology. To clarify this and also justify our use of DNase accessibility in choosing candidate sequences, we added the following sentence in the first results section:

'As DNase accessibility, centered on the midpoint of a peak, has been shown to be a good predictor of regulatory elements²⁶ and MPRA activity^{20,27}, we used it as our main selection criteria for cCREs.'

14. In the abstract the authors state "we used lentivirus-based massively parallel reporter assays (lentiMPRAs) to test the regulatory activity of over 680,000 sequences representing a nearly comprehensive set of all annotated CREs among three cell types (HepG2, K562, and WTC11)," even if one assumes annotated CREs=called DNase peaks, I don't feel the statement is true for WTC11 since only a minority of distal DNase peaks were tested.

We have now revised this sentence to read 'representing an extensive set of annotated cCREs among three cell types (HepG2, K562, and WTC11)'

15. The authors include 30bp adapter sequence as part of the input to their sequence based prediction models. I find that difficult to justify since improvement gains from that information

would be artificial and not reflect the ability to predict activity of the actual regulatory sequence of interest.

We confirm that training without adapters modestly decreases performance (Pearson correlation of 0.74 without adapters and 0.76 with adapters in HepG2 cells). Our main motivation in including the 30bp adaptor was to account for “edge effects”, i.e., TF motifs that occur at the edge of the adaptor. It is crucial to note that this training scheme does not impact the deployment of the neural network in “prediction/inference” mode. During the latter, we can simply replace the adaptor with the native genomic context of the element being predicted, so that the motif(s) at the edges of the element can be considered. We agree that this use of the model in prediction mode was not clear in the initial draft, and have now added the following text to the Methods section to better instruct readers about how to use the model:

‘Prediction using MPRALegNet. To generate predictions on an arbitrary sequence, we recommend generating predictions using all 90 pretrained models (considering test-time sequence augmentations such as orientation and shifting for extra precision), and then averaging the predictions to achieve the final prediction. We recommend replacing the fixed 15 bp adaptors with the surrounding natural genomic sequence context whenever available, to reduce the chances that artifactual motifs occurring at the adaptor-sequence boundaries may bias the results.’

16. The comparison between using enformer based features and biochemical features is confounded in that the authors are using a different and larger number of features for Enformer.

We address this point in the next comment.

17. Are the authors applying enformer at some of the same loci enformer was trained? If so, some of its features could represent memorization of biochemical features it was trained on than a true reflection of the predictability from sequence.

To address **point 16, 17, and 19**, we added the following lines to the results section: “Although we include EnformerMPRA and SeiMPRA for comparison purposes, we caution that they may exhibit inflated performance because i) they have an >8-fold larger feature set than the biochemical models, and were trained on additional cell types and biochemical marks, and ii) having been trained on nearly the entire genome, they had the opportunity to have observed biochemical marks associated with elements in the test set.” Hopefully this text clarifies that our goal here was merely to benchmark Enformer’s performance rather than promote it as an ideal model. We also incorporated a new model, MPRALegNet, to the revised manuscript which does not suffer from these limitations, yet performs competitively with Enformer. The focus of the remainder of our revised article uses MPRALegNet.

18. Related to the above points, some of the improvement of sequence predictions over biochemical features might be an artifact of the assay being limited to only test 200bp sequences. Presumably regulatory sequences that need more than 200bp to function will have biochemical

markings but not be active in the assay, while the sequence models can make a prediction that is more specific to the actual 200bp being tested.

We fully agree with this assessment, and next to the new lines added for **comments 16-17**, we also add the following text:

'Moreover, sequence-based models likely performed better than biochemical models because they have access to the precise 200bp sequence being tested, whereas biochemical signals lack this degree of spatial resolution.'

We should note that we don't necessarily view this as an artifact so much as simply an inherent limitation of the biochemical approach.

19. The authors in the main text are making the point that they are using a transformer/enformer architecture. However, they are not taking advantage of likely a key aspect of the transformer architecture in the ability to handle long range sequences as they are only using ~200bp sequences. Is there any advantage to using the enformer/transformer architecture with the extensive N padding relative to prior non-enformer/transformer sequence models focused on shorter sequences?

We have now benchmarked Sei (Chen et al, *Nature Genetics* 2022) alongside Enformer. Sei is a fully convolutional architecture that uses a local window of 4,096 bp to predict 21,907 biochemical marks derived from diverse cell types. Sei's performance was nearly identical but slightly worse than that of Enformer, as seen in our revised **Fig4b** below. A simple explanation for this result would be that the transformer layers of Enformer have the opportunity to account for and subsequently adjust for the contribution of distal elements towards a local signal, and thus better decompose how both local and distal sequences impact a signal, while Sei is unable to account for this.

Fig. 4b. Violin plots showing the performances of the trained sequence-based models on each of the ten cross-validation folds of held-out data, relative to the corresponding

performances from our biochemical lasso regression models. An improvement relative to another model was evaluated with a one-sided, paired t-test.

20. The authors implicate specific TF/motif by applying their convolutional neural network (MPRAnn) followed by ISM followed by TF-MoDISco-lite followed by Tomtom applied to the JASPAR database. However, I felt for implicating specific TF/motifs a simpler and more biologically informative analysis approach could have been to conduct standard TF motif scanning and enrichment analyses. For instance, in some of the figures I see a TF motif shown for two of the cell types but not a third one (e.g. REST extended data fig 6), and it is unclear if that is because the TF/motif is biologically different or the stochasticity of their procedure. Do the TF/motifs implicated by MPRAnn+ISM+TF-MoDISco-lite analysis differ than those that would be implicated by standard motif scanning and enrichment? If they do differ is there additional evidence to support the TF/motif predictions from the MPRAnn+ISM+TF-MoDISco-lite+Tomtom being better?

STREME is a toolkit for traditional *de novo* motif detection and enrichment analysis. Please see our response below in **point 21** for a further discussion of how we expanded the STREME analysis to give a comprehensive portrait of what traditional motif tools were able to discover for both promoters and enhancers. STREME was unable to discover the REST motif in any of the cell types; in contrast, the inability for TF-MoDISco-lite to detect the REST motif in ESCs could be either a biological result (i.e., REST is not as functionally active in ESCs) or a technical one (i.e., REST motifs, despite being functional in ESCs, are rare enough that their usage is indistinguishable from chance expectation in achieving statistical significance).

21. Related to the above, in Extended Data Fig. 4 the authors present *de novo* motif results based on STREME, while other places (Fig. 4d/5e; Extended Data Fig. 6b/11) they present them based on MPRAnn+ISM+TF-MoDISco-lite. Is there a reason they used one approach in one place and another in the other place? In general, do the authors see an advantage for *de novo* motif discovery by MPRAnn+ISM+TF-MoDISco-lite than more traditional approaches such as STREME?

We apologize that there could be some confusion about the use of both tools, as both tools strive to identify motifs. However, it is important to note that STREME and TF-MoDISco-lite are complementary approaches to one another because their approach to motif discovery is distinct. STREME performs motif enrichment analysis at the very extremes of the expression continuum (i.e., evaluating motifs enriched in top-ranked vs bottom-ranked elements), resulting in the possible oversight of motifs that are activating or repressive but perhaps more modestly so in the intermediate activity range. On the other hand, TF-MoDISco-lite would be more adept at finding common motifs that influence expression along the entire element activity continuum, with the caveat that infrequent motifs or motifs only at the extremes of the distributions might be overlooked if there are insufficient instances of such motifs. Given these methodological differences, we do not consider one technique “better” or “worse” than the other *per se*, but rather identifying motifs with different interpretations. We have added **Extended Data Fig. 4a-b** to comprehensively indicate the motifs that STREME discovers in all three cell types, for both promoters (**a**) and enhancers (**b**), and for both activating and repressive motifs. Comparing this

Figure to **Extended Data Fig. 6**, it is apparent that while many motifs are common to both procedures, each technique also identifies a subset of motifs that are missed by the other. Although each analysis has its limitations, taken together a reader can assess the totality of identifiable motifs detected through our multiple approaches. We have added this sentence in the main text in the MoDISco section to better articulate our view:

'Overall, a large suite of motifs, including several unknown TFBSs, were discovered which were overlooked by a classic motif enrichment analysis (**Extended Data Fig. 4**), supporting the TF-MoDISco approach as being complementary.'

22. Extended Fig. 6a at the bottom lists 1,700 trainable parameters and 3,957,751 non-trainable parameters. It wasn't clearly exactly what were all the trainable vs. non-trainable parameters and how the non-trainable parameters set. Also it wasn't described what the 'None' corresponded to in the figure.

Thank you for the comment. Unfortunately, we made a transcription error and mixed up trainable and non-trainable parameters when generating the figure. It should be 3,957,751 trainable and 1,700 non-trainable parameters. We updated the legend to make the two points clear:

'None' refers to the batch size used during model training.'

23. The methods section on Train MPRAnn could provide information on how the hyperparameters/architecture were decided upon.

We have now added a section entitled "MPRAnn hyperparameter optimization" in the **Supplemental Methods** section to describe the hyperparameter optimization strategy in detail. Briefly, MPRAnn's architecture and hyperparameters were optimized on open chromatin data (DNase-seq) and used without further optimization on the MPRA data. MPRAnn was initially intended primarily as a means to inspect motifs learned, rather than achieve optimal performance. In our revision, we now present a new model, MPRALegNet, whose goal is to optimize predictive performance. Justifications for the architectural design are described in the "*Training MPRALegNet*" **Methods** section. We also more meticulously evaluated how different data augmentations and architectural decisions influence performance in our new supplementary figure:

Extended Data Fig. 5a. Violin plots showing the performances of different variations of MPRALegNet on each of the ten cross-validation folds of held-out data, for different types of augmentations. “-” and “+” indicate removal or usage, respectively, of the following augmentations: i) “test-time”, whereby the mean prediction is computed for various augmentations of the test sequence; ii) “shift”, whereby a sequence was randomly shifted by 0 to +21 bp (i.e., closer to the TSS); iii) “RevComp”, whereby a sequence was randomly reverse complemented; iv) “Orientation”, whereby measured element activity scores were considered for each orientation tested, instead of the mean across both orientations; and v) “5th channel”, whereby a 5th channel was considered alongside the one-hot encoded sequence (i.e., the first 4 channels) to indicate the sequence’s orientation.

24. ‘we next set out to test whether our revised lentiMPRA approach could assay the majority of CREs of any given human cell type, i.e. over 200,000 sequences’ – the authors tested more than 200K sequences in an experiment but not over 200K CREs as might be implied by the sentence

We revised this sentence to read:

‘With our pilot libraries showing reproducible and robust results, we next set out to test whether our revised lentiMPRA approach could measure over 200,000 sequences in a single experiment, including the majority of cCREs of any given human cell type.’

25. “This analysis revealed a large suite of TFs whose activity was strongly correlated to the top features, and identified other relevant components of cohesin, including SMC3 and STAG1 (Extended Data Fig. 5a).” – I don’t see SMC3 or STAG1 in Extended Data Fig. 5a

We believe the reviewer may have accidentally missed SMC3/STAG1 in the original figure -- they are both shown on the very bottom left of the figure, reproduced below for convenience:

26. The section “Cell culture, lentivirus packaging, and titration” mentions HEK293T but that wasn’t mentioned elsewhere in the paper. Were experiments also done in HEK293T and if so why aren’t they also being shown in the paper?

HEK293T cells are used for the purpose of lentivirus packaging. Once packaged, the lentivirus libraries were harvested from HEK293T cells, and infected into HepG2, K562, or WTC11 cells for the MPRA experiments. To clarify, we modified the text to read:

‘Lentivirus packaging was performed using HEK293T (ATCC, CRL-3216), as previously described with modifications²⁵. Briefly,...’

‘The transfected cells were cultured for 2 days to complete lentivirus packaging. The lentivirus libraries in the culture media were separated from the HEK293T cells and...’

27. I would suggest adding some type of figure/table associated with the TF expression analysis described in the last sentence of the results.

We have now added the following **Supplementary Fig 10** to the manuscript, indicating the expression levels of the described TFs in the three cell types examined:

Supplementary Fig. 10. Gene expression levels of TFs in each of the three cell types, for the set of TFs predicted to be highly active in each cell type based upon motif enrichment analysis.

28. The authors made the point about limitation of sample size in training their previous models, but did not actually quantify how important the increased sample size for training was to the predictive models, which could have been done by downsampling their data points.

We have now incorporated the following downsampling analysis below. The results confirm that performance can substantially improve as the size of the training set grows.

Extended Data Fig. 5c. Impact of the size of the training set on model performance. Data from each cell type was downsampled to every 10th percentile (i.e., from 10 to 100%). Error bars represent the standard deviation of the Pearson correlations across the 10 held-out folds of data.

29. The authors present results for four regulatory elements saturation mutagenesis data set for which a number of other methods have been evaluated including recently the enformer method though it is not clear if the exact same evaluation procedures are applied. How does MPRAnn compare to existing method that have been previously evaluated with this data? Note it is a limitation compared to existing methods that MPRAnn cannot make predictions for all elements from the dataset.

We applied the same evaluation procedure as that previously published in the Enformer paper, using the entire saturation mutagenesis data as a test set (Avsec et al, *Nature Methods* 2021). We have added the following line to address both comments:

‘These results were comparable to those from Enformer, whose cell-type matched models achieved correlations of 0.63 for *SORT1*, 0.83 for *PKLR*, 0.62 for *LDLR*, and 0.28 for *F9* for the same task⁴⁴. Combined, our results show how our models can be used for the prediction of regulatory variant effects.’

30. In the introduction the authors state “Previous work has utilized self-transcribing active regulatory region sequencing (STARR-seq) to test a large number of sequences for regulatory activity in human cells^{11–14} – not all of the cited references used STARR-seq. Some used non-STARR-seq MPRA variants. Also note there have been various papers which applying STARR-seq after using a capture step based on ATAC or CHIP which among STARR-seq based papers might be closest to the author’s design.

We revised this text to read:

‘Previous work has utilized MPRA or the self-transcribing active regulatory region sequencing (STARR-seq) to test a large number of sequences for regulatory activity in human cells^{11–14}. This

includes, for example, the testing of various TF binding sequences (TFBSs) in 49 bp chunks, 500 bp human genome random fragments, 170 bp synthetic random sequences, and 150 bp synthetic enhancer–promoter combinations in human colon carcinoma cells (GP5d)¹⁴. Another example involved the deployment of whole-genome STARR-seq in human prostate cancer cells (LNCaP)¹². In addition, utilizing sequence capture, ChIP-seq, or ATAC-seq, numerous cCREs can be captured, cloned, and tested via MPRA or STARR-seq^{15–18}.’

31. In the model training description a held-out fold is mentioned for evaluating model performance, but from my reading of the description that fold could have an influence the selection of the lambda hyperparameter though it will likely be small.

Our cross-validation (CV) process derives the lambda hyperparameter from a separate model of the full data, where lambda.min was identified using the default CV process of the underlying R package. Independent of this model on the full-data, 10 CV bins are created. Using 9 out of 10 bins, new cv.glmnet models are fit (not using the 1 remaining fold at all). Then prediction is done for the remaining fold using these respective models and the lambda.min value of the full model is provided. So while the lambda hyperparameter is derived from the complete data, the coefficients that are included/excluded because of this lambda is determined in the individual submodels. Thus, the reviewer is right, selecting lambda uses the complete data, but since this regularization parameter is used independently on each CV submodel, this is a neglectable influence and does not allow any label leakage.

32. Some of the terminology for the categories in the supplementary tables did not match the categories given in the main text of the manuscript.

Thank you for flagging this error. We have now gone through and carefully replaced all discrepancies such that the categories match with those presented in the manuscript.

33. WTC11 data has not been deposited

Our data was previously deposited to support the ENCODE consortium across a time span of the last five years, meaning that data was deposited incrementally and with different pipelines and analytic strategies. We have now archived all older data and updated the raw and processed data, uniformly processed with the identical computational pipeline. We have amended the text to now read:

‘Raw sequencing data and processed files generated in this study are available in the ENCODE portal for the pilot libraries (HepG2: ENCSR463IRX; K562: ENCSR460LZI), large-scale libraries (HepG2: ENCSR022GQD; K562: ENCSR382BVV; WTC11: ENCSR244FWB), and joint libraries (HepG2: ENCSR405QCT; K562: ENCSR203UFY; WTC11: ENCSR336MKI). Code to train and interpret MPRA_{Ann} and MPRA_{LegNet} is available at https://github.com/visze/sequence_cnn_models and https://github.com/autosome-ru/human_legnet. Pretrained models and code have also been deposited in Zenodo (<https://zenodo.org/record/8219231>).’

Referee #2

In their manuscript entitled "Massively parallel characterization of transcriptional regulatory elements in three diverse human cell types", Agarwal et al. apply lentiMPRAs (lentivirus-based massively parallel reporter assays) to test the regulatory activity of 680,000 sequences containing candidate cis-regulatory elements (CREs) across three commonly used cell types (HepG2, K562, and WTC11). This is an ambitious analysis of a near comprehensive set of candidate regulatory sequences across three cell types. They report 41.7% of the tested sequences having functional activity, with promoters having significant strand orientation effects and enhancers having weaker orientation effects but increases cell-type specific characteristics. Their lentiMPRA results yielded a sequence-based prediction model of CRE function. Overall, their work provides: (1) an improvement to their previous lentiMPRA protocol, expanding the number of sequences/variants tested in a single experiment from ~14,000 to over 200,000, and (2) an extensive and highly valuable catalog of functional CREs for the field. The manuscript should be worthy of publication in *Nature* following revisions to replicate the cell-type-specific analyses across isogenic comparisons and add functional validation of top CRS, eGenes and/or networks.

We thank the reviewer for the positive assessment of the article and constructive comments.

Major Comments:

1. LentiMPRA technology – Although the authors refer to an “Optimization of lentiMPRA technology” in scaling up their methodology, it is unclear precisely what has been changed from their previous report in *Nature Protocols*. Please detail how optimization was undertaken and precisely how methods have been updated.

We thank the reviewer for pointing this out. We developed this protocol and the work presented here as part of our five year long ENCODE project and this publication is part of the ENCODE4 package. Due to that, we were limited in publication timing as part of the package and did not want to limit the adaptability of this technology by other labs due to this timing, thus publishing a detailed *Nature Protocols* on this methodology so that it can be adopted by other labs. We thus carefully went through the article and revised the text to clarify this.

a. The median barcodes for WTC11 cell lines are below 20 in all replicates (extended fig 3), does using fewer than 20 barcodes increase variation? In the methods, the authors note removal of CREs with fewer than 10 barcodes. Please state the percentage CREs removed and provide actual number of tested CREs.

b. What was the power to detect significant transcriptionally active elements with 20-50 median barcodes? Might this have missed active promoters/enhancers – especially cell-type specific effects that could manifest as more subtle shifts in transcription?

c. Given the size of the library analyzed and the number of cells required, can the authors assess if three replicates per condition was necessary or sufficient.

a) To address this question, we have added to the revised article the following plot to illustrate the relationship between barcode count and variation:

Supplementary Fig. 2b. Relationship between the mean number of barcodes in each library relative to the standard deviation of the element activity scores across replicates. The dashed line indicates our threshold of 10 minimum barcodes.

We have also added the following text to describe the effect of the barcode filter:

'This filter led to the following number of retained elements: i) HepG2 pilot library, 9,153/9,372 (97.7%); ii) K562 pilot library, 7,323/7,500 (97.6%); iii) HepG2 large-scale library, 139,886/164,307 (85.1%); iv) K562 large-scale library, 226,255/243,780 (92.8%); v) WTC11 large-scale library, 56,093/75,542 (74.2%); vi) HepG2 joint library, 56,018/60,000 (93.4%); vii) K562 joint library, 56,008/60,000 (93.3%); and viii) WTC11 joint library, 55,983/60,000 (93.3%).'

b-c) Naturally, more sequencing depth and more replicates will lead to greater measurement precision; however, due to sequencing cost constraints, three replicates were chosen as a reasonable compromise between experiment cost and measurement precision. To relay to the reader how these decisions, such as read depth and replicate number, impact our ability to detect significant elements, we have added to our revised article the multi-panel figure below to the supplement along with a power analysis to address both points. This analysis illustrates how downsampling the data impacts the robustness of the measurement. It is observed that between about 80-90% barcode downsampling, the final computed RNA/DNA ratio is almost identical to that of the full dataset (panel **g**). Between about 60-90% downsampling, the distribution of standard deviations across barcodes is highly similar (panel **h**). There is a modest reduction in the standard deviation among barcodes from 1 to 3 replicates (panel **i**). Finally, the discovery rate of significant elements has somewhat plateaued even after downsampling the data (panel **j**), suggesting that it is unlikely that a much greater proportion of elements would be classified as significant with increased sequencing depth.

Supplementary Fig. 2g-j. **g**, Correlation between the \log_2 RNA/DNA ratios of the downsampled and the full data. Barcodes were downsampled at fractions of 0.1, 0.2, 0.3, 0.4, 0.5, 0.7, and 0.9 of the total number of barcodes per sequence. The plot shows the average and the 95% confidence interval of 10 downsampling replicates for HepG2, WTC11, and K562 cells. **h**, Standard deviation between barcode \log_2 RNA/DNA ratios associated to the same element at downsampling fractions of 0.1, 0.2, 0.3, 0.4, 0.5, 0.7, and 0.9 for HepG2, WTC11, and K562 cells. For each sequence and downsampling fraction the downsampling was performed 10 times. **i**, Standard deviation between barcode \log_2 RNA/DNA ratios associated to the same element when using one, two, or three replicates for HepG2, WTC11, and K562 cells. **j**, Fraction of all sequences in HepG2, WTC11 and K562 that are found to be significantly active compared to the shuffled negative controls after downsampling barcodes at downsampling fractions of 0.1, 0.2, 0.3, 0.4, 0.5, 0.7, and 0.9. On the x-axis, the median number of barcodes after downsampling is displayed. The y-axis shows the average fraction and the 95% confidence interval of 10 downsampling replicates at four different significance levels. The significance of each sequence was calculated using the empirical p-value based on the MAD-score from the distribution of the downsampled shuffled negative controls.

2. Cell-type specificity versus donor specificity: The cell types compared are not donor matched, limiting the comparisons that can be made across cell types. HepG2 (human hepatoma 15yo white male), K562 (lymphoblast 53yo female), and WTC11(pluripotent stem cells, 30-34yo male Japanese) are distinct cell types each derived from different donors, of different ages and sexes, introducing possible transcriptional activity due to genotype.

a. For example, in the pilot MPRA, HepG2 cells revealed 30.6% putative enhancers to be more active than shuffled negative controls, while K562 showed 58.6%; since both libraries were prioritized with DNase peaks, what might cause the two-times activity difference between the two cell lines? Notably, activity differences across cell types diminished in the larger dataset.

b. The authors should address inter-donor and sex effects in their limitations. Can they determine what percent variability between cell-type activity may be attributed to genotype-specific (and sex-specific) regulation?

c. The authors should include two or more differentiated cell types from the WTC11 hiPSCs, even if only across a subset of predicted cell type specific CREs.

We thank the reviewer for bringing up the potential concern of having 'environmental' effects due to the cells not being donor matched. Previous lentiMPRA experiments from our lab using for example different donors of primary human cells from mid-gestation cortex or three different donor iPSC lines to generate organoids showed extremely high correlation between donors (see Deng et al. BioRxiv, 528663, 2023). In addition, even when testing human and chimpanzee sequences in both human and chimpanzee neural progenitor cells (NPCs) (Whalen et al. Neuron 2023), we observed extremely minor differences in either cell line for the MPRA, suggesting that the 'environment' does not have a large effect. To address this, we have added text in a new discussion paragraph that addresses many of the limitations of our lentiMPRA:

'We also tested sequences in three different cell lines originating from three different donors, which could lead to *trans* environmental effects. However, we should note that previous lentiMPRAs carried out by our lab in different donors for the same cell or organoid type⁶⁶ or even in cell lines from different species⁶⁷, chimpanzees and humans, show minor *trans* environmental effects.'

Regarding the explanation for why pilot MPRAs between cell types showed larger differences in the proportions of active elements, we have added the following line in the text to clarify:

'The differences in the proportion of active cCREs between cell types could not be directly compared because the set of negative controls were different in both cases (*i.e.*, one representing shuffled controls and the other not).'

The goal of our pilot libraries was mostly to demonstrate the scalability of our assay to our large-scale MPRAs, so they were not designed in a manner to perfectly compare the proportion of active enhancers in the same way that the large-scale MPRAs were.

3. Biochemical features predicting regulatory activity - The authors note the bias of the previous regression model towards a strong H3K27ac signal. Does their new model including MPRA signals have feature preferences. Across cell types, do they observe any difference/bias in prediction based on the number of features included? If including more features or datapoints increases prediction, does combining data from all three cell types give a more general and statistically powerful regression model, particularly since their data suggested little cell type-specific promoter activity?

Regarding the first proposed analysis, while it would be straightforward to define this analysis and execute it, it would be a difficult computational experiment to interpret. We could, for instance, randomly subsample a group of features to evaluate how predictive performance changes as a

function of feature count. However, it is important to note in interpreting this result that the result might simply represent whether a strongly predictive feature was included or excluded from the feature set. This means that the influence of feature count on performance is confounded with feature quality, and it is difficult to decouple the two. Nevertheless, we agree with the general sentiment that it is difficult to compare the performance head-to-head when each model has a different feature set. To help address this limitation, we performed the latter analysis proposed by the reviewer. We have now added this text to the manuscript along with the figure reproduced below:

'The variable feature count for each cell type led to the possibility of biasing performance. We thus benchmarked how a 'universal' feature set, merging features from all cell types, impacted model performance. The performance was only modestly better for models trained on the universal feature set, indicating that bias induced by variable feature size was minimal (**Supplementary Fig. 4d**).'

While there was a statistically significant improvement in performance when using the universal feature set, the effect size was modest (Note the small y-axis range).

Supplementary Fig. 4d. Violin plots showing the performances of the trained biochemical models on each of the ten cross-validation folds of held-out data. The lasso regression models evaluated are those that consider cell-type matched features (Bm) for the indicated cell type, or those that consider a universal set of biochemical features across all three cell types (Bu). An improvement relative to another model was evaluated with a one-sided, paired t-test.

4. Functional validation: It would be informative to test the extent that a catalog of thousands of functional CREs advances our ability to resolve genotype-to-phenotype associations.

a. Can the cis- and trans-target genes of these CREs be predicted by incorporating Hi-C and other epigenetic datasets, in order to apply cell-type specific activity-by-contact and activity-by-conformation models?

b. What percentage of top CREs can be further functionally validated by CRISPRi and/or prime editing? To what extent can their proximal and distal target genes be correctly predicted?

c. Likewise, can cell-type specific driver TFs be validated by manipulating their expression in the other cell types (e.g. WTC11) and exploring the impact on cell type identity?

d. Disease relevance: Can the authors apply their validated set of CREs and/or their novel sequence-based models to predict CRE function to improve fine-mapping of disease GWAS?

- a. We are currently working on a separate study with the ENCODE consortium (study led by Dr. Jesse Engrietz, creator of the ABC model), whose focus is to benchmark a large suite of algorithms and biochemical assays in their ability to predict the outcome of CRISPR perturbation data. Nevertheless, it is valuable and within the scope of this study to evaluate whether our lentiMPRA data can further bolster the ABC score. We benchmark whether our MPRA readout might better explain enhancer activity in K562 CRISPRi perturbation data from three studies: i) Gasperini et al 2019, ii) Fulco et al 2019, and iii) Schraivogel et al 2020. Using these datasets, we intersected our set of potential enhancers with those tested by CRISPRi. In the following analysis, we trained a binary classifier using logistic regression to predict whether a specific enhancer-gene linkage was statistically significant, considering the ABC model along with an interaction term with MPRA activity. We achieved the following results:

Gasperini et al 2019:

```
glm(formula = significant_EP_linkage ~ ABC.DNase.H3K27ac * MPRA, family =
"binomial")
Coefficients:
Estimate Std. Error z value Pr(>|z|)
(Intercept)      -4.59639    0.19142 -24.012 <2e-16
***
ABC.DNase.H3K27ac  0.62208    0.07319  8.499 <2e-16
***
MPRA              -0.23394    0.20270  -1.154
0.248
ABC.DNase.H3K27ac:MPRA 0.03669    0.07784  0.471  0.637
---
Signif. codes:  0 '***' 0.001 '**' 0.01 '*' 0.05 '.' 0.1 ' ' 1
1
(Dispersion parameter for binomial family taken to be
1)
Null deviance: 447.53  on 2899  degrees of freedom
Residual deviance: 358.01  on 2896  degrees of freedom
AIC: 366.01
```

Fulco et al 2019:

```
Call:
glm(formula = significant_EP_linkage ~ ABC.DNase.H3K27ac * MPRA, family =
"binomial")
Coefficients:
Estimate Std. Error z value Pr(>|z|)
(Intercept)      -3.750821    0.163289 -22.970 < 2e-16
***
```

```

ABC.DNase.H3K27ac      1.679589    0.149541   11.232   < 2e-16 ***
MPRA                   -0.006809    0.158644   -0.043   0.96577
ABC.DNase.H3K27ac:MPRA -0.303745    0.078973   -3.846   0.00012 ***
--
-
Signif. codes:  0 '***' 0.001 '**' 0.01 '*' 0.05 '.' 0.1 ' ' 1
1
(Dispersion parameter for binomial family taken to be
1)
Null deviance: 667.56  on 1948  degrees of
freedom
Residual deviance: 413.39  on 1945  degrees of
freedom
AIC: 421.39

```

Schraivogel et al 2020:

```

Call:
glm(formula = significant_EP_linkage ~ ABC.DNase.H3K27ac * MPRA, family
=
"binomial")
Coefficients:
Estimate Std. Error z value Pr(>|z|)
(Intercept)      -5.44408      0.21690 -25.099 < 2e-16 ***
ABC.DNase.H3K27ac  0.39039      0.05555  7.027 2.11e-12
***
MPRA              0.34648      0.17401  1.991 0.0465 *
ABC.DNase.H3K27ac:MPRA -0.06443      0.06779 -0.950 0.3419
---
Signif. codes:  0 '***' 0.001 '**' 0.01 '*' 0.05 '.' 0.1 ' ' 1
1
(Dispersion parameter for binomial family taken to be
1)
Null deviance: 379.70  on 5224  degrees of freedom
Residual deviance: 315.46  on 5221  degrees of freedom
AIC: 323.46

```

The results indicate that in two of the three datasets, considering the MPRA data or an interaction term between the MPRA and ABC score improves the model in a statistically significant manner. However, the z-values associated with the MPRA data as well as the decrease in Akaike Information Criterion (AIC) is subtle relative to a model that considers only the ABC score alone. Thus, we conclude that considering MPRA data negligibly improves predictive performance with respect to enhancer-promoter linkages. In section b) below, we add some discussion to the text addressing both a-b) based on these results.

- b. Using these same three datasets, we next ranked the intersected enhancers by their MPRA activity, split this set into 20 or 10 equally sized bins (depending upon the dataset size), and

evaluated the proportion of enhancers with a significantly supported promoter-enhancer interaction. The results are shown below:

Supplementary Fig. 3. Relationship between MPRA and CRISPRi datasets. Fraction of elements validated by CRISPRi (*i.e.*, called to significantly regulate a target gene) for equally sized bins of MPRA activity. The bins with the highest activity level exhibit a greater likelihood for validation. The left and right numerical intervals of the MPRA activity bins are shown comma-separated, with open parentheses meaning the exclusion of the left interval and square brackets meaning the inclusion of the right interval. Three CRISPRi datasets were cross-referenced for this analysis (Fulco et al, 2019; Schraivogel et al 2020; Gasperini et al 2019).

Based on these results, we have added the following line to address comments a-b):

‘To assess whether our potential enhancers could be used to validate significant enhancer–promoter interactions and/or predict CRISPRi results, we intersected this set against those tested in three different CRISPRi perturbation studies^{34–36}. We examined the proportion of our binned activity scores that were assessed as significantly regulating a promoter. We observed that the bins with highest activity had nearly a two-fold increase in the proportion of validated enhancers relative to bins with low activity (**Supplementary Fig. 3**), suggesting our MPRA datasets could be used to determine sequences with larger CRISPRi effects. Considering our activity scores alongside Activity-by-contact (ABC) scores³⁴, we observed that our scores only subtly improved performance in the task of discriminating significant from non-significant enhancer–promoter interactions in these three studies (data not shown). This small improvement in performance may be partially explained by the observation that the H3K27ac signal, which is already considered in the ABC model, has the advantage of integrating local and distal regulatory information, which may overshadow the consideration of an element’s local activity alone.’

To evaluate the extent of proximal/distal target genes that are correctly predicted, it is important to note that the field has not yet adequately established a gold standard dataset for assessing enhancer-promoter linkages. While it is possible to evaluate the agreement (or lack thereof) between MPRA and CRISPRi data, the latter datasets have critical limitations of their own, including false negatives derived from being underpowered to detect modest changes in gene expression levels (i.e., especially changes in poorly expressed genes due to lack of read depth), and false positives derived from the loss of spatial resolution caused by the potential spread of heterochromatin to neighboring enhancers (i.e., in the case of superenhancers or “stretch enhancers” which comprise many of the significant hits of such datasets). In the future, it will be important for the field to generate large-scale deletion datasets to enable better cross-comparisons.

c) It is not guaranteed that perturbing a single cell-type specific TF in isolation will drive differentiation to the corresponding cell type in which its activity was enriched. However, to further functionally validate TF activity, we set out to examine whether perturbing the TF can lead to perturbations in the regulation of elements to which it binds in the cell type of interest. The most appropriate dataset we could think of to answer this question is derived from Calderon et al., 2020 *bioRxiv*, a study that simultaneously knocked down different TFs with CRISPRi while evaluating the effects of these TF perturbations on the dysregulation of regulatory elements to which the TFs bind. We reanalyzed this dataset from the perspective of six TFs that were detected by MPRALegNet and examined by Calderon et al, and incorporated this result as a supplementary figure (**Extended Data Fig. 7**). We were able to validate 4 of 6 TFs as conferring significant regulatory activity on regulatory elements. It is important to note that the negative result for the remaining two TFs might be explained not just by the lack of TF activity, but possibly by poor knockdown efficiency of the TF by CRISPRi, or alternatively by compensatory effects conferred by other TF family members.

Extended Data Fig 7a. Effect of TF knockdown on loss of regulatory element activity. Shown are cumulative density plots for the subset of elements possessing TF binding sites to the corresponding knocked down TF, relative to a control (i.e., non-targeting guide RNA) shown in black. The x-axis indicates the minimum (i.e., most negative) fold change among three guide RNAs targeting the TF. Data shown is from the high MOI condition sampled at day 10 (Calderon et al., 2020). P-values indicate a shift in the distribution as assessed by a one-sided Kolmogorov-Smirnov (K-S) test, followed by a Bonferroni multiple hypothesis testing correction.

d) Regarding disease relevance, we tested seven genomic regions nominated by ENCODE to be tested across projects that are disease associated, substantially tiling them with MPRA tested sequences. We now show these results in detail in revised **Fig 1a** and added **Extended Data Fig 2**. We next utilized MPRALegNet, a novel modelling approach that we added to the revision, to showcase how our results along with modeling can be used for disease relevance and fine-mapping, and can help to better fine-map published GWAS SNPs. An example of one new visualization is shown below for the *BCL11A* locus:

a

Extended Data Fig 2a: MPRA activity in selected disease loci. UCSC genome browser tracks annotating, from top to bottom: i) Lead single nucleotide polymorphisms (SNPs) from published Genome-wide Association Studies (GWAS); ii) Common SNPs from the 1000 Genomes Phase 3 dataset; iii) GENCODE V44 gene track; iv) enhancer activity scores from the pilot K562 MPRA library for each of the five enhancers tested, with stronger red indicative of higher activity; v) scores corresponding to the large-scale K562 MPRA library, tested in both orientations; vi) H3K27Ac and vii) DNase I hypersensitivity signal in K562 cells; viii) base conservation among 100 vertebrate species; and ix) ENCODE-annotated cCREs in the locus.

Next, we illustrate that the MPRALegNet model itself can be used to generate variant effect predictions for every lead SNP and SNP in linkage disequilibrium (LD) with a lead SNP, to better fine-map the causal SNP in these loci. We give the example of *LMO2* and *RBM38* as loci in which MPRALegNet can facilitate identification of a causal SNP, which can represent a possible loss-of-function or gain-of-function SNP, as shown below:

Fig. 5: Performance of MPRALegNet in variant effect prediction. **a**, This panel follows that shown in Fig. 1a, except indicates variant effect predictions derived from MPRALegNet for variants observed to be in LD with GWAS lead SNPs ($R^2 \geq 0.8$) in the *LMO2* locus. Also shown in the bottom panel is a zoom-in of the implicated causal SNP (*i.e.*, expanded from the vertical dashed lines), located within a DNase I site in the intron of *LMO2*, having the strongest predicted effect size.

Extended Data Fig. 8: Variant effect predictions in the *RBM38* locus. This panel follows the same scheme as that shown in Fig. 5a, except displays the *RBM38* locus and two zoomed in regions, each expanded from within the corresponding dotted lines.

To further validate the accuracy/validity of variant effect predictions, we further demonstrate that the SNP predictions are largely concordant with the following results text and associated figure:

To further evaluate the meaningfulness of such predictions as a fine-mapping strategy, we benchmarked model predictions against two complementary tasks. First, we verified performance on variant effect data using six sets of allele-specific events (ASEs) found in chromatin accessibility (*i.e.*, from ATAC-seq and DNase-seq) and transcription factor binding data (*i.e.*, from CHIP-seq) in HepG2 and K562 cells available in the UDACHA⁵¹ and ADAstra⁵². The significant ASEs provide information on variant effects, including the preferential TF binding or chromatin accessibility as allelic imbalance towards the reference (Ref) or the alternative (Alt) allele. Thus, it was possible to validate MPRAlegNet by assessing whether the model predictions

for Ref and Alt were concordant with the ASEs, *i.e.*, whether the difference in predicted scores for Ref and Alt at each variant had the same sign as the ASE preferred allele. For all six tested combinations of ASE sources and cell types, we observed significant associations between the observed and predicted scores both before and after excluding cases in which the ASE was non-significant or model predictions were too uncertain (Fisher's exact test odds ratios >1.5 and $p < 0.05$, **Fig. 5b**, **Supplementary Table 7**). We conclude MRPAlegNet successfully recognizes allele-specific regulatory SNP (rSNP) effects in the matched cell types.'

Fig. 5b, Scatter plots of predicted variant effects and observed allele-specific differences detected in ChIP-seq, ATAC-seq, and DNase-seq data in K562 and HepG2 cells. Indicated are the number of cases in which the predictions and observations are concordant (C, shown in blue), discordant (D, shown in red), or not considered (shown in grey) because the ASE FDR > 0.05 or model predictions are too uncertain (p -value > 0.05, **Supplementary Methods**). The corresponding Odds Ratio (OR) is also indicated.

Please also see an expansion of our prior results validating our predictions against saturation mutagenesis MPRA data for multiple enhancers and promoter SNPs (**Extended Data Fig. 9**).

Referee #3/4

This paper presents a scaled-up version of lentiMPRA assay, which involves the parallelized insertion of sequence elements of interest into random positions in the genome multiple times, followed by a reporter gene expression readout to assess sequence element function. The authors performed this assay on three cell lines and analyzed the data. The results confirmed that human promoters are more orientation-specific than enhancers. The authors developed a custom neural network, MPRAnn, and built upon a previous neural network, EnformerMPRA, and a lasso regression model based on biochemical features to predict element functional activity. In the final section, the authors tested a common set of regulatory elements in the three cell lines.

The updated lentiMPRA version represents an advancement over its low-throughput predecessor, demonstrating impressive consistency and reproducibility across replicates. However, the biological questions it investigates appear limited in their capacity to reveal novel insights. It remains unclear what unique biological information this assay can offer that has not already been discovered. Moreover, the computational analysis is shallow with minimal insights provided; a similar result could be achieved through standard motif analysis of accessible chromatin sites.

We thank the reviewers for the feedback on the limited novel insight into biological questions. To address this, we have added the following to our revised version of our article:

1. We showcase how our MPRA results that extensively tiled seven different human disease loci can identify functional regulatory elements targeting these disease-associated genes (**Fig1a, Extended Data Fig 2**), providing candidate sequences for disease diagnostics and therapeutics.
2. A new neural network, MPRAlegNet, that achieves state-of-the-art performance on our dataset (**Fig 4a-d, Extended Data Fig 5**), with clearer distinction between TFBSs learned by classical motif search methods vs neural network interpretation methods (**Extended Data Fig 4, Extended Data Fig 6, Extended Data Fig 10**) allowing enhanced regulatory element identification and characterization.
3. Revised interpretation of this model to deeply interrogate more complex findings of the neural network model, such as the nonlinear/cooperative behavior and dosage-dependence of motifs on activating expression (**Fig 4e-h, Extended Data Fig 7**).
4. We show how our data can be used to select targeted sequences in CRISPRi screens that will have a large effect on gene expression and validate transcription factors engaged in regulation (**Supplementary Fig 3, Extended Data Fig 7a**).
5. Applications of our data and model to GWAS fine-mapping and genetic variant effect prediction and interpretation, especially applied to numerous disease-associated loci (**Fig1a, Fig 5, Extended Data Fig 2, Extended Data Fig 8, Extended Data Fig 9**).

Major issues:

- The experimental method utilized in this study is an interventional experiment, which has the potential to reveal causal relationships. However, due to the insertion of CREs at random

positions within the genome, the context within a CRE grammar is not replicated as in native sequences. As a result, the primary question being investigated is how short sequence elements function outside of their original context. This approach essentially measures the functional effect of sequence elements averaged across different random contexts, which should be taken into account when drawing conclusions from the results. It is important to note that this aspect is not explicitly stated in the text. Therefore, it is necessary to consider that the effect size is measured across random contexts when interpreting the results.

We completely agree with the reviewers that this is a major limitation of our assay. When initially developing this assay many years ago, we also considered a 'safe harbor' approach where the library will integrate into the same genomic location, thus not having the limitation that the reviewers raise of random integration. However, this approach also has limitations. One, is that sequences are only assayed in one location which could confound results. We reasoned, as shown in our high lentiMPRA replicate correlations, that having many random integrations for each assayed sequence will overcome this limitation. The second, is the inability to test a high number of sequences as we set out to do here, with a single integration into a 'safe harbor' locus limiting the number of sequences we can test. To address this comment, we have added an entire new paragraph to the discussion on the caveats of our work, including the following text to address these limitations:

'LentiMPRA tests sequences outside of their genomic context, which could lead to false positives. In addition, the transgene integrates in numerous different genomic locations which could lead to site-of-integration effects. However, to correct for this, we required at least 10 integrations per sequence, had antirepressors on either side of our integrated lentiMPRA construct, and observed high correlations between technical replicates.'

- It is unclear what the technical innovation of the revised lentiMPRA is compared to their previous version. Could the authors add a clear explanation of differences and the current contribution?

We thank the reviewers for pointing this out. We developed this protocol and the work presented here as part of our five year long ENCODE project and this publication is part of the ENCODE4 package. Due to that, we were limited in publication timing as part of the package and did not want to limit the adaptability of this technology by other labs due to this timing, thus publishing a detailed *Nature Protocols* on this methodology so that it can be adopted by other labs. We thus carefully went through the article and revised the text to clarify this.

- Although the loss of sequence context presents challenges, there remain several intriguing biological inquiries to explore. Nevertheless, interpreting the current results is not a straightforward task. For example, the experiment seeks to determine how well enhancers function as promoters by placing them upstream of a minimal promoter in a cross-comparison with promoter elements. The results reveal that enhancers exhibit low activity across all cell lines (as depicted in Figure 1d). The authors describe enhancers as exhibiting a "right skew" distribution, but this is also observed for negative controls, like K562.

When describing the right skew distribution, we should note that this is relative to promoters:

'Promoters exhibited, on average, higher activity scores and a bimodal distribution compared to potential enhancers, which exhibited a right-skewed distribution in all cell types (**Fig. 1d**).'

We also have added an entirely new paragraph to the discussion listing the various limitations of our study. In addition, we have the following text regarding the placement of promoter near a minimal promoter:

'Of note, our lentiMPRA design tested promoters along with a minimal promoter that is 32 bp long (minP), which could affect promoter activity. However, this approach enabled us to test *en masse* hundreds of thousands of enhancers along with thousands of promoters and compare them in the same assay. Our results were similar to previous reports^{37,38}, showing orientation biases for promoters, and identified an enrichment for motifs that are known to provide ubiquitous promoter expression, further supporting that the addition of these 32 bp to our assayed promoters likely did not affect our findings.'

- In terms of the orientation dependence results, the difference in directional preference between promoters versus enhancers is not a big effect (albeit statistically significant). However, the language used throughout the text is quite strong. The way in which the results are plotted in Fig 2, is difficult to assess. For instance, the Pearson's r for K562 FvF and RvR in Fig 2a seems to be under 0.8 on average. But, the correlation between replicate pairs is "0.88-0.96". Where is this dropoff coming from? I think a clearer description of the analysis could be helpful.

We concur that the effect size for promoters displaying greater strand asymmetry is relatively weak, and have now toned down the language of the strand orientation effects throughout the text:

Abstract: "By testing sequences in both orientations, we find promoters to have ~~significant~~-strand orientation effects."

Results: With respect to **Fig 2b**, we state the following: 'These findings suggest that the activity of cCREs is largely, but not completely, independent of orientation.'; which indicates that the strand asymmetry effect is small.

'Collectively, these results indicate that core promoters possess modest ~~reproducible~~-orientation dependence and little cell-type specificity.'

We have also edited the text to read:

'Consistent with prior work^{37,38}, we observed that promoters displayed modestly greater strand asymmetry effects relative to potential enhancers in all cell types examined (**Fig. 2c**), supporting the conclusion that they can contain TFBSs that promote transcription unidirectionally (or at least more unidirectionally than potential enhancers).'

Discussion: 'By testing all protein-coding promoters, we find ~~significant~~-promoter activity bias in terms of strand orientation.'" and "In addition to observing ~~significant~~-promoter activity bias in terms of strand orientation in line with previous work..."

Regarding the latter point, the 0.88-0.96 statistic mentioned alludes to this line in the text: 'The final element activity scores, i.e. $\log_2(\text{RNA}/\text{DNA})$ ratios, were highly concordant across replicates, ranging from a Pearson correlation of 0.88-0.96 between replicate pairs (**Supplementary Fig. 1c-d**).'; which is in the section for the pilot K562 MPRA. However, for the large-scale MPRA, the reproducibility between replicates is 0.75-0.80 (updated '**Supplementary Fig. 2d**'), which is consistent with the range for K562 shown in **Fig 2a**.

- The pilot experiment shows high replicability but the main CRE characterization results are drawn from the larger scale experiments which have a much lower replicability. The authors should provide a better explanation about these differences.

Based on the scatter plot (Fig. 2b), it appears quite symmetric. It appears that the reverse strand preference has just as much preference in many sequence elements as the forward strand. It could be beneficial to color promoters and enhancers using a different coloring scheme. The language about Fig 2c conclusions about TFBS promoting transcription unidirectionally is much stronger than the evidence.

We have modified the text to better indicate that the reproducibility was lower in the large-scale library, along with the following explanation:

'Element activity scores were also strongly concordant across replicate pairs, with Pearson correlations of 0.94 for HepG2 cells, 0.76 for K562 cells, and 0.76 for WTC11 cells (**Supplementary Table 4, Supplementary Fig. 2c-e**). Averaging across the three replicates, we also observed strong correlations among element activity scores between cCREs common to both our pilot and large-scale libraries (Pearson $r = 0.94$ in HepG2 cells and $r = 0.81$ in K562 cells, **Supplementary Fig. 2f**). Similarly, visualizing the large-scale K562 library in the *HBE1* locus (**Fig. 1a**) and the other six disease-associated loci (**Extended Data Fig. 2**) confirmed strong agreement with the K562 pilot library. However, the large-scale library tiled these loci with greater density and highlighted additional functional regulatory elements. Despite these benefits, the inter-replicate correlations for both large-scale libraries were less than those of the pilot libraries due to the tradeoff between the element library size and per-element sequencing depth. To further investigate this tradeoff, we downsampled barcodes to evaluate the robustness of our measurements and findings. Downsampling to 90% of the barcodes led to a near perfect Spearman correlation with respect to element activity scores relative to those derived from the full dataset, while smaller proportions of the barcodes degraded this correlation (**Supplementary Fig. 2g**). The distributions of standard deviations for element activity scores across barcodes became tighter when considering larger barcode downsampling proportions (**Supplementary Fig. 2h**) and more replicates (**Supplementary Fig. 2i**).'

As mentioned in the previous comment, we have toned down the language throughout the text surrounding orientation effects. Relevant to the comment on **Fig 2c**, we have now modified the text to read: 'Consistent with prior work^{34,35}, we observed that promoters displayed modestly greater strand asymmetry effects relative to potential enhancers in all cell types examined (**Fig. 2c**), supporting the conclusion that they can contain TFBSs that promote transcription unidirectionally (or at least more unidirectionally than enhancers).'

Regarding the comment about coloration by group, we understand the motivation for coloring promoters and enhancers differently in the plot. However, as noted by Dr. Lior Pachter and others, it is important to note that due to the high number of data points ($n=98,360$), this style of coloration can easily mislead a reader. We illustrate this point by coloring **Fig 2b** alternately by promoters (colored in blue) followed by enhancers (colored in red), or vice versa:

Despite the plot representing the same set of data, the high overlap between distributions can lead the reader to have two different interpretations depending upon the plotting style; moreover, it does not capture the relative density of both distributions, which is critical to the question at hand. Thus, we prefer to not represent the data this way relative to the summary of the data presented in **Fig 2c**.

- When placing promoters in different orientations, how does this affect promoter competition with the minimal promoter?

We have the following text in the discussion to address this point:

'Of note, our lentiMPRA design tested promoters along with a minimal promoter that is 32 bp long (minP), which could affect promoter activity. However, this approach enabled us to test *en masse* hundreds of thousands of enhancers along with thousands of promoters and compare them in the same assay. Our results were similar to previous reports^{37,38}, showing orientation biases for promoters, and identified an enrichment for motifs that are known to provide ubiquitous promoter expression, further supporting that the addition of these 32 bp to our assayed promoters likely did not affect our findings.'

Fig 2 overall is quite complex to interpret and would benefit from a better description of the takeaways. For instance, it seems that the lentiMPRA assay leads to the same correlation for RNA-seq across cell types for HepG2. Why is this and is there a better way to stratify the results to achieve a more precise conclusion from this analysis (eg. strong vs weak promoters, highly vs lowly expressed genes or housekeeping vs differential cell type specific)?

Our purpose in showing that the lentiMPRA leads to a similar correlation across RNA-seq of multiple cell types was to reinforce the theory that promoters inherently possess little cell-type specificity, as noted in the text: 'Furthermore, when comparing against endogenous expression levels, we observed: i) MPRA measurements from the matched cell type displayed nearly the same correlations as those from a different cell type...(Fig. 2d). These results indicate that core promoters possess...little cell-type specificity.'

Regarding the second point, **Fig 2d-e**, **Extended Data Fig 3a-e**, and the newly added **Extended Data Fig 3f-i**, which is focused on MPRA vs CAGE from the matched promoter (please see response to **Minor issue 1** below), show the relationship between the continuum of strong vs weak and highly expressed vs lowly expressed genes.

However, we concur that these results do not explain whether genes with cell-type specific expression (i.e., defined as those that exhibit large differences in RNA-seq signal across cell types) obey different properties than promoters corresponding to housekeeping genes (i.e., defined as those that exhibit little difference in RNA-seq signal across cell types). To address this point, we performed a new correlation analysis and added the following text to the results section:

'To further evaluate whether our promoter measurements explained cell-type-specific gene expression, we computed the deviations of promoter activity from their mean activity across cell types, as well as the corresponding deviations for endogenous gene expression levels across the same cell types. All cell types exhibited a Pearson correlation of ≤ 0.11 between the two sets of measurements; reinforcing our prior result that the presence of enhancers, superenhancers, and Polycomb targeting in each locus more dominantly explain the deviations of endogenous transcriptional activity from core promoter activity³⁹'

These results were robust to the choice of RNA-seq data or CAGE data matched to the promoter tested.

- The computational modeling provides minimal insights. It falls short of highlighting the benefits of this dataset and the power of deep learning modeling. The main insights are motifs learned by MPRAnn via attribution map clustering with TF-MoDISCo-lite. In principle, this insight could likely be accomplished with standard motif analysis. MPRAnn and other neural networks can learn more complex relationships between motifs, which could potentially reveal deeper insights into cis-regulatory grammars. But that was not done here, even though more thorough *in silico* interrogation of DNNs is now standard protocol for model interpretation.

Please see our response to **Reviewer 1, points 20-21**, in which we have added additional analyses to comprehensively show which motifs a typical workflow using STREME is capable of detecting. Based on this analysis, we find that the two approaches are complementary to each other and are each able to capture motifs that are difficult to recover with the opposite approach. Nevertheless, we concur that our motif analysis scratches the surface of what a deep learning architecture is capable of learning, such as combinatorial relationships between motif pairs. To

address this shortcoming, we have now added a more thorough examination of these motif interdependencies, adding 4 main text panels and 11 extended data figure panels which examine the behavior of the model in capturing cooperative (i.e., supermultiplicative) effects and saturating (i.e., submultiplicative) effects when considering homotypic or heterotypic TFBSs in each of the three cell types. We have added the following text to the results section to further explain these figures:

MPRALegNet predicts combinatorial TFBS effects

To gain insight into the nature of combinatorial TFBS effects learned by MPRALegNet, we further examined the top ten most abundant activating TFBS motifs detected in each cell type. First, we tested the impact of the number of copies (i.e., dosage) of homotypic (same) TFBSs on reporter expression. In each cell type, MPRALegNet could accurately predict the activation profile for elements containing between 1-5 sites of the indicated TFBS (**Fig. 4e, Extended Data Fig. 7b-e**). In most cases, TFs displayed a multiplicative (i.e., log-additive) pattern with respect to TF dosage (**Extended Data Fig. 7c**)⁴⁹. However, several TF families, such as STAT (considering the STAT1/4/5A/5B motif) in K562 cells and ETS-related (considering the ELK4 motif) in WTC11 cells, displayed sub-multiplicative patterns in the highest dosages (**Fig. 4f, Extended Data Fig. 7e**), indicative of a saturating expression effect. Super-multiplicative (i.e., cooperative) effects were also observed for certain dosages, such as the increase observed from 1 to 2 sites for the STAT TF family (**Fig. 4f**).

Next, we evaluated deviations from multiplicative effects for heterotypic (different) TFBS pairs, as quantified by an interaction term when considering the subset of elements with: i) a single site to either of the two TFs, or ii) co-occurring instances of both TFs⁵⁰. We observed both super-multiplicative and sub-multiplicative effects for different TF pairs in HepG2 cells, as indicated by positive or negative interaction term coefficients, respectively (**Fig. 4g**). The magnitude of these terms was strongly correlated between the predictions and observations ($r = 0.92$), suggesting that MPRALegNet learned more complex combinatorial properties among TF pairs (**Extended Data Fig. 7f**). For example, the co-occurrence of ATF3/FOS::JUN and FOXD2 sites led to the strongest super-multiplicative effect (**Fig. 4h**); conversely, the co-occurrence of HNF4A/G and NFYA/C sites led to a sub-multiplicative effect (**Extended Data Fig. 7g**). Similar findings were observed in both K562 and WTC11 cells (**Extended Data Fig. 7h-k**), such as the ability to predict a strong super-multiplicative effect among co-occurring NRF1 and FOS::JUN motifs in WTC11 cells (**Extended Data Fig. 7l**). Collectively, MPRALegNet was able to model nonlinear interdependencies between TFBS combinations in all cell types.'

The two new sets of panel figures are shown below:

Fig. 4e-h. **e**, Heatmap indicating the relationship between homotypic TFBS dosage (n=1 to 5 TFBSs) as well as the observed and MPRALegNet-predicted response in K562 cells. **f**, Relationship of the distributions of observed and MPRALegNet-predicted element activity scores in K562 cells for the subset of elements possessing 0 to 5 TFBSs corresponding to the STAT1/4/5A/5B TF family. **g**, Heatmap indicating interaction term coefficients reflecting supermultiplicative and submultiplicative effects, shown in red and blue, respectively, when considering elements possessing the indicated pair of heterotypic TFBSs in HepG2 cells. Coefficients fit to observed and predicted values are shown in the upper and lower triangles of the heatmap, respectively. **h**, Relationship of the distributions of observed and MPRALegNet-predicted element activity scores in HepG2 cells for the subset of elements possessing zero TFBSs, one TFBS corresponding to either the ATF3/FOS::JUN or FOXD2 families, or one TFBS for both TF families. The horizontal line corresponds to the expected activity score under a multiplicative model.

Extended Data Fig. 7b-I: Combinatorial TFBS effects learned by MPRALegNet. b-e, These panels are arranged in the same scheme as Fig. 4e-f, except display results for homotypic TFBSs in HepG2 (b-c) and WTC11 (d-e) cells for the indicated TF families. **f,** Scatter plot of interaction terms fit to predicted and observed values for TFBS pairs in HepG2 cells. The data is the same as that presented in Fig. 4g, but also includes Pearson

(r) and Spearman (ρ) correlation values. **g**, This panel is similar to that shown in **Fig. 4h**, but shows an example of a pair of heterotypic TFBSs that exhibit a submultiplicative effect when co-occurring. **h-k**, These panels are arranged in the same scheme as Fig. 4g and panel (f), except display results for K562 (**h-i**) and WTC11 (**j-k**) cells for the indicated TF families. **l**, This panel is similar to that shown in **Fig. 4h**, but shows an example of a pair of heterotypic TFBSs that exhibit a supermultiplicative effect when co-occurring in WTC11 cells.

- Moreover, the hyperparameter choices in MPRAnn are not standard; softmax hidden activations for conv layers and sigmoid for dense layers. It seems like this model was purposefully designed to be inhibited to make Enformer-MPRA appear better. A simple hyperparameter search would lead to significantly improved performance.

Please see our response to **Reviewer 1, point 23**, which provides an explanation regarding our previous hyperparameter optimization approach with MPRAnn. It was not our intention to demonstrate EnformerMPRA as the best model. Rather, we share the reviewers' goal of training the best model possible on our dataset, for the sake of deriving a model that is simpler to interpret and easier to deploy for downstream users in predictive tasks relative to Enformer. To bridge the performance gap, we have now performed a more systematic hyperparameter tuning using our newly introduced, state-of-the-art MPRALegNet model (**Fig. 4b, Extended Data Fig. 5a**), as well as corresponding ablation analysis to converge on a nearly optimal model. MPRALegNet's performance is now comparable or superior to that of Enformer on our datasets.

- In terms of the biochemical model, a change in regularization strength in the lasso model can alter the list of salient biochemical features. So modeling itself is only meaningful if generalizable knowledge is extracted from the analysis. But there is minimal effort to validate the computational modeling; it is descriptive with no depth.

Regarding the biochemical lasso regression model, it is true that regularization can change the features selected. Anticipating this failure mode, we note in our article the following:

'Despite these findings, model interpretation was inherently limited by the substantial degree of multicollinearity among features, as previously observed²¹. To minimize the chances that other important factors may have been overlooked by lasso regression, we identified alternative features which were highly correlated to the top-ranked features of our models (**Supplementary Fig. 4a-c**).'

This style of modeling allows us to observe any feature overlooked by the lasso regression procedure. Nevertheless, it is true that the biochemical model is limited in the mechanistic insight it can give because it can only detect associations.

We believe this is where the deep learning model has the opportunity to truly shine as a complementary approach that outperforms the biochemical model, because its interpretation is ultimately rooted in the TFBSs it learns from the input sequence. To extract generalizable knowledge from this model, we interpret the principles that the network has learned, and validate

whether our trained model could predict genetic variant effects derived from saturation mutagenesis and genetic variant testing experiments, which are an orthogonal data type that requires a successful model to have learned a causal principle. As mentioned in earlier discussion points, we now provide additional evidence that the model has learned combinatorial motif properties.

- Also, the lentiMPRA elements are embedded randomly in the genome, so it's not quite clear how the biochemical features make sense for this analysis; they correspond to the original location of the sequence element.

Modeling the biochemical signals at the endogenous region has been used productively by our group several times to identify likely TFs and epigenetic marks associated with transcriptional regulation (Inoue et al *Genome Research* 2017, Klein et al *Nature Comms* 2019, Klein et al *Nature Methods* 2020). Our rationale in this modeling decision is that the endogenous locus associated with these biochemical marks is likely also associated with many of these marks in the randomly integrated locus, and thus knowledge of these associations can be exploited to explain MPRA activity. We have modified the following sentence in the text to reflect this:

'Previous work by our labs trained regression models to characterize enhancer activity, guided by the observation that the biochemical signals associated with the endogenous loci strongly explain lentiMPRA activity^{19,21}.'

- The authors claim Enformer is not used for the motif analysis because of multicollinearity of features. But how about the colinearity of biochemical features used in the biochemical model? Similar to the biochemical model, Enformer can still be used for interpretability compared to MPRAAnn or a better explanation should be given for not using it.

We concur that the colinearity of biochemical features plagues model interpretation, which was previously addressed in the following excerpt: "Despite these findings, model interpretation was inherently limited by the substantial degree of multicollinearity among features, as previously observed¹⁹. To minimize the chances that other important factors may have been overlooked by lasso regression, we identified alternative features which were highly correlated to the top-ranked features of our models (**Supplementary Fig. 4**). This analysis revealed a large suite of TFs whose activity was strongly correlated to the top features, and identified other relevant components of cohesin, including SMC3 and STAG1 (**Supplementary Fig. 4a**)."

However, the scale of this problem is much vaster with Enformer because it encompasses upwards of an 8-fold greater number of features (i.e., depending upon the cell type examined), making it impossible to conveniently visualize the equivalent sort of heatmap for Enformer's ~5,300 features. Our revision now incorporates a high-performing neural network, MPRALegNet, whose performance is as strong or superior than Enformer's on 2 of 3 cell types while being much more convenient to interpret; given it's strong performance and succinctness, we find it a more compelling and a simpler narrative to focus our efforts on MPRALegNet's interpretation.

- More critically, there is no attempt to connect the insights (whether motifs or biochemical features) across the 3 models. What is driving the improved predictive performance?

With the addition of our newest neural network, MPRALegNet, we were able to demonstrate that there is nothing fundamentally superior about Enformer that cannot be achieved by training a better model on the dataset itself. We hope that integrating this new model coupled with additional results to interpret it will better illustrate this point. We keep Enformer results simply for benchmarking purposes. Please see **Reviewer 1, comments 17 and 19** for additional information.

Technical concerns

- Where are the lentivirus' integrating into the genome? Does the location have an effect on readout? What is the variance of the same CRE injected into different loci in the same cell (across the 20-50 elements)? The raw data should be provided (pre-averaging the elements).

Unfortunately, our technology does not directly measure the lentivirus integration site. Prior work has indeed demonstrated that location does have an impact on expression (Akhtar et al., *Cell* 2013, Maricque et al. *Nature Biotechnology* 2018). However, our strategy in lentiMPRA is to average the results across multiple sites, and therefore intentionally average away location effects to achieve a more precise estimate of the location-independent effect of an element on reporter expression. As mentioned in this reviewer's first Major point, we have also added text mentioning this caveat to our discussion. Although it is certainly possible to compute the variance for a given CRE, it is important to note that comparing the variances between CREs is confounded by the effect of the barcode itself (Lee et al, *Genome Res* 2021, "Sequence-based correction of barcode bias in massively parallel reporter assays") as well as PCR amplification bias of the sequence (Qiao et al, *Genet Epidemiol* 2021, Kim et al *Genome Res* 2021 "Correcting signal biases and detecting regulatory elements in STARR-seq data"). Our response to **Reviewer 2, Point 2** further illustrates that the variance is partially confounded by technical effects associated with the number of genomic integrations. Given these multiple confounding factors, it is not possible to decompose the influence of position-specific effects relative to technical sources of bias, in the absence of explicit models to correct for such biases. Given these limitations, we feel our approach of aggregating all data, which leads to a more robust and precise estimate of element activity, is more fitting for the focus of this article. Despite these caveats, all raw data for all MPRA experiments presented in this study are fully deposited in the ENCODE portal and could be easily be downloaded and used by other researchers to carry out these analyses.

- Element specificity score in Fig 5c is a confusing metric, the authors should directly plot distributions.

We can see how jumping straight into **Fig 5c** (now **Fig 6c**) would lead to confusion about this score and it's justification. However, we have tried our best to guide readers in the text into our thought process in computing this useful metric by stepping through the intermediate steps to

compute the score. For instance, the distributions requested had already been plotted in our **Extended Data Fig. 8** and **Extended Data Fig. 9** (now **Supplementary Fig. 6f** and **Supplementary Fig. 7**) as stepping stones towards the calculation of the Element Specificity Score in **Supplementary Fig 9**.

-HepG2 seems to have a much higher replicate consistency compared to K562 and WTC11. Comment on this would be helpful.

This is a great comment and one that we do not have an easy answer towards. This could definitely be due to various technical differences in the experiment. HepG2 are adherent cells versus K562 that grow in suspension. Due to that, getting lentivirus to integrate into these cells is much easier and the multiplicity of infection (MOI) is much higher. In our experiments, we had an MOI of 50 for HepG2 versus 10 for K562. WTC11, while also being adherent, are extremely sensitive and die very easily due to manipulations, such as lentivirus infection, and so we had to use a lower MOI of 10. We added the following text to the methods section to explain this:

'The higher MOI in HepG2 is due to these cells being adherent compared to K562 that grow in suspension.'

'For the large-scale WTC11 library, 38.4M WTC11 cells per replicate were seeded in four 10cm dishes (9.6M per dish), incubated for 24 hours, and infected with the library along with 8 µg/ml polybrene, with an estimated MOI of 10, due to higher MOIs being lethal for these cells.'

-Data availability: WTC11 data is missing and K562 is mislabelled -- ENCSR022GQD, according to ENCODE portal, is HepG2. A clearer description of these files and how the dataset was processed up to data splits would be greatly beneficial for reproducibility.

Thank you for noting this error. Please see **Reviewer 1, comment 33** for more information about our revised data upload which includes all data generated in this study. The **Methods** sections "MPRA processing pipeline" and "Data pre-processing and model training" describe in detail the processing parameters as well as data pre-processing steps involved in splitting folds. For convenience, a PDF document has also been deposited alongside each ENCODE ID to guide the reader into the experimental protocols and computational steps involved in generating the processed data.

To aid in the reproducibility of benchmarking, we have now integrated **Supplementary Table 6** and **Supplementary Table 9** to provide readers a mapping of exactly which sequences were assigned to which fold as well as ML performance information with respect to each fold. These tables are now referenced in the **Methods** sections described above. The matched processed MPRA data for each of these folds is presented in **Supplementary Table 4** and **Supplementary Table 8**. A downstream user attempting to benchmark their own models can now refer to these tables to compare their performance head-to-head with ours. These tables have now also been included in the Zenodo requested below.

- There is a gap between the major issues in the field and what is addressed in this paper. The presentation should be a bit more clear on what is addressed and those speculations of what could be addressed with this technology should be in discussion. For instance, in the introductory paragraphs, they highlight a major open issue in the field is our poor understanding of disease associated variants. The article does not address this issue directly.

We apologize if the introduction provided a misimpression of the contents of the manuscript. We should note that the first paragraph deals with general regulatory element challenges to allow the reader to understand the difficulties the field faces in general (including linking elements to target genes), while the second provides challenges relevant to this article. We have gone through the introduction carefully, taking these comments in mind, and significantly revised it, in particular relating to the challenges our work is trying to address. In addition, as previously mentioned, we have added an entire paragraph to the discussion section listing the limitations of our work. We have also added new analyses to address the question about disease-associated regulatory variant identification to complement our prior figures on several disease-associated enhancers/promoters whose sequences were tested with saturation mutagenesis coupled to MPRA.

- The authors should deposit all “processed data” for the training, validation, testing, along with the model weights on zenodo. Also a static snapshot of the code.

All raw and processed data has now been deposited in the ENCODE portal. We have also deposited all pre-trained model weights (i.e., for our new MPRALegNet model), training code (i.e., for both MPRA and our new MPRALegNet model), and *in silico* mutagenesis interpretation code in Zenodo, along with a static snapshot of the Github code used to train the models. A link to this Zenodo (<https://zenodo.org/record/8219231>) is now included in the main text to allow readers to easily download and run the models.

Minor issues:

- I don't follow the rationale for why WTC11 cells had a cohort of transcriptionally active genes whose promoters were inactive in our MPRA (extended Data fig 4b), suggestive of a shift towards a euchromatic promoter state in this cell line. Could this also be that these promoters are not a sufficient size for certain genes (i.e. not as compact)?

This comment inspired us to evaluate the reason more deeply that WTC11 cells appeared to have a larger cohort of transcriptionally active genes relative to expression levels reported by the MPRA. We investigated the possibility that the high preponderance of alternative promoters in WTC11 might explain this observation. Thus, we used CAGE-seq data from the FANTOM consortium, which can be matched at much higher resolution to the precise promoter tested in our MPRA. We observed greater alignment between MPRA data vs CAGE-seq relative to MPRA data vs RNA-seq, with correlations consistent among all cell types. We have now added **Extended Data Fig 3g-i**, reproduced below, with corresponding results text modified:

Extended Data Fig 3g-i. Relationship between MPRA-measured activity (x-axis) to gene expression, as measured by CAGE signal at the tested promoter.

'Interestingly, in WTC11 cells, we found a larger cohort of transcriptionally active genes whose promoters were inactive in our MPRA (**Extended Data Fig. 3b,f**). Additional analysis revealed that this observation could largely be explained by the use of alternative promoters in WTC11 cells, as the CAGE signals in the precise promoters tested were congruent to MPRA activity in a similar manner among all three cell types (Pearson $r \approx 0.60$, **Extended Data Fig. 3g-i**).'

- I don't agree with this statement that "putative enhancers span the range in between both extrema" as a description of Fig 1d. I only agree with the promoters, not the enhancers.

Our sentence is unfortunately written in a grammatically ambiguous manner that can be interpreted multiple ways: it could mean that i) the majority of the density of the distributions of promoters and enhancer scores lie within the greatest extremes of the control distribution [our intended meaning], or ii) the "extrema" allude to the control distributions themselves, in which case it is not true that the enhancer distribution largely lies between the control distributions [the reviewer's alternative yet valid interpretation]. We have now changed the text to read:

'Averaging across the three replicates, we observed that the distribution of element activity scores was strongly divergent between most positive and negative controls (**Supplementary Fig. 1e**). An exception to this trend was observed for controls derived from CRISPRi-based screening efforts^{29,30}, which were initially chosen for testing based upon their favorable epigenetic context (e.g., strong H3K27ac activity). In these cases, negative controls (*i.e.*, which displayed no significant association to a target gene) exhibited only a slightly weaker signal than positive controls (*i.e.*, which displayed significant association to a target gene), indicating that, in our reporter assays and outside their epigenetic context, they were still capable of activating transcription relative to dinucleotide shuffled negative controls (**Supplementary Fig. 1e**).'

- Table references are not all correct.

We are not sure which table(s) the author is alluding to. We double-checked each reference to a supplementary table throughout the entire text and did not find an apparent incorrect reference.

- It would be nice to get quantitative values in table form of the performance of each model.

To enable future reproducibility and benchmarking, we have now integrated **Supplementary Table 6** and **Supplementary Table 9** to provide readers both a mapping of which sequences were assigned to which fold, a summary of the target value(s) predicted, as well as the performance of each model on each of the ten folds, for each cell type examined and for every approach tested.

- Can the authors add analysis similar to that in Figure 4 for other elements in the saturation mutagenesis dataset (corresponding to K562)?

We have now integrated **Extended Data Fig. 9** to reflect experimental data from saturation mutagenesis of all loci examined in the study, alongside MPRALegNet predictions and TFBS annotations.

- Why are negative controls in Fig 5c for WTC11 significantly different from 0?

We have now added the following explanation in the text to indicate a possible reason for this observation:

'A possible explanation for the stronger activity of negative controls in WTC11 cells could be that stem cells tend to exhibit a more globally open chromatin state⁵⁶, which may make them susceptible to greater levels of background transcription relative to other cell types.'

- The written presentation is a bit repetitive, especially with the pilot experiment, CRE characterization experiment, and joint library experiment.

We have carefully gone through the text and revised to make it less repetitive.

Author Rebuttal To First Revision:

RESPONSE TO REVIEWERS

Editor comments

We feel that it is very important to address the remaining technical concerns, presentation issues and overstatements that were brought up by reviewers #1 and #3/4 in a revised manuscript.

We want to thank the editor and reviewers for their comments which we feel have significantly improved our revised manuscript. Below in blue font we provide a point-by-point response to these comments.

Referee #1

The authors made revisions that addressed some of my previous comments, but my overall assessment of this manuscript that I gave in my original review has not changed substantially.

The authors toned down their claims in some places about providing an exhaustive catalog of CREs, but I still do not think the authors results are consistent with what they are stating as their overall conclusion:

“In summary, notwithstanding these caveats, our work provides a nearly comprehensive catalog of functional regulatory elements in three established cell lines that are widely used in biological research: HepG2, K562, and iPSCs.”

Considering the authors only tested a small fraction of genomic sequences and extrapolating from the figures they show in response to points 3 and 4, I suspect if they tested more of these sequences the authors would call far more functional CREs in a cell type than they are currently reporting or is generally appreciated.

We agree with the reviewer that there could definitely be additional sequences that we are missing in our analyses, both computational and experimental. To address this, we went carefully through the article and revised the text to both mention this and tone down the ‘nearly comprehensive’ terms we used throughout. We also added the following sentence to the discussion:

‘It is important to note that while the number of tested sequences are high, there could be several additional CREs that were missed in our annotations and subsequently in our assays, due to our selection criteria, additional CRE annotation assays/marks/tools that were not used, technical issues of the biochemical assays used to select sequences and other factors.’

Given their results are challenging current dogma in terms of validating sites without DNase signal in the cell type, as the authors note in response to point 3, I am questioning why the authors are choosing to relegate this dogma challenging result to a supplementary figure. This also raises questions of whether the result of validating 41.7% of cCREs to be functional in the abstract might not be as meaningful as might be interpreted.

Our initial inclination was to be guarded about the meaning of this finding but concur with the point raised that the finding has important implications that are worth highlighting. We have now integrated one of the panels of our previous supplementary figure into the main text as **Fig. 5b**, and emphasized this result further in the **Results** section:

“For instance, we found that sequences with nearly absent DNase signal in K562 cells, but high DNase signal in both HepG2 and WTC11 cells, could still lead to high MPRA activity in K562 cells (**Fig. 5b, Supplementary Fig. 8a**).”

We have also added the following text to the discussion mentioned in the point above to further clarify that our 41.7% statistic should be narrowly interpreted to represent the subset of open chromatin peaks:

‘It is important to note that while the number of tested sequences are high, there could be several additional CREs that were missed in our annotations and subsequently in our assays, due to our selection criteria, additional CRE annotation assays/marks/tools that were not used, technical issues of the biochemical assays used to select sequences and other factors.’

Given the authors are still validating a substantial number of sites with relatively low DNase signal particularly in K562 and HepG2 and there are many more sites below the calling threshold not being tested I think this suggests many additional sites if tested would likely be considered functional based on their procedures. I think these results should be disclosed to the readers and preferably also with a quantitative statistic associated with them.

As mentioned in detail in the comment below, we have added a new supplementary figure (**Supplementary Fig. 8b**) showing a point estimate of the proportion of active elements as a function of DNase signal in each of three cell types along with results text for this.

With respect to the authors point about a 5% false discovery rate, they could do a more local FDR estimate based on the DNase signal.

Thank you for this suggestion, which helps to further reinforce the points above. We have now added the following as **Supplementary Fig. 8b**:

Supplementary Fig. 8b, Point estimate of the proportion of active elements (*i.e.*, exceeding the signal for shuffled negative controls with a 5% FDR) as a function of DNase signal in each of three cell types. Each point along the x-axis reflects this proportion for elements with a DNase signal in a range of $(x-0.1$ to $x)$.

As well as the associated results text:

'Further reinforcing this observation, we observed ~15-25% of elements lacking DNase signal were more active than shuffled negative controls (5% FDR), although increased DNase signal was clearly associated with an increased proportion of active elements (**Supplementary Fig. 8b**).'

In response to my comment on not applying the approach to hard to transfect cell lines, the authors mention other MPRA papers for hard to transfect samples, but the novelty of this paper is the scaled up-version of the assay which still hasn't been demonstrated in hard to transfect cell lines as far as I understand.

We have used this assay in its scaled-up version in a variety of cell lines, including hard to transfect cell lines such as iPSC-differentiated neurons (over 80,000 sequences; Kosicki et al. *BioRxiv* 590634, 2024), human primary neurons and even cerebral organoids (over 60,000 sequences per library; see Deng et al., *Science* 2024 PMID 38781390). As this article is part of the ENCODE project, we were required by the consortium to use ENCODE cell lines, and hence these experiments were carried out in HepG2, K562 and WTC11, the latter of which is also considered hard to transfect. However, we completely agree with the reviewer that we need to better relate our text in the introduction to what was done in this article. We thus have significantly revised the introduction, see details in comment later below. In addition, we have added references to these two aforementioned articles in the introduction:

'To address these and other limitations of episomal MPRA, we previously developed a lentivirus-based MPRA (lentiMPRA) that enables reproducibility and multiplexability, the ability to carry out these assays in 'hard to transfect cell lines', such as neurons or organoids^{19,20}, and provides an 'in genome' readout.'

I agree there were technical differences between the author's assay and other approaches that were reported in their previous paper. I am less clear if there is a main result being reported in this paper that was specifically enabled because of one of those differences.

The major difference is scale, allowing us to test a significantly large number of sequences using this assay. This further allowed us to have a large dataset that was used to build various improved machine learning models, as shown in the manuscript. In addition, it also allowed us to better characterize cell-type specificity. We should also add that since our *BioRxiv* publication of this article, numerous groups have used these data to improve their machine learning tools to both predict regulatory activity and tissue-specificity, highlighting the importance of these datasets for improving regulatory grammar, characterization, variant analyses and many more.

I still think the introduction could give a misimpression of the contents of the manuscript since it mentions a set of limitations, a number aren't addressed in the manuscript and is not explicit in the introduction on which ones are being addressed in the manuscript.

We have significantly revised the introduction to only highlight the limitations which our work addresses. The following two first paragraphs now read:

'Sequence variation in gene regulatory elements is a major cause of human disease¹⁻³. For example, the overwhelming majority of genome-wide association studies (GWAS) for common disease unambiguously implicate noncoding haplotypes that contain distal gene regulatory elements, such as enhancers, in numerous disorders⁴⁻⁷. However, identifying, characterizing, and/or predicting the functional effects of nucleotide variation in gene regulatory elements remains challenging. One of the major limitations is the lack of a comprehensive functional delineation of the likely millions of regulatory elements in the human genome, many of which have tissue/cell type specific activity. This impediment also limits the ability to develop machine learning tools that can predict tissue-specific regulatory element activity with high-precision.

The emergence of genome-scale biochemical technologies to globally catalog regions of open chromatin (DNase-seq, ATAC-seq), transcription factor (TF) binding, histone modifications (ChIP-seq or CUT&RUN) and mRNA expression levels (RNA-seq) has provided a framework to investigate gene regulatory and transcriptional landscapes in hundreds of cell types^{8,9}. These efforts have led to the discovery of millions of cCREs in the human genome. However, these approaches are descriptive, providing the potential for a sequence to be a cCRE, but not a functional assay or readout that tests this. For example, DNase I hypersensitivity will not distinguish between spurious and functional binding events of a given TF, nor does it inform the functional identity of a regulatory element (e.g., active vs. repressive), its magnitude of activity, or the gene(s) that it regulates.'

The authors misinterpreted my suggestion for doing standard TF motif-scanning enrichment analysis. For this I meant using a set of known motifs for motif scanning and enrichment and not performing de novo motif discovery as the authors did in response. A TF motif-scanning enrichment analysis approach would be simpler and allow a more direct comparison across cell types not confounded by the stochasticity of motif discovery procedures. Furthermore, it could

give a more quantitative interpretation to the motif results. As currently it is hard to interpret the relative importance of individual motifs among the large set of discovered motifs the authors are presenting.

Thank you for clarifying the meaning of the initial request. We concur that your recommendation of using a known set of motifs would add complementary insight into the analyses presented. We have now added **Fig 2f-g** to indicate the enrichment of known motifs present in the HOCOMOCO v12 database, in either the top vs. bottom promoters or potential enhancers in each of the three cell types, reproduced below:

Fig. 2f-g, Volcano plots indicating the enrichment of HOCOMOCO v12-derived³⁹ TF families in the top vs. bottom 1,000 promoters (f) and potential enhancers (g), as ranked by MPRA activity within each of the three cell types. Enrichment (i.e., measured as an odds ratio) and Benjamini-Hochberg (BH) corrected q-values were computed using Fisher's exact test. Significant families above the p-value acceptance threshold (dotted horizontal line) are labeled with one representative TF family member and its general TF family in parentheses. A complete table of TF families considered and corresponding results is provided in **Supplementary Table 5**.

We have also added several paragraphs to a new “Classical motif enrichment analysis” section in **Supplementary Methods** that further describes our methodology in detail, and a new **Supplementary Table 5** to report the full list of motifs searched and numerical values associated in **Fig. 2f-g**. A description of the table is provided here:

Supplementary Table 5. Annotation of motifs belonging to HOCOMOCO motif clusters, and representative motif for each cluster. Also provided are the \log_2 (odds ratios) and q-values indicating enrichment or depletion of each representative motif in the promoters or potential enhancers of each of the three cell types.’

And an excerpt of the amended **Results** text is provided here: “We performed two complementary analyses to gain further insight into which TF families might bind to highly vs. lowly expressed promoters and potential enhancers: i) an enrichment analysis using motifs annotated in the HOCOMOCO v12(Vorontsov et al. 2024) database (**Fig. 2f-g**, **Supplementary Methods**, **Supplementary Table 5**), and ii) a *de novo* motif enrichment search (**Extended Data Fig. 4**).”

The authors variant/GWAS analysis is similar to what has been done previously with sequence models trained to predict other biochemical assays. There is a disadvantage here in that the analysis is only provides cell type specific information for a limited number of cell types with their MPRA data compared to what has been done before for some other biochemical assays. Based

on the comparison with Enformer results there wasn't a clear advantage even with cell types for which the MPRA data was available (2 loci were better and 2 loci were worse).

Although we would have of course ourselves been pleased to see MPRALegNet perform consistently better than our prior work on Enformer, our main goal was to transparently benchmark our model to existing state-of-the-art models, and demonstrate that training a model based on MPRA data can provide an alternative path towards giving insight into the task of variant effect prediction as a model trained on ChIP-seq, epigenetic marks, and CAGE (e.g., the Enformer model). It is also important to note that Enformer consists of 249M parameters relative to our ~1.3M, leading to a ~200-fold parametric reduction. In general, Enformer is computationally intractable to run on a genome-wide scale relative to our more lightweight MPRALegNet model. We have now added some sentences to the discussion to summarize these points:

'Although MPRALegNet was trained upon three cell types, the similar performance of cell-type agnostic models to cell-type specific models in the variant effect prediction task⁴⁷, and observation of similar measured effect sizes of the same variants in multiple cell types⁵⁶, support its potential general use in additional cell types. Although MPRALegNet only performs competitively with Enformer on this task, its ~200-fold reduction of parametric complexity from Enformer's ~249 million to its ~1.3 million parameters provide a computationally efficient and practical way to rapidly compute predicted variant effects on a genome-wide scale.'

The authors are misinterpreting the results presented in the section "MPRALegNet predicts combinatorial TFBS effects" and reaching conclusions about the 'nature of combinatorial TFBS effects learned by MPRALegNet' that aren't justified by what they presented. In principle one could have a linear model with many features that could lead to the same types of results the authors are presenting even though a linear model does not learn combinatorial feature interactions. The issue is that the authors are making predictions with many features, but then evaluating the combinatorial effects with only two features at a time. It thus could appear the combinatorial term is making a contribution, but might not be in the context of all the features. Also from a biological perspective the biological results about the pairwise association of features with activity or correlation with the number of copies could have been produced directly from analyzing the observed data so it is not clear the added value of MPRALegNet in this analysis.

We concur with the point that both the TF copy number analysis and pairwise association results can be produced directly from the data (e.g., **Fig 3e-h**). Our goal in showing the patterns labeled "Observed" in these plots is to show that these nonlinear patterns exist in the data itself, while the goal of "Predicted" was not to claim that MPRALegNet is adding value in the interpretation of the observed data. Instead, our goal was simply to validate that MPRALegNet has itself learned nonlinear behaviors by learning these from the data during its training process, something that a classic linear model would be unable to do. The following lines in the text capture the motivations of the analysis: "To gain insight into the nature of combinatorial TFBS effects learned by MPRALegNet, we further examined the top ten most abundant activating TFBS motifs detected in each cell type." and "Collectively, MPRALegNet was able to model nonlinear interdependencies between TFBS combinations in all cell types."

Although we took careful steps to account for confounding variables in the original analysis, we acknowledge the reviewer is correct that it is possible that features not considered in the original approach could dilute the combinatorial effects observed. We have therefore performed the analysis again considering these additional features; please see our response to **Reviewer 3, Major comment 2**, which explains the confounding variables we considered while addressing both this comment and Reviewer 3's concern that our model didn't account for variable binding affinities of the same TFs. Briefly, we have now constructed a linear model considering *all* TFBSs as well as their respective binding affinities, not just the top 10 considered before in a binary sense. We have repeated the analysis after correcting the data for effects induced by any other TF, after which we found the results to be nearly the same as before. Thus, our initial results, *i.e.*, the detection of interaction terms which reflect super-multiplicative or sub-multiplicative behavior, have been revised but are robust to the presence of all other TFs as well as the strength of their binding.

The authors added a sentence 'Considering our activity scores alongside Activity-by-contact (ABC) scores³⁴, we observed that our scores only subtly improved performance in the task of discriminating significant from non-significant enhancer-promoter interactions in these three studies (data not shown).' – I think the authors should show these results so a reader can directly assess this subtle improvement being claimed.

We have adjusted the text as follows: 'Considering our activity scores alongside Activity-by-contact (ABC) scores³⁶, we observed that our scores only subtly improved performance in the task of discriminating significant from non-significant enhancer-promoter interactions in two of these three studies (**Supplementary Fig. 3b**).' and updated the corresponding figure and its legend accordingly, integrating the results we previously provided in the rebuttal but omitted in the last manuscript revision:

Supplementary Fig. 3: Relationship between MPRA and CRISPRi datasets. a, Fraction of elements validated by CRISPRi (*i.e.*, called to significantly regulate a target gene) for equally sized bins of MPRA activity. The bins with the highest activity level exhibit a greater likelihood for validation. The left and right numerical intervals of the MPRA activity bins are shown comma-separated, with open parentheses meaning the exclusion of the left interval and square brackets meaning the inclusion of the right interval. Three CRISPRi datasets were cross-referenced for this analysis^{12,14,15}. **b**, Using the datasets in part (a), we intersected our set of potential enhancers

with those tested by CRISPRi. We then trained a binary classifier, using logistic regression to predict whether a specific enhancer-gene linkage was statistically significant, considering the ABC model along with an interaction term with MPRA activity. The tables shown indicate the coefficients, z-value, and statistical significance of each term. The Fulco et al., 2019 and Schraivogel et al., 2020 datasets exhibited support for at least one MPRA term while there was no evidence for the benefit of this term in the Gasperini et al., 2019 dataset.

Referee #2

I commend the authors on their thoughtful and detailed response to the feedback of myself and the other reviewers. They now apply a new neural network to improve regulatory element identification, interrogate cooperative and dose-dependent regulatory behavior, and improve fine-mapping of seven different human diseases. Moreover, they cleverly apply existing CRISPRi perturbation studies to validate significant enhancer–promoter interactions identified as well as transcription factors implicated by their study. Additionally, the textual revisions clarify the methods and analyses applied throughout, and also discuss in detail the limitations of the results presented. This manuscript is a technical tour de force and a joy to read! I am happy to recommend this manuscript for publication at Nature at this time.

We thank the reviewer for their valuable insights which improved the quality of our manuscript and are glad to hear that we have adequately addressed all points.

Referee #3/4

Major concerns:

- The biochemical features predict regulatory activity section doesn't seem to add much to the overall story. It's a statement that yes we can do it, but we don't gain much insights (due to colinearity). The additional experiments using a universal set don't seem to be so modest (Supp Fig 4D) and thus the conclusions drawn from it seem to be reaching. Since no insights can be garnered from the biochemical regression model, I suggest removing this section from the main text and only including the model's performance within the section "Sequence-based models predict regulatory activity with higher precision" as a baseline for comparison with the deep learning models.

Thank you for the feedback. We have now moved the figure of the biochemical model from **Fig. 3** to **Supplementary Fig. 4**, and the corresponding main text to a new "**Supplementary Results**" section. We still believe this analysis is worth mentioning in the main text as a preface to the deep learning model, but following this comment have significantly shrunk the text describing this model to a single paragraph in the section suggested.

- Regarding the section "MPRALegNet predicts combinatorial TFBS effects" details on the observed values should be provided in the main text to better appreciate what is being compared in the figures. Importantly, the combinatorial TFBS effects analysis is based on observational analysis of motifs in the sequences and is thus riddled with confounding factors. The analysis is not measuring dosage or combinatorial features learned by the model because of the other factors that can influence a sequence beyond the motifs that were scanned by FIMO. Moreover, each motif scanned by FIMO has a range of affinities, when treated as a binary value for presence or not, it introduces another confounder to this analysis. This analysis is simply showing prediction agreements for subsets of the sequences, whose correspondence was already shown as part of the test performance. Uncovering TFBS combinatorial rules requires more carefully designed interventional experiments. Without properly controlling for confounding, the statements about combinatorial TFBS effects cannot be made from this analysis.

In our last iteration, we took several measures to account for confounding effects, as explained in these excerpts from the **Methods**: 'For homotypic analysis, we plotted the median element activity (i.e., both predicted and observed) for elements possessing 0, 1, 2, 3, 4, or 5 motifs, filtering away elements with ≥ 1 site to any other TF to reduce the chances of a confounding effect. Groups with a sample size of < 10 were also filtered out to minimize the impact of noise. For heterotypic analysis, we evaluated every pair of the 10 motifs, isolating cases in which the element possessed 0 counts of both TFs, 1 count of one TF or the other, or 1 count each of the first and second TF. Again, all elements were filtered to those with ≥ 1 site to any other TFs other than the TF pair.'

We believe this filtering scheme is already heavily accounting for confounding effects by making inferences on subsets of the data while maintaining sufficient sample sizes to observe trends with median statistics. Regrettably, this removal of elements with binding sites towards other TFs was

not clearly explained in the main text, possibly leading to some confusion, something we have now correct in the revised text. This being said, we acknowledge that there are several paths to further sharpen the analysis and assess its robustness, including the suggested consideration of binding affinity. By default, FIMO uses a p-value threshold of $1e-4$ to consider a sequence a hit. We observed that there is a sensitivity/specificity tradeoff as this threshold is loosened, with thresholds below $1e-3$ leading to many spurious hits and false positives, in turn leading to misleading results. In addition, Reviewer 1 felt that the confounding aspect of the analysis was rooted in the possibility that a linear model, integrating the effects of many TFs, could potentially account for apparent nonlinearities we have observed. To account for this comment regarding binding affinities as well as Reviewer 1's comments, we have now constructed a linear model to globally correct the data while accounting for variable binding affinities. This approach is explained in the heavily revised **Methods** section:

*“Modeling dose-dependent and combinatorial motif effects learned by MPRALegNet. A non-redundant set of positional weight matrices (PWMs) from each cell type, as ranked by TF-MoDISco-lite (**Extended Data Fig. 6**), were extracted and scanned (using FIMO 5.5.4⁸⁰, parameters “--text --thresh 0.001”) along each promoter and potential enhancer that was tested bidirectionally. The motif scans were summarized into a matrix of counts for each TF and element tested, as well as log-likelihood [*i.e.*, sum of the log(probabilities)], reflecting the likelihood of a given TF binding the element while considering all TFBS instances in both orientations and their respective binding affinities. We then performed an analysis of homotypic (*i.e.*, dose-dependent effects for a single TF) as well as heterotypic (*i.e.*, combinatorial effects among pairs of TFs) for the top 10 activating TFs of each cell type.*

For homotypic analysis, we plotted the median element activity (*i.e.*, both predicted and observed) for elements possessing 0, 1, 2, 3, 4, or 5 motifs, filtering away elements with ≥ 1 site to any of the other top 10 TFs to reduce the chances of a confounding effect. Groups with a sample size of < 10 were also filtered out to minimize the impact of noise. The expected dose-dependent responses (e.g., dotted lines shown in Fig. 3f) were computed using linear regression models examining the relationship between either the observed or MPRALegNet-predicted MPRA activity and the number of TFBSs, given log-transformed and untransformed space to model either multiplicative or additive effects, respectively. The expected trend for multiple sites was extrapolated based upon the slope and intercept terms of these linear models.

For heterotypic analysis, we evaluated every pair of the 10 activating motifs, isolating cases in which the element possessed 0 counts of both TFs, 1 count of one TF or the other, or 1 count each of the first and second TF. Again, all elements were filtered to those with ≥ 1 site to any of the other top 10 TFs other than the TF pair considered. To further account for confounding effects that could be attributable to all *other* TFs (*i.e.*, including those beyond the top 10), we computed the residuals from a linear model which considered the log-likelihood values for all other TFs besides the pair of TFs under consideration. We call these “Adjusted $\log_2(\text{RNA/DNA})$ ” (e.g., y-axis of **Fig. 3h**) because they removed variability explained by the binding affinities and occurrences of other TFs. Finally, a regression model was fit independently to the predicted and observed activity scores. This model sought to predict activity as a function of the presence of TF1, TF2, or an interaction term (*i.e.*, TF1*TF2). The coefficient for the interaction term

represented the strength of the super-multiplicative effect (*i.e.*, if the coefficient was positive) or the sub-multiplicative effect (*i.e.*, if the coefficient was negative)^{48,49}.

We have repeated the analysis after correcting the data for effects induced by any other TF while considering their binding affinities, after which we found the results to be nearly the same as before. Thus, our initial results, *i.e.*, the detection of interaction terms which reflect super-multiplicative or sub-multiplicative behavior, have been revised but are robust to the presence of all other TFs as well as the strength of their binding. An example of the results of the revised analysis is shown for **Fig. 3g-h**.

Fig. 3g-h: **g**, Heatmap indicating interaction term coefficients reflecting super-multiplicative and sub-multiplicative effects, shown in red and blue, respectively, when considering elements possessing the indicated pair of heterotypic TFBSs in HepG2 cells. Coefficients fit to observed and predicted values are shown in the upper and lower triangles of the heatmap, respectively. **h**, Relationship of the distributions of observed and MPRAlegNet-predicted element activity scores in HepG2 cells for the subset of elements possessing zero TFBSs, one TFBS corresponding to either the ATF3/FOS::JUN or FOXD2 families, or one TFBS for both TF families, adjusted for potential confounding effects induced by the presence of other TFs (**Methods**). The horizontal line corresponds to the expected activity score under a multiplicative model.

- Previously the following issue was stated in our review (in major point 1): "This approach essentially measures the functional effect of sequence elements averaged across different random contexts, which should be taken into account when drawing conclusions from the results. It is important to note that this aspect is not explicitly stated in the text. "The point is not whether or not a constant integration site through a safe harbor (single integration site) is better or not compared to random integration. The point is rather that the authors have to take this into account and make sure they acknowledge that the results do not necessarily agree with those expected from sequences in their native contexts (and promoters). The experiment marginalizes out the

effects of the genomic context and it is unclear how this is different from episomal assays in essence. This is not sufficiently addressed in the revised text.

We have added the following text to the introduction:

'As lentivirus integrates at different locations throughout the genome, lentiMPRA measures the functional effect of cCREs averaged across different genomic locations.'

Minor concerns:

- The introduction is still misleading. The last paragraph is relevant to the study but the rest of the introduction is a broad overview. It seems like the text could benefit from shrinking the 'review of the field' part (i.e. the first 2 paragraphs of the introduction) and making it clear what limitations the paper is going to address and what it discovers. It would help to guide the reader with the exact goals of the study, e.g. test orientation effects, characterize cell type specific factors in promoters and enhancers (more similar to the text in the last paragraph of the introduction).

We have significantly revised the first two paragraphs of the introduction following this comment. It now reads:

'Sequence variation in gene regulatory elements is a major cause of human disease¹⁻³. For example, the overwhelming majority of genome-wide association studies (GWAS) for common disease unambiguously implicate noncoding haplotypes that contain distal gene regulatory elements, such as enhancers, in numerous disorders⁴⁻⁷. However, identifying, characterizing, and/or predicting the functional effects of nucleotide variation in gene regulatory elements remains challenging. One of the major limitations is the lack of a comprehensive functional delineation of the likely millions of regulatory elements in the human genome, many of which have tissue/cell type specific activity. This impediment also limits the ability to develop machine learning tools that can predict tissue-specific regulatory element activity with high-precision.

The emergence of genome-scale biochemical technologies to globally catalog regions of open chromatin (DNase-seq, ATAC-seq), transcription factor (TF) binding, histone modifications (ChIP-seq or CUT&RUN) and mRNA expression levels (RNA-seq) has provided a framework to investigate gene regulatory and transcriptional landscapes in hundreds of cell types^{8,9}. These efforts have led to the discovery of millions of cCREs in the human genome. However, these approaches are descriptive, providing the potential for a sequence to be a cCRE, but not a functional assay or readout that tests this. For example, DNase I hypersensitivity will not distinguish between spurious and functional binding events of a given TF, nor does it inform the functional identity of a regulatory element (e.g., active vs. repressive), its magnitude of activity, or the gene(s) that it regulates.'

- the new addition to paragraph 1 on page 10 seems to indicate that lentiMPRA assay does not capture the cell type specific trans factor effects and essentially measures the promoter activity directly, marginalizing out context effects and not incorporating the trans effects. If this is the case the authors should make it clear that the conclusion that promoters are not cell type specific refers to their intrinsic activity.

Thanks for this suggestion. To better clarify and address this, we have added the following text to this paragraph:

'Collectively, these results indicate that core promoters possess weak orientation dependence and when tested individually have little cell-type specificity.'

- On page 10 paragraph 2 -- a correlation drop from 0.55 to 0.43 does not seem 'modest'. This indicates that if we marginalize promoter context and remove trans effects the transcriptional activity will only have a correlation of 0.43; it does not strongly predict gene expression levels.

We have now removed the term "modest" as it is admittedly imprecise. The text now reads: "Removing all non-expressed genes led to a reduction in the correlation between MPRA measurements of promoter activity and endogenous expression levels (Pearson $r \approx 0.43$, **Extended Data Fig. 3c-f**)".

- The format of figure 2d is still confusing. Visually it would be easier to understand this as a beeswarm plot (similar to Fig 2a), as it is now the reader has to decipher the pearson correlation values from the size and color of the circles which would be more straightforward to do with a y axis scale. Although I can see a pattern of alternating big and small circles it is unclear how big this difference is (in relation to the other effect sizes discussed in the paper).

While we feel that the current plot allows one to more easily see the pairwise structure of all comparisons in the limited space available, we agree that a plot indicating the y-axis value would complement this figure to make the y-axis values clearer. Therefore, we have now added **Extended Data Fig 3d**, reproduced below, to show all pairs of comparisons:

Extended Data Fig. 3: Properties of promoter activity in three cell types. d, Alternative representation of the data shown in **Fig. 2d** and panel (c), showing the Pearson correlation between each pair of

measurements indicated below the horizontal line. Black points represent all genes (*i.e.*, akin to **Fig. 2d**) and red points represent the expressed subset of genes [*i.e.*, akin to panel (c)].

- With regards to the promoters having more strand specificity than enhancers. Figure 2c is the only evidence of such a difference and it is based on absolute values. If anything, both enhancers and promoters show comparable strand asymmetry.

We agree that the difference is not as strong, but it is statistically significant and in line with previous findings (Klein et al, *Nature Methods* 2020). To address this, we revised the following sentence:

‘Consistent with prior work^{40,41}, we observed that promoters displayed slightly stronger strand asymmetry effects relative to potential enhancers in all cell types examined (**Fig. 2c**)’

- "Both MPRAnn and MPRAlegNet underwent optimization procedures to detect hyperparameter and data augmentation settings that improve model performance " Only description of MPRAnn is in the supplementary methods.

We have included a new section dedicated to describing the particular changes of MPRAlegNet architecture, including details of the alternative augmentation strategies (as shown in **Extended data Fig. 5b**), in the updated **Supplementary Methods**.

- A statistical test to compare enhancer or promoter distributions to negative controls for Fig 1D should be performed.

Thank you for the suggestion. We have now added the requested statistical tests and adjusted the legend to reflect the test accordingly:

‘**d**, Violin plots of element activity, measured as $\log_2(\text{RNA/DNA})$ ratios, for cCREs, negative controls, and positive controls for each library. Promoter and enhancer distributions were compared against the “shuffled” category using a one-sided Wilcoxon rank-sum test, followed by a Bonferroni multiple hypothesis testing correction ($*p < 1e-8$).’

- For figures 4f and extended data fig 7e it would be helpful to plot out the theoretical values of purely multiplicative effects based on the single motif effects. Also, the multiplicative pattern should be supported with a linear fit. It seems to be plateauing. A direct comparison with an additive model would also make it more compelling. It seems that all dosage curves saturate, indicating a more sigmoidal trend. Also, the number of sequences for each number of sites should be presented, I presume sequences with 5 of the same motifs occurs much less than the number of sequences with 1 motif.

Thank you for the suggestion. We have followed it and recorded the sample size in each plot along with the expected outcomes if either a multiplicative or additive model were considered. We concur that a saturation effect seems apparent with high TFBS dosage and had noted this in the

main text as such: ‘However, several TF families, such as STAT (considering the STAT1/4/5A/5B motif) in K562 cells and ETS-related (considering the ELK4 motif) in WTC11 cells, displayed sub-multiplicative patterns in the highest dosages (Fig. 3f, Extended Data Fig. 7e), indicative of a saturating expression effect’. Please find the updated figures below.

Fig. 3: Sequence-based models predict regulatory element activity. **f**, Relationship of the distributions of observed and MPRALegNet-predicted element activity scores in K562 cells for the subset of elements possessing 0 to 5 TFBSs corresponding to the STAT1/4/5A/5B TF family, along with the expected dose-dependent effect in the scenario of either a multiplicative or additive model. The number of elements represented in each group is indicated above the plot.

Extended Data Fig. 7: Combinatorial TFBS effects learned by MPRALegNet. **b-e**, These panels are arranged in the same scheme as Fig. 3e-f, except display results for homotypic TFBSs in HepG2 (**b-c**) and WTC11 (**d-e**) cells for the indicated TF families.

- there seems to be a typo in figure 5b OR labels (extra decimal place dot)

We believe that the values are correct; however, we were not precise enough in the figure legend regarding how they were computed. The OR values denoted on the original plot were obtained with Fisher's exact test on 2x2 contingency tables (R implementation, conditional maximum likelihood estimate) built by counting events (concordant and discordant predictions) in different quadrants of the plot independently. We have modified the legend to articulate this a bit more clearly. It now reads:

'The corresponding Odds Ratio (OR) is also indicated, and was computed with a Fisher's exact test using the 2x2 contingency tables provided in **Supplementary Table 8**.'

- in supplementary fig 7 what do the top right corner numbers indicate?

Thank you for noticing this apparent error. The scale (which was auto-generated using the R 'psych' package) seems unnecessary and has now been removed as it does not contribute to the figure interpretation.

- The term modest seems to range in effect sizes across different examples depending on whether it supports the results or not. There is a lot of work done here and stating the results as is could be more beneficial than to obfuscate it with murky, imprecise language.

Thank you for this feedback. Our intention was to use the term 'modest' as the colloquially used meaning 'weak' but agree that it's an imprecise term. Thus, we've now replaced the following excerpts accordingly:

'Consistent with prior work^{40,41}, we observed that promoters displayed weakly ~~modestly~~ greater strand asymmetry effects relative to potential enhancers in all cell types examined (**Fig. 2c**)'

'Collectively, these results indicate that core promoters possess weak ~~modest~~ orientation dependence and little cell-type specificity.'

'Comparing MPRALegNet predictions for the PKLR promoter to MPRA data revealed that most of the relevant TFBSs (e.g., GATA3, KLF9, SP5, and NFIB) could be detected, although the predicted effect sizes were relatively smaller ~~modest~~ for KLF4 and GATA2 (**Fig. 4c**).'

'We observed the distribution of element activity scores to be strongly divergent and weakly ~~modestly~~ cell-type-specific between positive and negative controls in each cell type (**Supplementary Fig. 6f**).'

'The performance was only weakly ~~modestly~~ better for models trained on the universal feature set, indicating that bias induced by variable feature size was minimal (**Supplementary Fig. 4f**).'

‘For instance, we found that sequences with nearly absent DNase signal in K562 cells, but high DNase signal in both HepG2 and WTC11 cells, still could lead to ~~modest to~~ high MPRA activity in K562 cells (**Supplementary Fig. 8**).’

‘These scores recapitulated the expected patterns of enrichment or depletion of element activity for different element categories, with HepG2 and K562 controls showing strong relative activity in their respective cell types; potential enhancers showing strong relative activity in their respective cell types; and promoters and negative controls showing weakly ~~modestly~~ stronger activity in WTC11 cells relative to others (**Fig. 5c, Supplementary Fig. 9**).’

‘An examination of the gene expression levels of these TFs in our three cell types revealed that HNF4A/G, GATA1, TAL1, KLF1, SPI1, POU5F1, SOX2, and ZIC2/3 exhibited strong cell-type-specific expression in the expected cell types; additionally, CTCF showed weakly ~~modestly~~ enriched expression in WTC11 cells (**Supplementary Fig. 10**).’

Author Rebuttal To Second Revision:

RESPONSE TO REVIEWER #1

Below in blue font we provide a point-by-point response to Reviewer 1's comments:

Previously I raised issue with the authors reaching this conclusion:

“In summary, notwithstanding these caveats, our work provides a nearly comprehensive catalog of functional regulatory elements in three established cell lines that are widely used in biological research: HepG2, K562, and iPSCs”

In response the authors noted they added this sentence to the discussion instead:

‘It is important to note that while the number of tested sequences are high, there could be several additional CREs that were missed in our annotations and subsequently in our assays,...’

I do not consider this conclusion to be fundamentally different than the author's previous conclusion. The word ‘several’ implies that not many CREs were missed, but I still think there are many sequences that would show activity according to their thresholds if actually tested in the assay.

We have revised the manuscript to read: ‘there could be many additional CREs that were missed’

The authors may have misinterpreted my point with respect to the 41.7% statistic in the abstract. My concern with the 41.7% statistic given in the abstract (‘...and found 41.7% to be functional.’) is that people might interpret ‘functional’ to have a stronger meaning here than it traditionally has, particularly given the questions around why the authors assay also considers many sequences to be ‘functional’ that do not have DNase support in the tested cell type. The authors response was pointing out the added sentence about how they may have missed some additional CREs.

We have revised the abstract to read: ‘and found 41.7% to be active.’

As part of the authors response to my previous point about not showing that the technical differences of the authors assay compared to other assays impacted the main results in this paper, the authors respond:

‘The major difference is scale, allowing us to test a significantly large number of sequences using this assay’

However, it is not the case that the authors assay was significantly larger in terms of the number sequences tested than all previous applications of plasmid based MPRA/STARR assays.

To address this, we have the following sentences in the discussion:

‘Large-scale MPRA datasets are available for other cell lines⁶⁻⁹. However, they are primarily tested via episomal STARR-seq, require a very large number of cells, provide an episomal readout, and tend to use a strong promoter to increase the ability to detect activity⁷. In contrast, our modified lentiMPRA provides large functional datasets with an ‘in genome’ readout.

While the authors made some improvements to the introduction there are still issues with it.

In particular, the authors state:

“For example, DNase I hypersensitivity will not distinguish between spurious and functional binding events of a given TF, nor does it inform the functional identity of a regulatory element (e.g., active vs. repressive), its magnitude of activity, or the gene(s) that it regulates.”

and then claim:

“Massively parallel reporter assays (MPRAs) overcome these limitations by testing thousands of sequences/variants for their regulatory activity in a multiplex fashion.”

However, it is not the case that MPRA determines the target gene. Also MPRAs do not directly determine spurious vs. functional binding of a given TF directly, as it is testing a CRE sequence as a whole.

We removed the sentence: ‘For example, DNase I hypersensitivity will not distinguish between spurious and functional binding events of a given TF, nor does it inform the functional identity of a regulatory element (e.g., active vs. repressive), its magnitude of activity, or the gene(s) that it regulates.’

In response to my previous point about the authors not actually showing an overall improvement for variant effect applications relative to previous sequence-based models trained on other assay types which are already available in more cell and tissue types, the authors respond about the reduced model complexity of their model relative to Enformer. I did not consider this to be a satisfying response, even if one accepts that model complexity should be a consideration here, since the authors haven’t shown that the LegNet architecture or other existing sequence models that have far fewer parameters than Enformer trained on individual chromatin datasets do not achieve comparable performance to MPRALegNet in their evaluations.

The relevant excerpts in results and discussion are the following: “Collectively, we observed a correlation of 0.49 for *SORT1*, 0.65 for *PKLR*, 0.66 for *LDLR*, and 0.51 for *F9* between model predictions and observed data (**Extended Data Fig. 9**), confirming that MPRALegNet, despite being trained on cCRE activity, could partially model the regulatory effects of individual genetic variants. These results were comparable to those from Enformer³⁶ (0.63 for *SORT1*, 0.83 for *PKLR*, 0.62 for *LDLR*, and 0.28 for *F9*).”

and “While MPRALegNet only performs competitively with Enformer, its ~200-fold reduction of parametric complexity from Enformer’s ~249 million to ~1.3 million parameters provide a computationally efficient and practical way to rapidly predict variant effects on a genome-wide scale.”

We found that in two cases, we outperformed Enformer, and in two cases, Enformer outperformed MPRALegNet. It was not specifically our claim that MPRALegNet is superior to our previous work on Enformer, but rather our goal was to illustrate how the two tools are complementary approaches that can be used to address the variant effect prediction problem. Enformer is trained on natural genome-wide observations, while MPRALegNet is trained on functional element activity measured on a small fraction of the genome. Nevertheless, both tools are able to partially explain the magnitude by which genetic variants alter expression with varying degrees of success, depending upon the locus examined.

With respect to my points and those of Reviewer 3/4 about reaching conclusions about the combinatorial effects of TFs that aren't justified by their analyses, the authors attempted to control for some potential confounders. However, there still exists many potential confounders in terms of additional information MPRALegNet had access to that haven't been controlled for and that could explain the observations instead of true combinatorial effects.

In the last revision, we addressed the specific confounders Reviewers 1 and 3/4 were concerned with; specifically, the binding energy of the TF and additional TFs that were not initially accounted for. We did so by revising our analysis to: i) compute a log-likelihood score that accounted for the similarity of a motif match to the positional weight matrix representing the TFBS (thus modeling the binding energy), and ii) account for a comprehensive list of all TFs detected by MPRALegNet to play a role.

We apologize if this was not clearer, but MPRALegNet is a sequence-based model, such that we are not sure what the reviewer means by “additional information MPRALegNet had access to”. As a sequence-based model, MPRALegNet only had access to nothing more than the input sequences and MPRA activity results [$\log_2(\text{RNA/DNA})$ scores as labels] during its training process. To the best of our understanding, this means that MPRALegNet has the opportunity to only model linear and nonlinear dependencies among the set of TFs. Our approach was led by the thinking that adjusting and accounting for all linear dependencies was the path towards revealing the nonlinear dependencies between pairs of TFBSs, an approach similar to and inspired by past work we have cited (Grossman et al, *PNAS* 2017, see “Analysis of TF interactions” section of “Supporting Information”; and more recently, Kim et al., *Nat Genetics* 2022, see “Motif pair interactions” section in “Methods”).

This may address the reviewer's concern, but if not, to address it further we would need to know precisely which potential confounders the reviewer is referring to.

For the residual analysis that the authors added, I am somewhat surprised that conducting the analysis on the residuals had essentially no effect. Have the authors verified that the model they fit is able to get meaningful predictions? In particular, I am concerned the authors appeared to have used the ‘log-likelihood values’ for TFs for determining residuals, but then switched to a binary representation for the model on the TF values, which could be confounding this analysis.

We apologize if this was not clear. We did not use ‘log-likelihood values’ and then switch to a binary representation. As noted in our previous response to Reviewer 3/4 and the corresponding section of the methods:

“For heterotypic analysis, we evaluated every pair of the 10 activating motifs, isolating cases in which the element possessed 0 counts of both TFs, 1 count of one TF or the other, or 1 count each of the first and second TF. Again, all elements were filtered to those with ≥ 1 site to any of the other top 10 TFs other than the TF pair considered. To further account for confounding effects that could be attributable to all other TFs (i.e., including those beyond the top 10), we computed the residuals from a linear model which considered the log-likelihood values for all other TFs besides the pair of TFs under consideration. We call these “Adjusted $\log_2(\text{RNA/DNA})$ ” (e.g., y-axis of

Fig. 3h) because they removed variability explained by the binding affinities and occurrences of other TFs.”

To break down the meaning of this section, described another way: we first utilized the binary representation to stringently filter the data to isolate elements harboring the first TF alone, the second TF alone, or both TFs together, removing the subset containing the most dominant TFs. We note that this was our original approach and already highly conservative in removing the possibility that other TFBSs confound the analysis. However, in our last revision, we acknowledged that the reviewers are correct that, in principle, this leaves open the possibility that less dominant TFs, and dominant TFs with weaker binding energies, still confound the analysis. Hence, we subsequently used the log-likelihood scores for *all other* TFs to adjust any linear contribution they may confer.

In summary, we have made our already conservative approach even more conservative by successively applying a filter on the binary representation (which heavily filters away sequences that could confound the analysis), followed by adjusting the remainder of the data for the influence of the binding energies of factors. In practice, the latter adjustment has a very minor numerical influence on the data because the first filter is already very stringent. Hence, it is not surprising to us that the downstream results were largely the same and robust to controlling for the additional potential confounders (binding energies and presence of other TFBSs) as the reviewers noted.

Also, in the author’s response they acknowledge that the point of the analysis is not to show that MPRAlegNet was adding value in the biological interpretation to directly analyzing the observed data, which makes me question why they are highlighting this analysis so prominently. From the perspective of validating a methodological contribution, this analysis is also not directly comparing/benchmarking to competing sequence-based prediction models, so the value from that perspective also seems limited.

In our first round of reviewer feedback, Reviewer 1 stated: “The computational modeling...falls short of highlighting the benefits of this dataset and the power of deep learning modeling. The main insights are motifs learned by MPRAnn via attribution map clustering with TF-MoDISCo-lite. In principle, this insight could likely be accomplished with standard motif analysis. MPRAnn and other neural networks can learn more complex relationships between motifs, which could potentially reveal deeper insights into cis-regulatory grammars. But that was not done here, even though more thorough *in silico* interrogation of DNNs is now standard protocol for model interpretation.”

We interpreted this critique as meaning the reviewer felt it would be a critical aspect of the paper to demonstrate the value of deep learning models beyond those of classical methods. Thus, to address this, we incorporated the new figures under question on the topic of nonlinearities captured in the deep learning model, which standard motif analysis overlooks, and which classical machine learning models would struggle expressing without arduous feature engineering. However, we also addressed this point on novel biological findings through other means noted in our first revision (excerpts of our rebuttal below):

“1. We showcase how our MPRA results that extensively tiled seven different human disease loci can identify functional regulatory elements targeting these disease-associated genes (**Fig1a, Extended Data Fig 2**), providing candidate sequences for disease diagnostics and therapeutics.

...

4. Functional validation of element activity by CRISPRi and validation of TFs engaged in regulation (**Supplementary Fig 3, Extended Data Fig 7a**).

5. Applications of our data and model to GWAS fine-mapping and genetic variant effect prediction and interpretation, specifically applied to numerous disease-associated loci (**Fig1a, Fig 5, Extended Data Fig 2, Extended Data Fig 8, Extended Data Fig 9**).”

Thus, we are not strongly wedded to the nonlinear findings being prominently displayed in the main figure (**Fig. 3e-h**) if the consensus among the reviewers and editor(s) is that it isn't necessary to have addressed the reviewers' initial concern. However, given that we invested a lot of time and effort performing these complex analyses on what we felt to be an important and under-explored topic, our instinct (which fit with our interpretation of Reviewer 1's comment from the first round) was that adding it more prominently improves the quality and scope of the manuscript. Thus, we found it a worthwhile and important analysis to add given that it's a relatively recent topic of investigation in other deep learning models (e.g., Kim et al., *Nat Genetics* 2022 and de Almeida et al., *Nat Genetics* 2022).